# Privacy-Protected Causal Survival Analysis Under Distribution Shift

**Yi Liu**[1], **Alexander W. Levis**[2], **Ke Zhu**[1,3], **Shu Yang**[1], **Peter B. Gilbert**[4], **Larry Han**[4,5*]

[1]North Carolina State University    [2]University of Pennsylvania    [3]Duke University
[4]Fred Hutchinson Cancer Center    [5]Northeastern University

## Abstract

Causal inference across multiple data sources can improve the generalizability and reproducibility of scientific findings. However, for time-to-event outcomes, data integration methods remain underdeveloped, especially when populations are heterogeneous and privacy constraints prevent direct data pooling. We propose a federated learning method for estimating target site-specific causal effects in multi-source survival settings. Our approach dynamically re-weights source contributions to correct for distributional shifts, while preserving privacy. Leveraging semiparametric efficiency theory under a site-specific exchangeability assumption, data-adaptive weighting and flexible machine learning, the method achieves double robustness, and it improves efficiency if at least one source site provides a consistent estimate. Through simulations and two real data applications: (i) multi-site randomized trials of monoclonal antibodies for HIV-1 prevention among cisgender men and transgender persons in the United States, Brazil, Peru, and Switzerland, as well as women in sub-Saharan Africa, and (ii) an analysis of sex disparities across biomarker groups for all-cause mortality using the "flchain" dataset, we demonstrate the validity, efficiency gains, and practical utility of the approach. Our findings highlight the promise of federated methods for efficient, privacy-preserving causal survival analysis under distribution shift.

## 1 Introduction

Data fusion matters in applications with time-to-event outcomes, such as clinical studies of progression or readmission. Integrating survival data across sources can improve efficiency, particularly for rare events, but naïve pooling can be invalid under shifts in covariates, outcomes, or censoring. In addition, time-stamped event histories are considered identifiable information under General Data Protection Regulation (GDPR) and Health Insurance Portability and Accountability Act (HIPAA) regulations, limiting cross-institution data sharing. Federated learning provides a practical alternative by enabling collaboration through aggregate-level statistics rather than raw survival trajectories.

We consider multiple right-censored survival datasets each with two treatment groups, with restrictions on data sharing, and possible heterogeneity in covariates, outcomes, and censoring. Our goal is to estimate the survival function for a given target site while borrowing information from the additional source sites in a federated learning-based approach. We use an adaptive weighting strategy that aims to asymptotically exclude biased sources while retaining efficiency gains when at least one source provides a consistent estimate (see Corollary 2.11).

**Related work.**    A growing literature studies data fusion for causal inference (Yang & Ding, 2019; Li & Luedtke, 2023; Colnet et al., 2024), including recent advances in federated data fusion where full data sharing across sites is not permitted (Han et al., 2025; 2024; 2023; Xiong et al., 2023; Rémi et al., 2025; Li et al., 2023; Makhija et al., 2024; Almodóvar et al., 2024). Most of these works focus on continuous, ordinal, or binary outcomes and do not address time-to-event data. Archetti et al. (2023) have begun examining federated survival settings but they focus on data generation and simulation frameworks rather than estimation and inference.

Existing extensions to survival outcome data often rely on restrictive assumptions. For example, the Cox proportional hazards model imposes a log-linear hazard structure (Hernán, 2010; Han, 2023;

---

[*]Corresponding author: Larry Han; `lar.han@northeastern.edu`

Nagpal et al., 2023), or common conditional outcome distribution (CCOD) assumption across sites (Lee et al., 2022; Cao et al., 2024; Wen et al., 2025) may fail under heterogeneous data distributions. Violations of these assumptions yield biased estimates and inference. Related meta-analysis approaches, which aggregate site-specific estimators using inverse-variance weighting, possibly after density-ratio correction, also implicitly require such conditional homogeneity assumptions across sites (DerSimonian & Laird, 1986; Marín-Martínez & Sánchez-Meca, 2010). In addition, privacy-preserving methods avoiding sharing raw data across sites for survival outcomes remain scarce (Jia et al., 2021). Recent work such as FedECA (Ogier du Terrail et al., 2025) develops federated external control arms for single-arm trials, but this setting differs from the more general multi-source integration problem.

Regarding estimation, time-to-event data are typically analyzed within single-site studies using nonparametric methods such as the Kaplan–Meier estimator (Kaplan & Meier, 1958). With covariate-rich data, semiparametric extensions such as the Cox model (Cox, 1972; Xie & Liu, 2005; Bull & Spiegelhalter, 1997), doubly robust estimators (Bai et al., 2013) are standard. In addition, targeted maximum likelihood estimation (TMLE) can improve the finite-sample performance of doubly robust estimators (van der Laan & Rubin, 2006; Díaz et al., 2019), and the collaborative TMLE (C-TMLE) further enhances double/multiple robustness to model misspecification (Stitelman & van der Laan, 2010). More recently, Westling et al. (2024) integrated double machine learning (Chernozhukov et al., 2018) to flexibly estimate nuisance functions in survival analysis (Wolock et al., 2024; Cui et al., 2023; van der Laan et al., 2007). However, these methods remain focused on single-study contexts and do not address how to combine survival data across multiple sources.

**Contributions.** Recognizing that pooling is often infeasible and that CCOD may not hold, we develop a federated estimator with adaptive site weighting that accommodates both continuous- and discrete-time outcomes. Our approach leverages influence function theory to construct site-specific estimators based only on local summary statistics, combined through a constrained convex optimization that upweights informative sites and downweights or excludes biased ones. We establish consistency, asymptotic normality, and conditions under which our method improves efficiency over target-only analysis. By integrating cross-fitting (Chernozhukov et al., 2018) and ensemble learning (Díaz et al., 2019; Díaz, 2020; Westling et al., 2024; van der Laan et al., 2007), our estimator avoids restrictive assumptions while retaining fast convergence rates.

We validate the method through extensive Monte Carlo simulation studies and two real applications: (i) multi-site randomized trials of monoclonal antibodies for HIV-1 prevention among cisgender men and transgender persons in the United States, Brazil, Peru, and Switzerland, as well as women in sub-Saharan Africa, and (ii) an analysis of sex disparities in all-cause mortality using the `flchain` dataset in the `survival` R package, stratified into biomarker-defined groups. Together, these examples highlight the potential of federated methods to enable efficient, privacy-preserving causal inference for time-to-event outcomes in realistic multi-source settings.

## 2 METHODOLOGY

### 2.1 PROBLEM SETUP AND TARGET ESTIMAND

**Observed data.** Consider $K$ studies, each of which may be randomized or observational. For each participant, we observe baseline covariates $\mathbf{X}$, a binary treatment $A \in \{0, 1\}$, and right-censored outcomes. Let $T^{(a)}$ and $C^{(a)}$ denote the potential event and censoring times under treatment $a \in \{0, 1\}$. By the stable unit treatment value assumption (SUTVA) (Rosenbaum & Rubin, 1983), the observed event and censoring times are $T = AT^{(1)} + (1 - A)T^{(0)}$, $C = AC^{(1)} + (1 - A)C^{(0)}$. With right censoring, however, we only observe $Y = \min(T, C)$ and $\Delta = \mathbb{I}(T \leq C)$.

Denote a copy of the independent and identically distributed (i.i.d.) data by $\mathcal{O}$. The observed data across all sites are then given by $\{\mathcal{O}_i = (\mathbf{X}_i, A_i, Y_i, \Delta_i, R_i) : i = 1, \ldots, n\}$, where $R \in \{0, 1, \ldots, K - 1\}$ denotes the site, with $R = 0$ indicating the target site and $R = 1, \ldots, K - 1$ the external sources.

**Target estimand.** Throughout, $\mathbb{P}$ denotes the population-level probability under the true data-generating process, and with a subscript "$n$", $\mathbb{P}_n[f(\mathcal{O})] = n^{-1} \sum_{i=1}^{n} f(\mathcal{O}_i)$ denotes the empirical average. Our goal is to estimate the treatment-specific survival function in the target population over

a finite horizon $\tau < \infty$:

$$\theta^0(t, a) = \mathbb{P}(T^{(a)} > t \mid R = 0), \quad a \in \{0, 1\}, \ t \in [0, \tau].$$

This function gives the probability that a target-site individual on treatment $a$ ($a = 1$ for treated, $a = 0$ for control) survives beyond time $t$.

**Conditional survival functions.** For each site $k$, define the conditional survival function $S^k(t \mid a, \mathbf{X}) = \mathbb{P}(T > t \mid A = a, \mathbf{X}, R = k)$. To simultaneously accommodate continuous- and discrete-time outcomes, we use the product integral representation (Gill & Johansen, 1990):

$$S^k(t \mid a, \mathbf{X}) = \prod_{(0,t]} \{1 - \Lambda^k(du \mid a, \mathbf{X})\},$$

where $\Lambda^k(t \mid a, \mathbf{X})$ is the conditional cumulative hazard function. This notation unifies both discrete and continuous-time survival models, because in discrete time the product integral becomes the standard discrete product $\prod$, and in continuous time it becomes $\exp\{-\Lambda^k(t \mid a, \mathbf{X})\}$.

We impose three standard assumptions for causal survival analysis:

**Assumption 2.1** (Unconfoundedness). $A \perp\!\!\!\perp T^{(a)} \mid \mathbf{X}, R$ and $A \perp\!\!\!\perp C^{(a)} \mid \mathbf{X}, R$.

**Assumption 2.2** (Treatment-specific non-informative censoring). $C^{(a)} \perp\!\!\!\perp T^{(a)} \mid A = a, \mathbf{X}, R$.

**Assumption 2.3** (Positivity). There exists $\eta > 0$ such that $\mathbb{P}(R = k) \geq 1/\eta$, and for almost all $\mathbf{X}$,

$$\min_{k=0,\dots,K-1} \{\pi^k(a \mid \mathbf{X}), \ G^k(t \mid a, \mathbf{X})\} \geq 1/\eta, \quad \min_k S^k(t \mid a, \mathbf{X}) > 0.$$

Here $\pi^k(a \mid \mathbf{X}) = \mathbb{P}(A = a \mid \mathbf{X}, R = k)$ is the site-specific propensity score for treatment $A = a$, and $G^k(t \mid a, \mathbf{X}) = \mathbb{P}(C > t \mid A = a, \mathbf{X}, R = k)$ the conditional survival function of censoring. Each treatment and censoring mechanism has non-vanishing probability, and each site contributes a non-negligible fraction of participants. These quantities are referred to as nuisance functions, auxiliary components that are not of primary scientific interest but are essential for estimating the target parameter $\theta^0(t, a)$.

## 2.2 SINGLE-SITE ESTIMATION

**Auxiliary process.** For later use, define

$$\mathcal{H}_{t,a}(\mathcal{O}; S^k, G^k) = \frac{\mathbb{I}(Y \leq t, \Delta = 1)}{S^k(Y \mid a, \mathbf{X})G^k(Y \mid a, \mathbf{X})} - \int_0^{t \wedge Y} \frac{\Lambda^k(du \mid a, \mathbf{X})}{S^k(u \mid a, \mathbf{X})G^k(u \mid a, \mathbf{X})}, \quad (1)$$

where $t \wedge Y = \min(t, Y)$. This functional plays a role as an inverse probability-weighted mean-zero residual (part of an augmentation term) in doubly robust estimators for right-censored data.

**Efficient influence function (EIF).** When using only target-site data ($R = 0$), the nonparametric EIF of $\theta^0(t, a)$ given $t \in [0, \tau]$ and $a \in \{0, 1\}$ is given by (Westling et al., 2024):

$$\varphi_{t,a}^{*0}(\mathcal{O}; \mathbb{P}) = \frac{\mathbb{I}(R = 0)}{\mathbb{P}(R = 0)} \left[ \left\{ 1 - \frac{\mathbb{I}(A = a)}{\pi^0(a \mid \mathbf{X})} \mathcal{H}_{t,a}(\mathcal{O}; S^0, G^0) \right\} S^0(t \mid a, \mathbf{X}) - \theta^0(t, a) \right].$$

Here, $\mathbb{P}$ in $\varphi_{t,a}^{*0}(\mathcal{O}; \mathbb{P})$ indicates that the EIF depends on nuisance functions under the true data distribution. In other words, $\varphi_{t,a}^{*0}(\mathcal{O}; \mathbb{P}) = \varphi_{t,a}^{*0}(\mathcal{O}; S^0, G^0, \pi^0)$. Furthermore, we use $\widehat{\mathbb{P}}$ to denote the EIF evaluated with estimated nuisance functions. This should not be confused with the empirical average $\mathbb{P}_n$ introduced earlier. The same convention applies to all other EIFs throughout the paper.

The EIF $\varphi_{t,a}^{*0}(\mathcal{O}; \mathbb{P})$ highlights two components: (i) an *anchor term* that $S^0(t \mid a, \mathbf{X}) - \theta^0(t, a)$, which anchors estimation through the conditional survival function under an outcome model by using target data; and (ii) an *augmentation term*—the weighted part involving $\mathcal{H}_{t,a}(\mathcal{O}; S^0, G^0)$ and $\pi^0(a \mid \mathbf{X})$, which adjusts for censoring and treatment assignment. Furthermore, the weighting term $\mathbb{I}(R = 0)/\mathbb{P}(R = 0)$ selects target-site observations, and $\mathbb{I}(A = a)/\pi^0(a \mid X)$ restricts to units with treatment $A = a$ while reweighting them to represent the full target population.

**Target-only estimator.** Motivated by the EIF, we define $\widehat{\theta}_n^0(t, a)$ as the solution to the estimating equation

$$0 = \mathbb{P}_n[\widehat{\varphi}_{t,a}^{*0}(\mathcal{O}; \widehat{\mathbb{P}})].$$

Under regularity conditions, $\widehat{\theta}_n^0(t, a)$ is regular and asymptotically linear (RAL) and achieves the semiparametric efficiency bound uniformly over $t \in [0, \tau]$ when only target-site data are available.

## 2.3 THE CCOD ASSUMPTION

When multiple data sources are available, precision can be improved by data fusion. A common simplifying assumption is that conditional survival functions are identical across sites given covariates.

**Assumption 2.4** (Common conditional outcome distribution). $T^{(a)} \perp\!\!\!\perp R \mid \mathbf{X}$ for $a \in \{0, 1\}$.

Assumption 2.4 implies that $S^k(t \mid a, \mathbf{X}) = \bar{S}(t \mid a, \mathbf{X}) \equiv \mathbb{P}(T > t \mid A = a, \mathbf{X})$ for all $k$, while still allowing shifts in the covariate distribution $\mathbf{X}$ across sites, i.e., adjusted for covariates, the event-time distribution no longer depends on the site.

Figure 1 illustrates the data structure through a directed acyclic graph (DAG), depicting the relationships among covariates $\mathbf{X}$, treatment $A$, site indicator $R$, event time $T$, and censoring time $C$, and compares scenarios with and without the CCOD assumption.

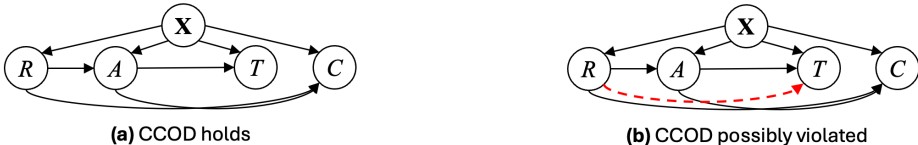

**(a)** CCOD holds          **(b)** CCOD possibly violated

Figure 1: Data structures under and without CCOD. **(a)** Under CCOD, site $R$ and event time $T$ are conditionally independent given treatment $A$ and covariates $\mathbf{X}$. **(b)** When CCOD is possibly violated, indicated by the red dashed arrow, $R$ and $T$ may not be conditionally independent.

## 2.4 FEDERATED ESTIMATION UNDER DISTRIBUTION SHIFTS AND PRIVACY

**Motivation.** In many settings, pooling individual-level data across sites is infeasible due to privacy constraints. At the same time, CCOD may fail, so naïve pooling is invalid. Still, some sites may provide information that improves estimation for the target population. We propose a federated method that adaptively re-weights source sites using only summary-level information.

### 2.4.1 LOCAL SITE-LEVEL ESTIMATION

For each source $k$, we posit a "site-$k$ CCOD" assumption, $S^k(t \mid a, \mathbf{X}) = S^0(t \mid a, \mathbf{X})$ almost surely, in order to derive the EIF. This assumption is only for formulating site-level estimators; its violation will be tested and corrected by adaptive weighting in Section 2.4.2.

**Theorem 2.5.** *For each $k \in \{0, 1, \ldots, K-1\}$, $\theta^0(t, a)$ is a pathwise differentiable parameter given $(t, a) \in [0, \tau] \times \{0, 1\}$. Under the site-$k$ CCOD assumption, the semiparametric EIF is given by*
$\varphi_{t,a}^{*k,0}(\mathcal{O}; \mathbb{P}) =$

$$\underbrace{\frac{\mathbb{I}(R = 0)}{\mathbb{P}(R = 0)} \{S^0(t \mid a, \mathbf{X}) - \theta^0(t, a)\}}_{\text{Anchoring term using target data}} - \underbrace{\frac{\mathbb{I}(R = k)}{\mathbb{P}(R = k)} \omega^{k,0}(\mathbf{X}) S^k(t \mid a, \mathbf{X}) \frac{\mathbb{I}(A = a)}{\pi^k(a \mid \mathbf{X})} \mathcal{H}_{t,a}(\mathcal{O}; S^k, G^k)}_{\text{Augmented term using source data}},$$

*where $\omega^{k,0}(\mathbf{X}) = \mathbb{P}(\mathbf{X} \mid R = 0)/\mathbb{P}(\mathbf{X} \mid R = k)$ is a density ratio comparing covariate distributions between the target site and source site $k$.*

We prove Theorem 2.5 in Appendix E.1. With the derived EIF, each site computes a source-site estimator $\widehat{\theta}_n^{k,0}(t, a)$ by solving $0 = \mathbb{P}_n[\widehat{\varphi}_{t,a}^{*k,0}(\mathcal{O}; \widehat{\mathbb{P}})]$.

**RAL property.** A central result of this paper is the regular and asymptotically linear (RAL) property of the local estimator $\widehat{\theta}_n^{k,0}(t, a)$, stated in the following theorem. An estimator is RAL if it can be written as an i.i.d. average of influence functions plus a negligible remainder. This property allows the central limit theorem to be applied to obtain its asymptotic normal distribution. Below, we use $(\pi_\infty^k, \omega_\infty^{k,0}, G_\infty^k, S_\infty^k)$ to denote the probability limits of the estimated nuisance functions $(\widehat{\pi}^k, \widehat{\omega}^{k,0}, \widehat{G}^k, \widehat{S}^k)$ for estimating $\theta^0(t, a)$ using data from source site $k$. These limits may differ from the nuisance truths $(\pi^k, \omega^{k,0}, G^k, S^k)$.

**Theorem 2.6.** *Under Conditions E.1–E.3 in Appendix E.1 with $(\pi_\infty^k, \omega_\infty^{k,0}, G_\infty^k, S_\infty^k) = (\pi^k, \omega^{k,0}, G^k, S^k)$, $\sqrt{n}(\widehat{\theta}_n^{k,0}(t, a) - \theta^0(t, a)) \to_d \mathcal{N}(0, \mathbb{P}[(\varphi_{t,a}^{*k,0})^2]), \text{for } (t, a) \in [0, \tau] \times \{0, 1\}$.*

We prove Theorem 2.6 in Appendix E.1, but summarize the regularity conditions here. Condition E.1 requires nuisance estimators to converge to well-defined limits; Condition E.2 bounds these limits and their estimates from extreme values (0 or infinity); and Condition E.3 controls product-type errors. They ensure that $\widehat{\theta}_n^{k,0}(t,a)$ converges to some well-defined limits and is asymptotically normal.

**Theoretical novelty.** Theorem 2.6 establishes the semiparametric efficiency bound under the site-$k$ CCOD assumption in 2.5, and the source-site estimators attain this bound under this assumption. These results add theoretical novelty to prior work on continuous outcomes (Han et al., 2025). The interactions between the density ratio and the other nuisance functions also represent previously unexplored theoretical components.

**Remark 2.7** (Density ratio model)**.** To estimate the density ratio while respecting data-sharing constraints, a common approach is to adopt the exponential tilt model detailed in Han et al. (2025): $\omega^{k,0}(\mathbf{X}) = \exp(\boldsymbol{\gamma}_k' \psi(\mathbf{X}))$, where $\boldsymbol{\gamma}_k$ is the model parameter and $\psi(\cdot)$ is a set of basis functions of the covariates. A simple choice is $\psi(\mathbf{X}) = \mathbf{X}$ for a linear component, and higher-order terms can be added to capture non-linearities in estimating $\omega^{k,0}$. To estimate each $\boldsymbol{\gamma}_k$ via maximum likelihood, only the target-site sample mean of $\psi(\mathbf{X})$ needs to be shared with the source sites. In addition to this model, more flexible nonparametric or machine learning approaches may be used, but these typically require sharing covariance matrices and/or other higher dimensional summaries. Thus, greater model flexibility comes at the cost of sharing more information. Finally, while the $\omega^{k,0}$ model may be misspecified, this does not necessarily invalidate our framework or estimators. As noted in Theorem 2.8, our estimator is doubly robust: under Condition E.3, errors in estimating $\omega^{k,0}$ influence the final estimator only through a product-type term that enters the second-order remainder term.

**Theorem 2.8** (Double robustness)**.** *For consistency of $\widehat{\theta}_n^{k,0}(t,a)$ under site-$k$ CCOD, it is not necessary that $(\pi_\infty^k, \omega_\infty^{k,0}, G_\infty^k, S_\infty^k) = (\pi^k, \omega^{k,0}, G^k, S^k)$ in Theorem 2.6 must hold. Instead, at any single time point $t$, if either (i) the conditional survival model $S^k$; or (ii) other nuisance functions $G^k$, $\pi^k$ and $\omega^{k,0}$ are correctly specified, $\widehat{\theta}_n^{k,0}(t,a)$ is consistent.*

A more technical version of Theorem 2.8 (Remark E.4) and its proof are presented in Appendix E.1.

### 2.4.2 Aggregation across sites

**Data-adaptive weighting.** We define the site-specific discrepancy measure $\widehat{\chi}_{n,t,a}^{k,0} = \widehat{\theta}^{k,0}(t,a) - \widehat{\theta}^0(t,a)$ and the weight vector $\boldsymbol{\eta}_{t,a} = (\eta_{t,a}^0, \eta_{t,a}^1, \ldots, \eta_{t,a}^{K-1})$. To aggregate information, we solve an $\ell_1$-penalized convex optimization problem: we minimize $Q(\boldsymbol{\eta}_{t,a})$, where

$$Q(\boldsymbol{\eta}_{t,a}) = \mathbb{P}_n \left[ \left\{ \widehat{\varphi}_{t,a}^{*0}(\mathcal{O}; \widehat{\mathbb{P}}) - \sum_{k=1}^{K-1} \eta_{t,a}^k \widehat{\varphi}_{t,a}^{*k,0}(\mathcal{O}; \widehat{\mathbb{P}}) \right\}^2 \right] + \frac{1}{n} \lambda \sum_{k=1}^{K-1} |\eta_{t,a}^k| (\widehat{\chi}_{n,t,a}^{k,0})^2, \qquad (2)$$

subject to $\eta_{t,a}^k \geq 0$ and $\sum_{k=0}^{K-1} \eta_{t,a}^k = 1$; $\lambda$ is a tuning parameter that controls the bias-variance trade-off and is chosen by cross-validation.

**Interpretation.** The objective function balances two goals: aligning site-level EIFs with the target distribution and excluding sites that would induce bias (possibly due to site-$k$ CCOD violation). The quadratic term ensures that sites well-aligned with the target survival distribution contribute more to the estimation, while the $\ell_1$ penalty induces sparsity by driving the weights of misaligned sites exactly to zero. This contrasts with an $\ell_2$ penalty, which merely shrinks weights without fully removing them. As a result, the procedure asymptotically includes only the informative sources.

**Federated estimator.** The final estimator is obtained as a weighted average of the estimated local survival curves: $\widehat{\theta}_n^{\text{fed}}(t,a) = \sum_{k=0}^{K-1} \widehat{\eta}_{t,a}^k \widehat{\theta}_n^{k,0}(t,a)$. The variance of $\widehat{\theta}_n^{\text{fed}}(t,a)$ can be estimated from its influence function, with the explicit formula given in Appendix E.2.

**Remark 2.9.** We summarize the procedure of the federated method in Algorithm 1 and illustrate its flow in Figure 2. Implementation details can be found in Appendix D, including for the cross-fitting procedure for nuisance fitting. Figure 2 emphasizes that our approach follows a *federated learning* paradigm (McMahan et al., 2017). Importantly, all steps require only summary-level transmission, never raw participant data. Source sites receive only the target-site $S^0$ model parameters and summary statistics for the density-ratio model, and the leading analysis center receives only EIFs. This contrasts

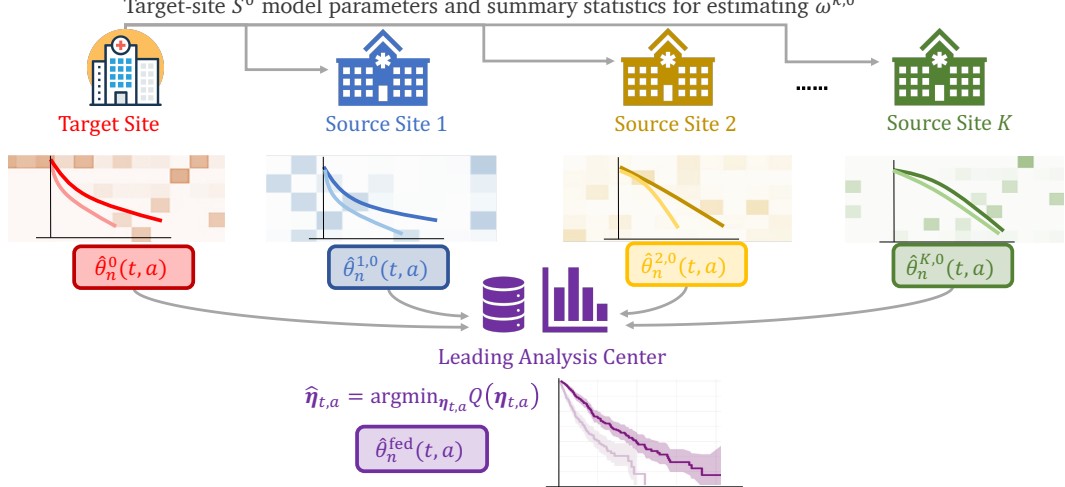

Figure 2: Algorithm flow. Target-site $S^0$ model parameters and summary statistics for estimating $\omega^{k,0}$ are transmitted to source sites; each site estimates its survival functions locally; EIFs are sent and aggregated in a leading analysis center to compute federated weights by minimizing the $Q(\cdot)$.

---

**Algorithm 1** Federated Learning for Multi-Source Causal Survival Analysis.

---

1: **Input:** Multi-source right-censored data $\{\mathcal{O}_i = (\mathbf{X}_i, A_i, Y_i, \Delta_i, R_i), i = 1, \ldots, n\}$, a time horizon $\tau > 0$; a fine time grid $\{0, \epsilon, 2\epsilon, \ldots, \tau\}$ for $[0, \tau]$ with a small $\epsilon > 0$; and the number of disjoint folds into which the data are split, $M$.

2: **Output:** Estimated treatment-specific survival curves $\widehat{\theta}_n^{\text{fed}}(t, a)$ and its estimated variance $\widehat{\mathcal{V}}_{t,a}^{\text{fed}}$ for $a \in \{0, 1\}$ and $t \in \{0, \epsilon, 2\epsilon, \ldots, \tau\}$.

3: **for** $(t, a) \in \{0, \epsilon, 2\epsilon, \ldots, \tau\} \times \{0, 1\}$ **do**

4:      Estimate the EIFs via an $M$-fold local cross-fitting (see full detail in Algorithm 2).

5:      Obtain local estimates $\widehat{\theta}_n^{k,0}(t, a)$ as solutions of $0 = \mathbb{P}_n[\widehat{\varphi}_{t,a}^{k,0}(\mathcal{O}; \widehat{\mathbb{P}})]$, for $k = 0, \ldots, K-1$.

6:      Obtain the site-specific discrepancy measure (difference of the target and source estimators) as
$$\widehat{\chi}_{n,t,a}^{k,0} = \mathbb{P}_n\left[\widehat{\varphi}_{t,a}^{k,0}(\mathcal{O}; \widehat{\mathbb{P}}) - \widehat{\varphi}_{t,a}^0(\mathcal{O}; \widehat{\mathbb{P}})\right], \text{ for } k = 1, \cdots K-1.$$

7:      Solve for treatment- and time-specific weights $\widehat{\boldsymbol{\eta}}_{t,a} = (\widehat{\eta}_{t,a}^0, \widehat{\eta}_{t,a}^1, \ldots, \widehat{\eta}_{t,a}^{K-1})$ that minimizes

$$Q(\boldsymbol{\eta}_{t,a}) = \mathbb{P}_n\left[\left\{\widehat{\varphi}_{t,a}^{*0}(\mathcal{O}; \widehat{\mathbb{P}}) - \sum_{k=1}^{K-1} \eta_{t,a}^k \widehat{\varphi}_{t,a}^{*k,0}(\mathcal{O}; \widehat{\mathbb{P}})\right\}^2\right] + \frac{1}{n}\lambda \sum_{k=1}^{K-1} |\eta_{t,a}^k|(\widehat{\chi}_{n,t,a}^{k,0})^2,$$

     subject to $0 \leq \eta_{t,a}^k \leq 1$, for all $k \in \{0, 1, \ldots, K-1\}$ and $\sum_{k=0}^{K-1} \eta_{t,a}^k = 1$, and $\lambda$ is a tuning parameter chosen by cross-validation centrally at the leading analysis center; no additional communication between sites is required.

8: **end for**

9: **Return:**
$$\widehat{\theta}_n^{\text{fed}}(t, a) = \sum_{k=0}^{K-1} \widehat{\eta}_{t,a}^k \widehat{\theta}_n^{k,0}(t, a), \text{ and } \widehat{\mathcal{V}}_{t,a}^{\text{fed}\dagger} \text{ for } (t, a) \in \{0, \epsilon, 2\epsilon, \ldots, \tau\} \times \{0, 1\}.$$

$\dagger$: $\widehat{\mathcal{V}}_{t,a}^{\text{fed}}$ is computed based on the influence function of $\widehat{\theta}_n^{\text{fed}}(t, a)$ (see Remark E.5 in Appendix E.2).

---

with *fully decentralized learning* (Lian et al., 2017), where there is no central aggregator and sites interact directly to reach consensus. Our method also differs from *meta-analysis* (Borenstein et al., 2021), which relies only on coarse population-level summaries (such information is insufficient in our setting) and often targets the pooled population.

### 2.4.3 THEORETICAL PROPERTIES

We now summarize the asymptotic results and efficiency gain of the federated estimator; proofs are in Appendices E.1 and E.2. Below, we say source site $k$ has "site-$k$ consistency" if it provides a consistent estimator of $\theta^0(t, a)$, and we refer to the collection of all such $k$ as the oracle selection set.

**Theorem 2.10** (Asymptotic distribution). *If regularity conditions for local estimates (Conditions E.1–E.3 in Appendix E.1) and the adaptive weights $\widehat{\boldsymbol{\eta}}_{t,a}$ recover the oracle selection set, then $\widehat{\theta}_n^{fed}(t, a)$, at each $(t, a) \in [0, \tau] \times \{0, 1\}$, has asymptotic distribution*

$$\sqrt{n/\widehat{\mathcal{V}}_{t,a}^{fed}} \left\{ \widehat{\theta}_n^{fed}(t, a) - \theta^0(t, a) \right\} \to_d \mathcal{N}(0, 1).$$

*where $\widehat{\mathcal{V}}_{t,a}^{fed}$ is an influence-function-based consistent estimator for the underlying asymptotic variance of $\widehat{\theta}_n^{fed}(t, a)$ (see Appendix E.2).*

**Corollary 2.11** (Asymptotic efficiency). *The asymptotic variance $\mathcal{V}_{t,a}^{fed}$ is no greater than that of the target-only estimator $\widehat{\theta}_n^0(t, a)$. Further, if at least one source site provides a consistent estimate of $\theta^0(t, a)$, then $\widehat{\theta}_n^{fed}(t, a)$ is strictly more efficient (smaller asymptotic variance).*

**Remark 2.12** (Selection consistency). The asymptotic validity of $\widehat{\theta}_n^{\text{fed}}(t, a)$ relies on selection consistency with respect to the oracle selection set. This guarantees that post-selection inference by our variance estimator $\widehat{\mathcal{V}}_{t,a}^{\text{fed}}$ remains valid, even in the presence of biased sources.

**Remark 2.13** (Efficiency gains). Under the mild regularity conditions stated in Appendix E.2, namely compact covariate support, bounded density ratios, and finite variance–covariance matrices of the EIFs across sites, we show that our federated estimator recovers the following oracle-optimal weights:

$$\bar{\boldsymbol{\eta}}_{t,a} = \underset{\eta_{t,a}^k = 0, \, \forall k \notin \mathcal{S}_{t,a}^*}{\arg\min} \; \mathcal{V}_{t,a}^{\text{fed}}(\boldsymbol{\eta}_{t,a}),$$

where $\mathcal{S}_{t,a}^*$ denotes the oracle selection set, $\mathcal{V}_{t,a}^{\text{fed}}(\boldsymbol{\eta}_{t,a})$ denotes the asymptotic variance of the federated estimator under weight vector $\boldsymbol{\eta}_{t,a}$. The target-only estimator corresponds to the special case $\boldsymbol{\eta}_{t,a} = (1, 0, \ldots, 0)$, so its variance is no larger than that of any federated estimator. If the bias term $\widehat{\chi}_{n,t,a}^{k,0}$ remains asymptotically non-zero, then $\eta_{t,a}^k \to 0$, ensuring exclusion of biased sites. Proofs are adapted from Han et al. (2023; 2025).

We note from Corollary 2.11 that strict efficiency gains rely on site-$k$ consistency for at least some $k$. This can be achieved under site-$k$ CCOD. It is also possible that site-$k$ CCOD is violated, but site-$k$ consistency still holds by chance; in this case, our method can still yield efficiency gains.

To preserve target-anchored validity while borrowing information, our methodology anchors each source estimator to the target-site estimate and incorporates source data only through the augmented term of the EIF in Theorem 2.5. In the federation stage, we solve a global optimization problem that determines each site's contribution to the final estimator. While this resembles one-shot aggregation, the optimization operates over all source summaries and explicitly targets variance reduction while preserving the target estimand.

Although the framework allows both randomized and observational sites, many survival-data applications involve randomized treatment assignment, in which case the propensity score $\pi^0$ is known and the target-site estimator is consistent. For observational studies, it is common to have larger sample sizes that enable flexible nonparametric and machine learning methods to estimate nuisance functions with greater robustness to model misspecification.

## 3 SIMULATION STUDY

We conducted simulations to assess the performance of our federated estimator (FED) to four competing approaches: target-only (TGT), pooling (POOL), inverse variance weighting (IVW) and a meta-analytic IVW (META-IVW) estimators. TGT uses only target-site data ($R = 0$), POOL aggregates data from all sites without adjustment, IVW combines site-specific estimators using inverse-variance weights, and META-IVW applies additional covariate density–ratio correction to IVW. We refer to it as "meta-analysis" as it parallels classical IVW meta-analytic pooling, augmented to correct for covariate shift. This comparison allows us to assess the efficiency gains and double robustness properties of FED under varying degrees of site heterogeneity and against several baselines.

### 3.1 DATA GENERATING PROCESS

Under a given data generating process (DGP), we generate 500 independent datasets, with $n = \sum_{k=0}^{K-1} n_k$ observations distributed across $K = 5$ sites. The target site ($k = 0$) was fixed at $n_0 = 300$ observations, while source sample sizes were varied as $n_k \in \{300, 600, 1000\}$ for $k = 1, \ldots, 4$, representing small, moderate, and large external data. Covariates, treatments, and outcomes were generated according to the mechanisms described in Appendix B.1. The estimand truth was derived by averaging survival outcomes over a super-population of size $n_{\text{super}} = 10^8$ from the target distribution. Our main DGP reflects nearly randomized studies that are more common in practice, where each site $k$ has the propensity score $\pi^k(a \mid \mathbf{X})$ weakly associated with $\mathbf{X}$ and good overlap. We also include a scenario with $n_k = 300$ ($k = 1, 2, 3, 4$) where the target-site propensity score $\pi^0(a \mid \mathbf{X})$ is more dependent on $\mathbf{X}$ ("limited overlap") to highlight a regime with larger efficiency gains.

We modeled survival outcomes over a one-year horizon, with administrative censoring at day 200. Performance was evaluated at days 30, 60, and 90. To investigate consistency and efficiency under distribution shifts, we introduced **five** cases: (i) Homogeneous: all sites follow identical DGP; (ii) Covariate Shift: covariate distributions vary; (iii) Outcome Shift: conditional outcome distributions vary; (iv) Censoring Shift: censoring mechanisms vary; and (v) All Shifts: simultaneous covariate, outcome, and censoring vary across sites. Figure 5 in Appendix B.1 depicts survival curves under outcome and covariate shifts, illustrating how site-specific heterogeneity can affect target estimation.

### 3.2 PERFORMANCE METRICS AND RESULTS

We evaluated methods using three metrics: (i) **Bias**: assessed via boxplots of estimation bias across 500 replications, (ii) **Relative root mean square error (RRMSE)**: defined as the RMSE of a method divided by that of TGT; values below 1 indicate efficiency gains, and (iii) **Coverage probability (CP%)**: the proportion of 95% Wald-type confidence intervals containing the truth. Values near 95 indicate better inference. Details of these metrics are provided in Appendix B.2. Simulation results for all scenarios appear in Appendix B.3; here we summarize representative findings in Figure 3.

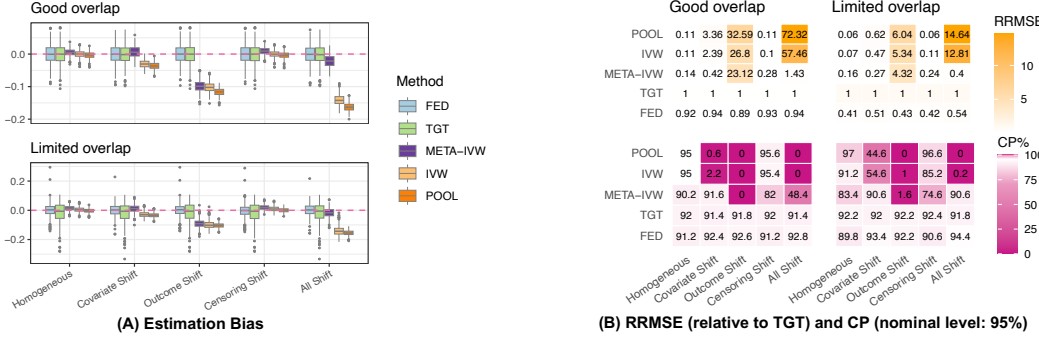

Figure 3: Simulation results for the target-site treated-arm ($A = 1$) survival function at day 30 with $n_0 = 300$ and $n_k = 300$ ($k = 1, 2, 3, 4$) under good and limited target-site propensity score overlaps.

From Figure 3(A), FED exhibits negligible bias across all scenarios. META-IVW shows small biases under Homogeneous, Covariate Shift, and Censoring Shift, but becomes substantially biased under Outcome Shift and All Shifts. In terms of efficiency, FED consistently outperforms TGT: Panel (B) shows up to 59% reductions in RMSE by FED across all settings, especially under the limited overlap scenarios. META-IVW is more efficient when the outcome does not shift, but its efficiency drops sharply under Outcome Shift. These confirm that FED preserves consistency compared to META-IVW and consistently improves efficiency relative to TGT. Across all scenarios in Appendix B.3, FED achieves larger efficiency gains at earlier time points, when site-specific survival curves more closely resemble the target (see Figure 5) and the source-site EIFs align better with the target. Under limited target-site propensity score overlap, FED also attains higher efficiency, likely because the improved overlaps at the source data help stabilize the source-site estimators.

In terms of inferential validity, both FED and TGT maintain CP% closer to 95% across scenarios, validating the influence-function-based variance estimator. Further diagnostics, reported in Figures

6 and 7, show that federated weights $\widehat{\eta}_{t,a}$ decrease systematically as site-specific bias measures $(\widehat{\chi}_{n,t,a}^{k,0})^2$ increase. Thus, FED adaptively upweights sites aligned with the target and downweights or excludes biased ones; the target site receives higher weights under covariate or outcome shifts, while contributions vary over time depending on alignment of survival functions.

Although POOL and IVW exhibit lower variability (narrower boxplots), they perform poorly under Covariate, Outcome, or All shifts: bias is substantial such that RRMSE is elevated, and CP% drops far below 95%. The exception is under Censoring Shift, but this arises because censoring is treated as a nuisance function and estimated separately within each site, reducing sensitivity to between-site heterogeneity in censoring distributions.

## 4 REAL DATA ANALYSIS

We illustrate our method with two applications. The first analyzes the coordinated antibody-mediated prevention (AMP) trials (Corey et al., 2021; Ning et al., 2023), which enrolled 4,611 participants to assess whether a bnAb reduces HIV-1 acquisition. The second uses the `flchain` dataset (7,874 participants across three biomarker-defined groups) to study sex differences in all-cause mortality. For brevity, we focus on the AMP trials and present the `flchain` results in Appendix C.2.

The AMP trials considered HIV diagnosis by week-80 as the primary endpoint, a rare event with only 3.77% incidence. Loss to follow-up was relatively low (less than 10% per treatment arm) (Corey et al., 2021). The participants were from four regions (sites): (i) **SA:** South Africa, (ii) **OA:** other sub-Saharan African countries, (iii) **BP:** Brazil or Peru, and (iv) **US:** United States or Switzerland. Participants in (i) and (ii) were women, while those in (iii) and (iv) were cisgender men or transgender individuals, reflecting population differences. Because of event sparsity, we applied a 2-fold cross-fitting. Conditional survival and censoring functions were estimated via an ensemble of Kaplan-Meier, Cox proportional hazards regression, and survival random forests from the `survSuperLearner` package (Westling et al., 2024). Propensity scores and density ratios were estimated using ensembles of logistic regression and LASSO via `SuperLearner` (van der Laan et al., 2007). Predictors included baseline age, a standardized machine-learning-derived HIV risk score, and body weight.

**Results with South Africa as target site.** We highlight main results in Figure 4. Additional analyses treating OA, BP, or US as the target, as well as comparisons of regional survival curves and baseline covariates, appear in Appendix C. Table 1 shows that OA closely resembles SA, while BP and US differ markedly in baseline risk score, weight, and HIV prevalence, consistent with covariate and outcome shifts. This pattern is reflected in the federated weights: Figure 4(B) shows SA receiving the highest weights on average, followed by OA, US, and BP.

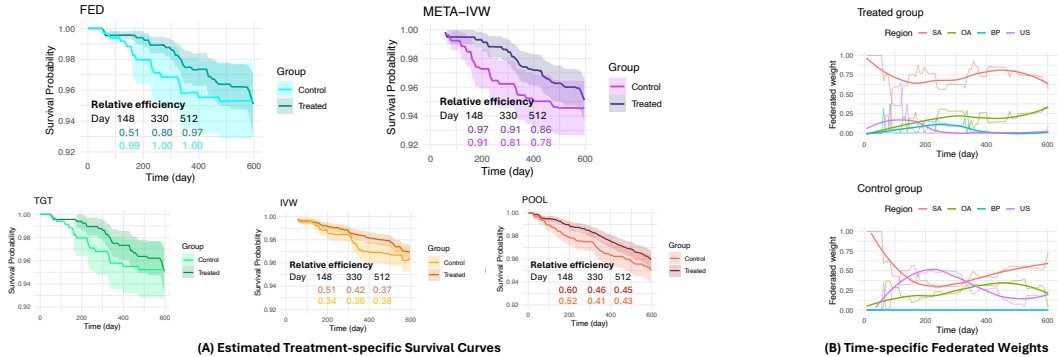

Figure 4: AMP results with SA as the target site. **(A):** Relative efficiency is the ratio of the estimated standard deviation to that of the TGT estimator, at 148, 330, and 512 days. **(B):** Federated weights with locally weighted smoothing (only a representation tool; see Cleveland & Devlin (1988)).

Figure 4(A) shows that FED, META-IVW and TGT produce similar survival curves, although META-IVW does not provide point estimates at earlier time points arising from inverses of small site-specific variances. Compared to TGT, FED offers narrower confidence intervals in some cases. In particular,

TGT fails to yield valid intervals at certain early time points due to unstable or unavailable variance estimates, driven by the insufficient effective sample size of individuals who experience the event at those times, while FED can recover intervals by borrowing information from aligned sites. These efficiency gains mirror our simulation findings, highlighting the ability of FED to improve inference without introducing bias.

Although POOL and IVW exhibit lower variance (smaller relative efficiency), they deviate from the trends of TGT and FED, suggesting bias under distribution shifts. Moreover, same as META-IVW, IVW fails to return estimates at early times due to extreme weights, underscoring a practical limitation in survival applications.

## 5 DISCUSSION

We developed a federated learning framework for estimating treatment-specific survival functions in a target population. By leveraging external sources with potentially shifted covariate and outcome distributions, while preserving privacy, our method achieves efficiency gains under oracle selection and mild regularity conditions. In the absence of timing and censoring, our estimator reduces to the FACE estimator of Han et al. (2025) when the survival outcome is replaced by the binary indicator $\mathbb{I}(T^{(a)} > \tau)$. With censoring (i.e., missingness in this binary outcome), FACE would require modification to incorporate inverse-probability weights under a missing-at-random assumption. Our method also extends to multiple non-mergeable target sites by anchoring each one separately and solving a target-specific federated aggregation problem. When target sites are comparable (e.g., satisfy CCOD), transfer learning may be leveraged to further improve nuisance estimation.

**Limitations and future directions.** Several limitations suggest opportunities for future work. First, although Theorem 2.10 and our simulations demonstrate efficiency gains, developing potentially more efficient covariate-adaptive weighting schemes remains crucial. In addition, when data sharing is permitted but the CCOD assumption fails, it is unclear whether any method—including the pooled estimator—can outperform the target-only semiparametric efficient estimator (TGT in our simulation) and our federated approach. Future work may pursue more aggressive borrowing without CCOD by allowing non-regular behavior (trading regularity for efficiency). Furthermore, while time-specific weights are flexible, they may yield non-smooth trajectories and increase computation and communication costs in continuous-time settings (communication scales with the evaluation grid size $n_\tau$ at $O(n \cdot n_\tau)$). Future work should develop smoothing strategies to capture temporal trends more efficiently and extend the framework to incorporate time-varying covariates.

Additionally, violations of the positivity assumption can render target estimand unidentifiable, e.g., when the two treatment groups in some sites differ systematically in their covariate distributions, or when certain participants are ineligible for specific treatments. Future work should investigate or leverage techniques to address such violations in our framework (Cheng et al., 2022; Xue et al., 2024). Furthermore, future work should consider settings where covariates differ across sites or have limited overlap; in such cases, density ratio estimation becomes difficult and requires additional sensitivity analysis. Finally, although our density-ratio weighting effectively addresses covariate shift, investigating alternative weighting strategies, such as extending the collaborative propensity score weighting (Guo et al., 2024) to survival data, is left for future work.

Our framework also connects to several other extensions, including incorporating alternative estimators such as TMLE and C-TMLE (van der Laan & Rubin, 2006; Stitelman & van der Laan, 2010; van der Laan & Gruber, 2010), adapting to external controls settings such as FedECA (Ogier du Terrail et al., 2025) as well as extending their inverse probability weighted-Cox approach to incorporate EIF and ensemble learning, surrogate-assisted causal inference (Han et al., 2022; Gao et al., 2024), dynamic treatment regimes (Zhang et al., 2013), and data-driven selection of external sources (Gao et al., 2025). It also opens opportunities for constructing two-sided conformalized prediction intervals for event times by leveraging the EIF–based conformal scores for survival outcomes developed (Farina et al., 2025) with federated learning for predicting missing outcomes (Liu et al., 2024a). Our approach could be adapted to other estimands such as restricted mean survival time (Han, 2023), competing risks (Lok et al., 2018) or left-truncation (Han, 2024; Wang et al., 2024).

ACKNOWLEDGMENTS

Liu is supported by the National Heart, Lung, And Blood Institute (NHLBI) of the National Institutes of Health (NIH) under Award Number T32HL079896. Yang is partially supported by National Science Foundation (NSF) grant SES 2242776. The content is solely the responsibility of the authors and does not necessarily represent the official views of the NIH and NSF.

REPRODUCIBILITY STATEMENT

**Software.** A user-friendly R package `FuseSurv` (data **fusion** for causal **survival** analysis) implementing our method is available from the first author's GitHub: `https://github.com/yiliu1998/FuseSurv`.

**Programming.** All simulation studies and real data analyses were performed using the statistical language `R` (version 4.4.2). The dependent `R` packages include: `CFsurvival`, `survSuperLearner`, `superLearner` (version 2.0.29), `glmnet` (version 4.1.8), `caret` (version 6.0.94) and `tidyverse` (version 2.0.0). To enhance computational efficiency, parallel computing packages `foreach` (version 1.5.2) and `doParellel` (version 1.0.17) were employed. The replication of simulations was carried out using 200 CPU cores by Duke Computing Cluster.

**Real data.** The two real datasets are publicly available. The AMP trial data can be found at `https://atlas.scharp.org/project/HVTN%20Public%20Data/HVTN%20704%20HPTN%20085%20and%20HVTN%20703%20HPTN%20081%20AMP/begin.view`, and the "flchain" data can be found at `https://rdrr.io/cran/survival/man/flchain.html` or by typing command `data(flchain)` in R after loading the `survival` R package.

ETHICS STATEMENT

This work complies with the ICLR Code of Ethics. We used only publicly available datasets with appropriate licenses and did not involve human subjects or sensitive personal information. We acknowledge potential risks of misuse (e.g., unfair application, misinterpretation, or unintended deployment beyond the intended research scope) and discuss limitations and safeguards in the paper. All results are reported transparently, and code will be released to support reproducibility.

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

## A   USE OF LLMS

We acknowledge the use of ChatGPT-5.0 exclusively for language polishing and grammatical corrections. No large language models (LLMs) were used for any other aspects of this work. The research ideas, conceptualization, methodology development, and all experiments are entirely original contributions of the authors.

## B   SIMULATION DETAILS AND ADDITIONAL RESULTS

### B.1   DETAILS OF DATA GENERATING PROCESS

Three covariates $X_1$, $X_2$, and $X_3$ are sampled as transformations of Beta random variables with site-specific parameters:

$$X_1 \sim 33 \cdot \text{Beta}(1.1 - 0.05\gamma(k), 1.1 + 0.2\gamma(k)) + 9 + 2\gamma(k),$$
$$X_2 \sim 52 \cdot \text{Beta}(1.5 + (X_1 + 0.5\gamma(k))/20, 4 + 2\gamma(k)) + 7 + 2\gamma(k),$$
$$X_3 \sim (4 + 2\gamma(k)) \cdot \text{Beta}(1.5 + |X_1 - 50 + 3\gamma(k)|/20, 3 + 0.1\gamma(k)),$$

where $\gamma(k)$ represents some function of site $k$, specified later. We then generate the treatment assignment probabilities $\pi(\mathbf{X})$ using the logistic function:

$$\text{logit}(\pi(\mathbf{X})) = -1.05 + \log\left(1.3 + \exp(-12 + X_1/10) + \exp(-2 + X_2/3) + \exp(-2 + X_3/12)\right)$$

and treatments $A$ are sampled as $A \sim \text{Bernoulli}(\pi(\mathbf{X}))$. For the scenario with limited propensity score overlap in the target site, we modify the target-site propensity score model $\pi(\mathbf{X})$ to such that

$$\text{logit}(\pi(\mathbf{X})) = -1.05 + \log\left(0.3 + \exp(-120 + X_1) + \exp(-6 + X_2) + \exp(-6 + X_3/4)\right),$$

and generate $A \sim \text{Bernoulli}(\pi(\mathbf{X}))$ accordingly for target-site samples only. This increases the dependence of $A$ on $\mathbf{X}$ and induces reduced overlap.

Next, we consider the mechanisms of event and censoring times. The hazard rates for event times and censoring times are given by the following $\exp(h_t)$ and $\exp(h_c)$, respectively, where $h_t = -5.02 + 0.1(X_1 - 25) - 0.1(X_2 - 25) + 0.05(X_3 - 2) + D_T(k) \cdot 0.1(X_2 - 25) + A \cdot \delta_T(k) \cdot 0.1(X_1 + X_2 + X_3 - 50)$, and $h_c = -4.87 + 0.01(X_1 - 25) - 0.02(X_2 - 25) + 0.01(X_3 - 2) - D_C(k) \cdot 0.1(X_2 - 25) + A \cdot \delta_C(k) \cdot 0.1(X_1 + X_2 + X_3 - 50)$.

Here, $D_T(k)$, $D_C(k)$, $\delta_T(k)$ and $\delta_C(k)$ are some site-specific indicators, specified later, for varying the treatment effects and trends of survival curves for different sites. Then, event times and censoring times are sampled as:

$$T = \left(-\frac{\log(U_1)}{\exp(h_t) \cdot \lambda}\right)^{1/\rho}, \quad C = \left(-\frac{\log(U_2)}{\exp(h_c) \cdot \lambda}\right)^{1/\rho},$$

with $\rho = 1.2$, $\lambda = 0.6$, and $U_1, U_2 \sim \text{Uniform}(0, 1)$. This technique follows Austin (2012). Thus, the observed times and event indicators are $Y = \min(T, C)$, $\Delta = \mathbb{I}(T \leq C)$, respectively.

Under this data generating process (DGP), the event time is generated to mimic days in a year (365 days), and we truncate the censoring time at $\tau = 200$ days to mimic the end of follow-up in survival analysis. Our DGP allows the following scenarios based on site-specific distributional heterogeneity:

- **Homogeneous**: Homogeneous covariates and hazard rates across sites. We let $\gamma(k) = D_T(k) = D_C(k) = \delta_T(k) = \delta_C(k) = 0$ for $k = 0, 1, \ldots, 4$.
- **Covariate Shift**: Covariates $X_1$, $X_2$, and $X_3$ vary across sites. We let $\gamma(k) = k$ and $D_T(k) = D_C(k) = \delta_T(k) = \delta_C(k) = 0$, for $k = 0, 1, \ldots, 4$.
- **Outcome Shift**: Conditional outcome distribution varies across sites. We assign $\gamma(k) = 0$, $D_T(k) = \delta_T(k) = k$, and $D_C(k) = \delta_C(k) = 0$ for $k = 0, 1, \ldots, 4$.
- **Censoring Shift**: Censoring mechanism varies across sites. We let $\gamma(k) = 0$, $D_T(k) = \delta_T(k) = 0$ and $D_C(k) = \delta_C(k) = k$, for $k = 0, 1, \ldots, 4$.
- **All Shift**: Covariates and both event and censoring effects vary across sites. We let $\gamma(k) = D_T(k) = D_C(k) = \delta_T(k) = \delta_C(k) = k$, for $k = 0, 1, \ldots, 4$.

Figure 5 below plots the true treatment-specific survival curves under the Covariate Shift and Outcome Shift scenarios, as defined by our designed DGPs, to illustrate the effect of site differences on survival outcomes.

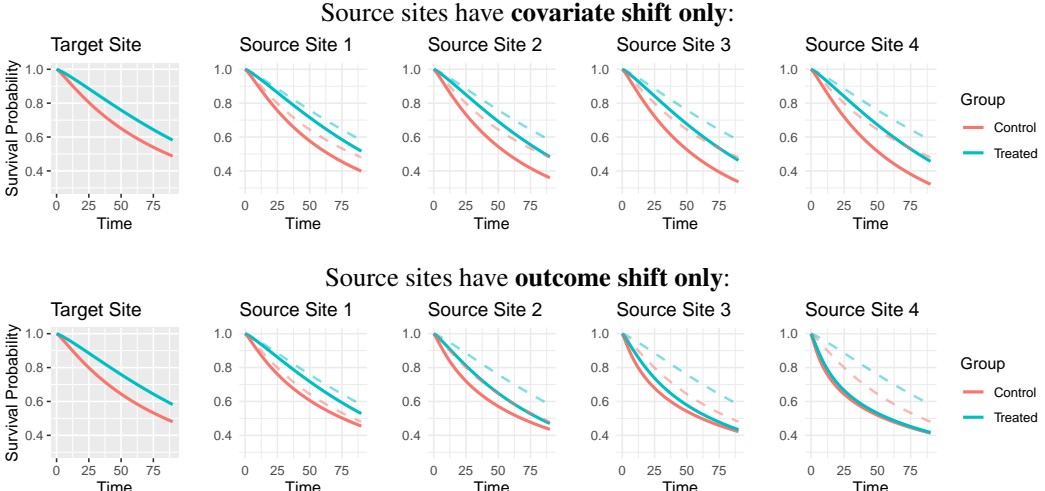

Figure 5: Site- and treatment-specific survival curves, each based on a random sample of $n = 10^4$ from the true DGP of each site. The two dashed curves in each source site panel are the target site survival functions for reference. Under covariate shift, curves preserve their shapes and trends but differ in scale, whereas outcome shift produces marked changes in shape and treatment effects.

## B.2 PERFORMANCE CRITERIA DEFINITIONS

The simulation performance criteria considered in Section 3.2 with an additional metric 95% confidence interval (CI) width for the complete simulation results are defined as follows.

Let $\theta$ denote the true target parameter, and let $\widehat{\theta}_i$ and $\widehat{\sigma}_i$ be the point and standard error estimates, respectively, from the $i$th Monte Carlo replication of a competing method, $i = 1, \ldots, 500$. Then:

- **Estimation bias:** $\widehat{\theta}_i - \theta$, $i = 1, \ldots, 500$, summarized via boxplots;

- **RRMSE:** the RMSE of a method relative to that of the TGT estimator, where RMSE = $\sqrt{500^{-1} \sum_{i=1}^{500} (\widehat{\theta}_i - \theta)^2}$. By definition, the TGT estimator has RRMSE = 1. Smaller RRMSE values indicate higher efficiency relative to TGT;

- **CP%:** the proportion of replications in which the Wald-type CI contains $\theta$: $100\% \times 500^{-1} \sum_{i=1}^{500} \mathbb{I}\{\theta \in [\widehat{\theta}_i - 1.96\widehat{\sigma}_i, \ \widehat{\theta}_i + 1.96\widehat{\sigma}_i]\}$. The closer CP% is to 95, the more reliable the inference based on $\widehat{\sigma}_i$; and

- **95% CI width:** the average CI width across replications, where the CI from the $i$th replication is $\widehat{\theta}_i \pm 1.96, \widehat{\sigma}_i$. Thus, CI width = $3.92 \times 500^{-1} \sum_{i=1}^{500} \widehat{\sigma}_i$.

## B.3 COMPLETE SIMULATION RESULTS

Figures 6–7 summarize the results comparing the federated and source-site shifts, as well as the corresponding discrepancy values $(\widehat{\chi}_{n,t,a}^k)^2$. Figures 8–9 report the full simulation results under good target-site propensity score for varying source sample sizes. Figure 11 shows the corresponding results under limited target-site overlap, where FED consistently achieves higher relative efficiency compared with TGT.

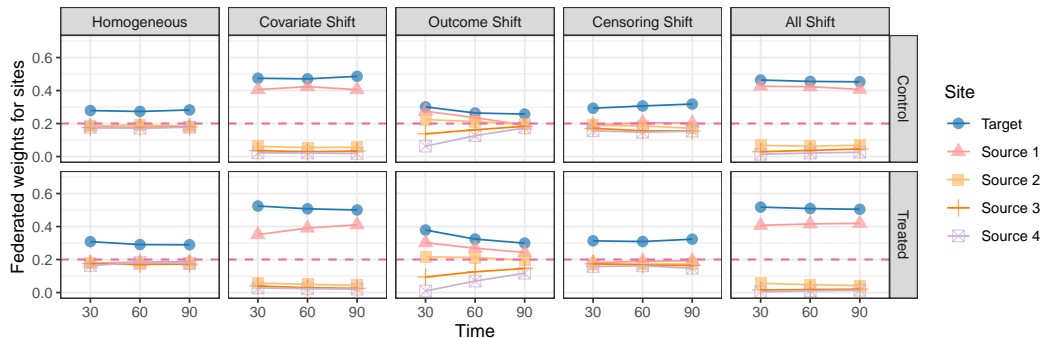

Figure 6: Average federated weights of each site at different time point by site heterogeneity cases. This figure uses the case where $n_k = 300 \ (k \geq 1)$ as an illustration for weights.

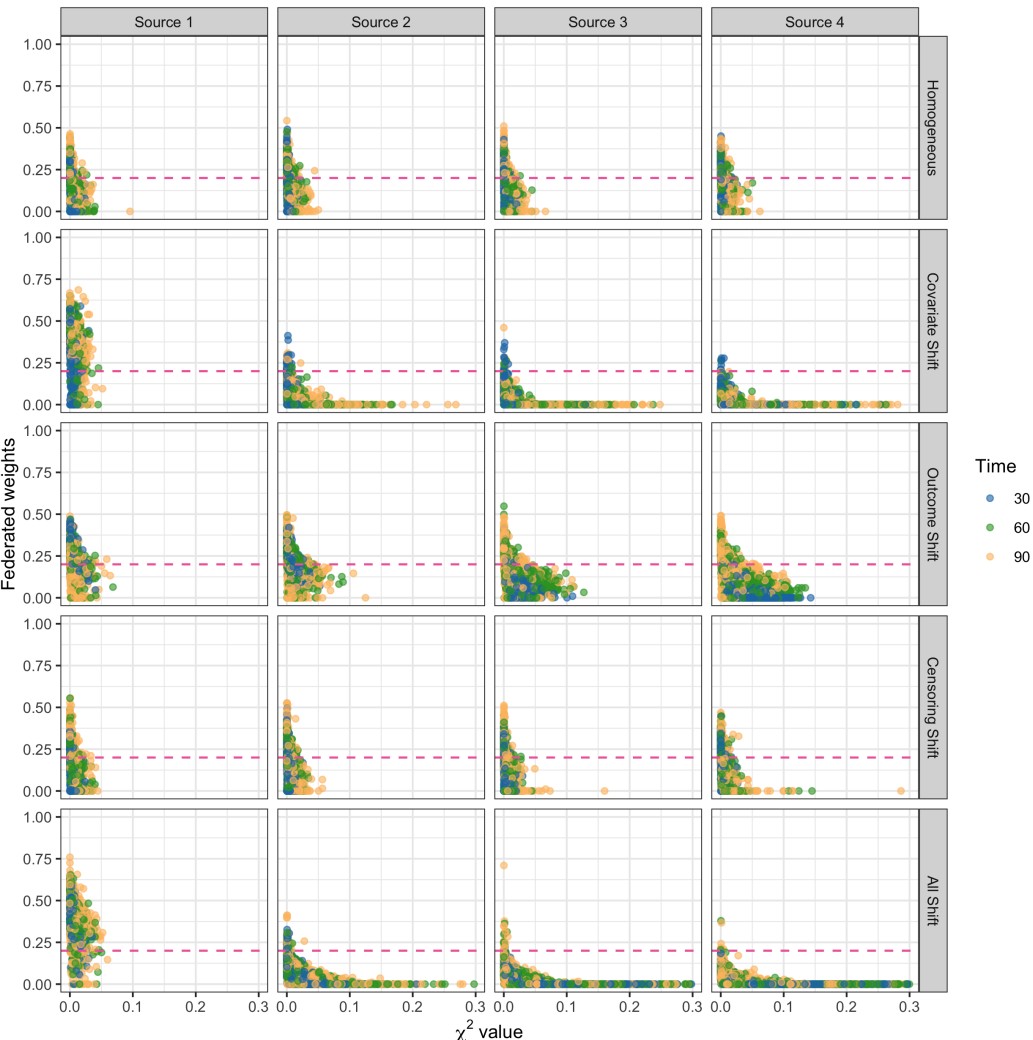

Figure 7: Scatter plots of site-specific federated weights vs. discrepancy measure $(\widehat{\chi}_{n,t,a}^k)^2$ values, under 5 scenarios of site heterogeneity and 3 selected time points (days 30, 60 and 90). Sites 2–4 under Covariate Shift and All Shift have more larger $(\widehat{\chi}_{n,t,a}^k)^2$ values with clear trends of decreasing weights. The pink dashed lines indicate weight $= 1/5$, i.e., one over five sites. This figure uses the case where $n_k = 300 \ (k \geq 1)$ as an illustration for weights.

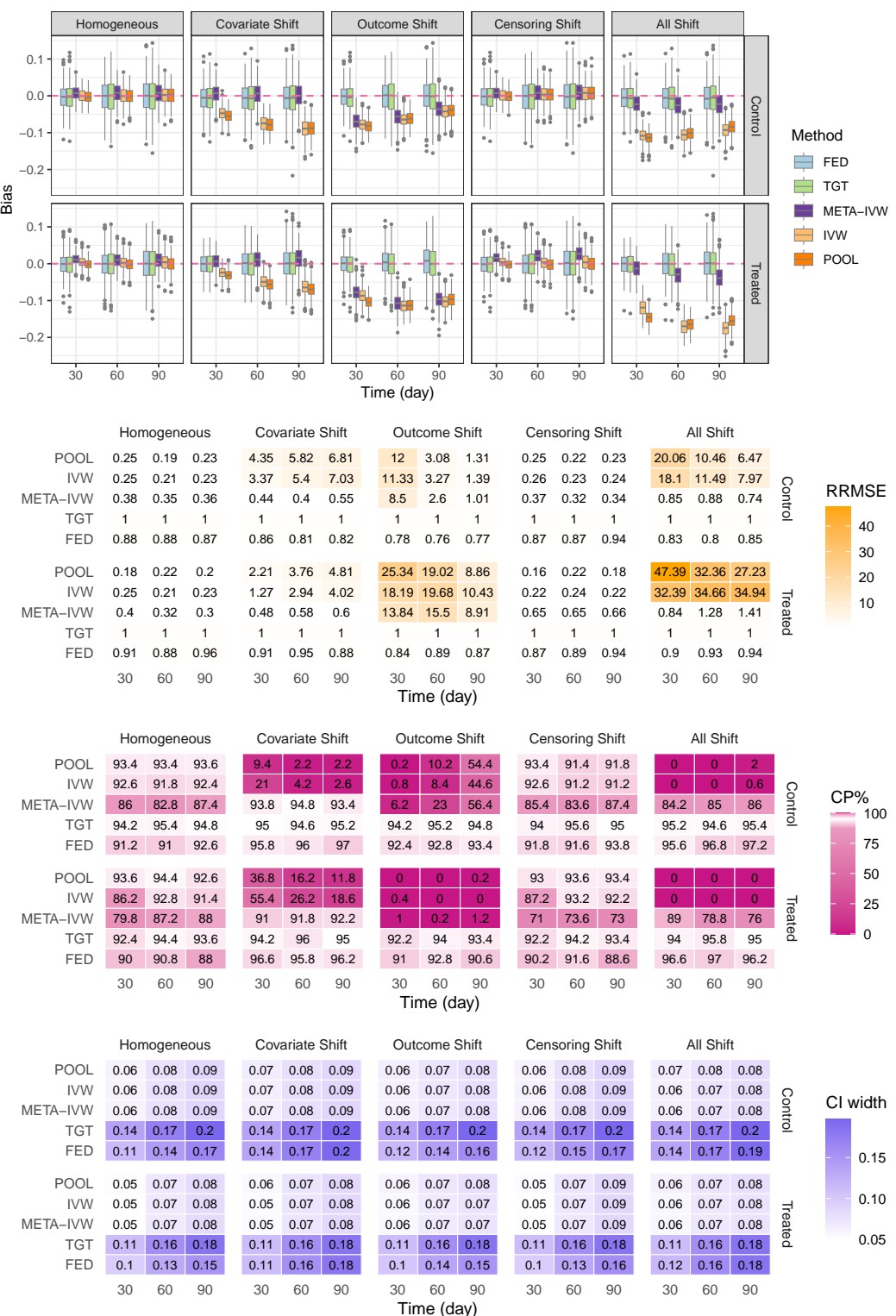

Figure 8: Estimation bias (boxplots), relative root mean square error (RRMSE) compared to TGT, coverage probability (CP%) with 95% nominal coverage level, and width of 95% CI under $n_k = 300$ ($k = 1, 2, 3, 4$), with good propensity score overlap in the target site, evaluated at days 30, 60 and 90 in simulation.

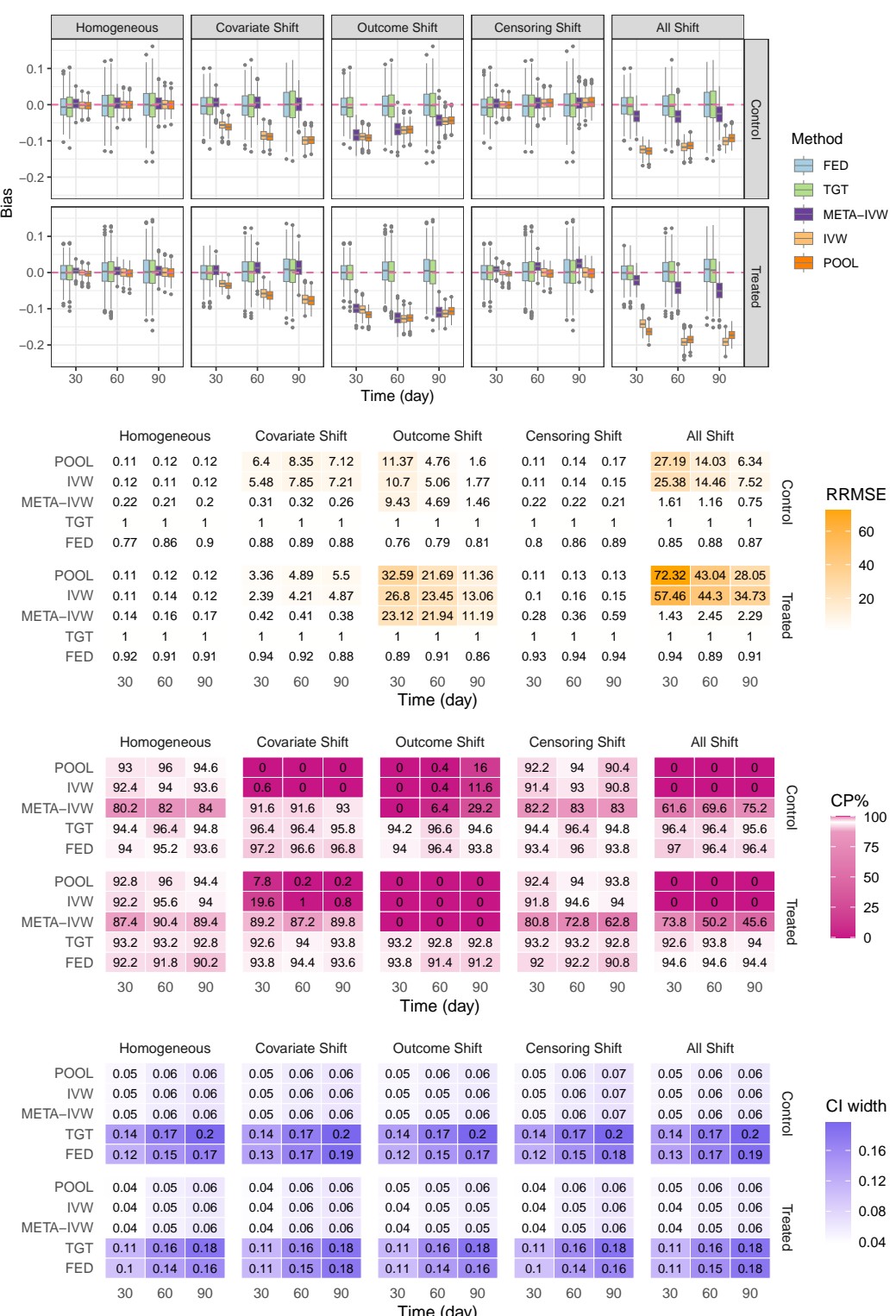

Figure 9: Estimation bias (boxplots), relative root mean square error (RRMSE) compared to TGT, coverage probability (CP%) with 95% nominal coverage level, and width of 95% CI under $n_k = 600$ ($k = 1, 2, 3, 4$), with good propensity score overlap in the target site, evaluated at days 30, 60 and 90 in simulation.

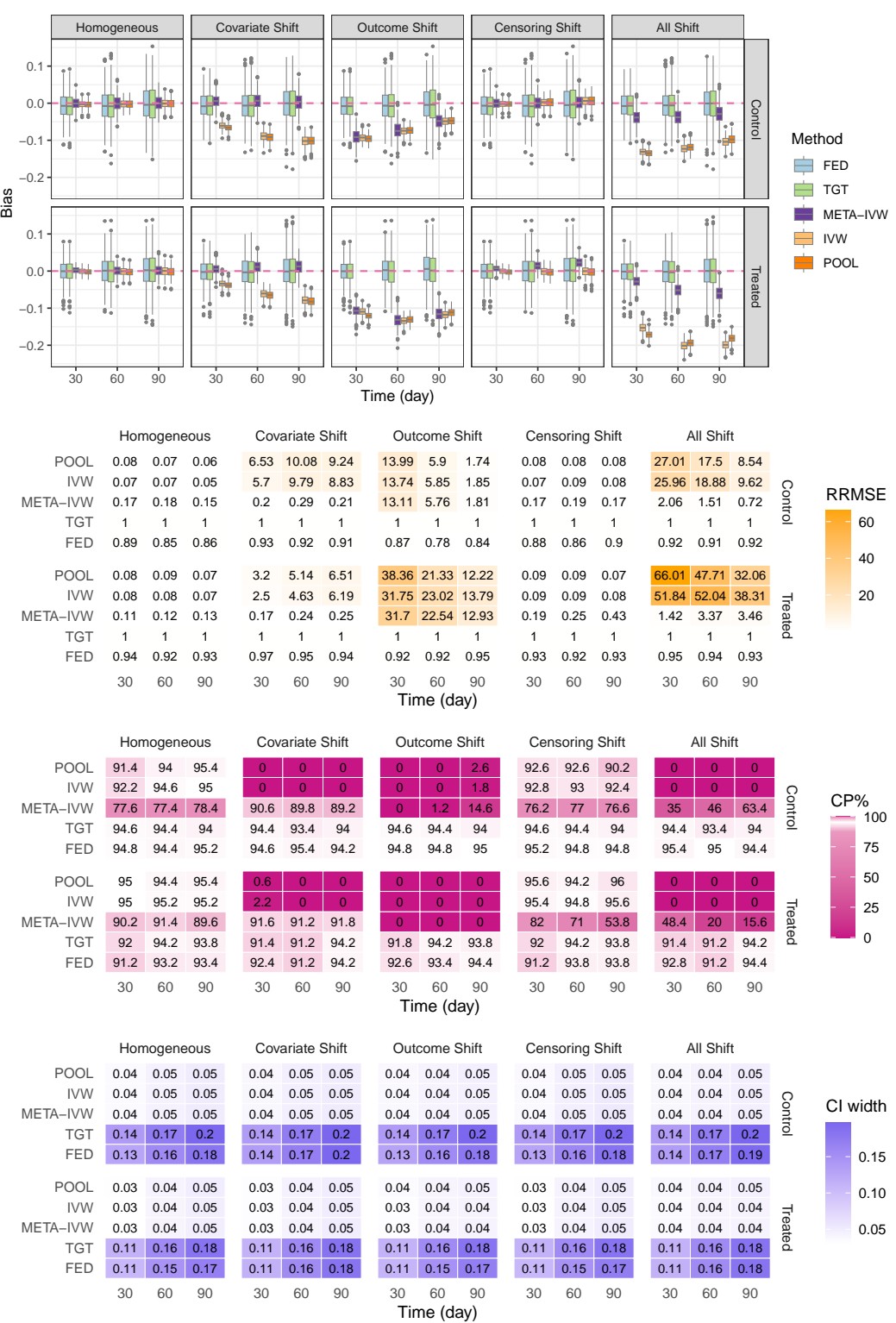

Figure 10: Estimation bias (boxplots), relative root mean square error (RRMSE) compared to TGT, coverage probability (CP%) with 95% nominal coverage level, and width of 95% CI under $n_k = 1000$ ($k = 1, 2, 3, 4$), with good propensity score overlap in the target site, evaluated at days 30, 60 and 90 in simulation.

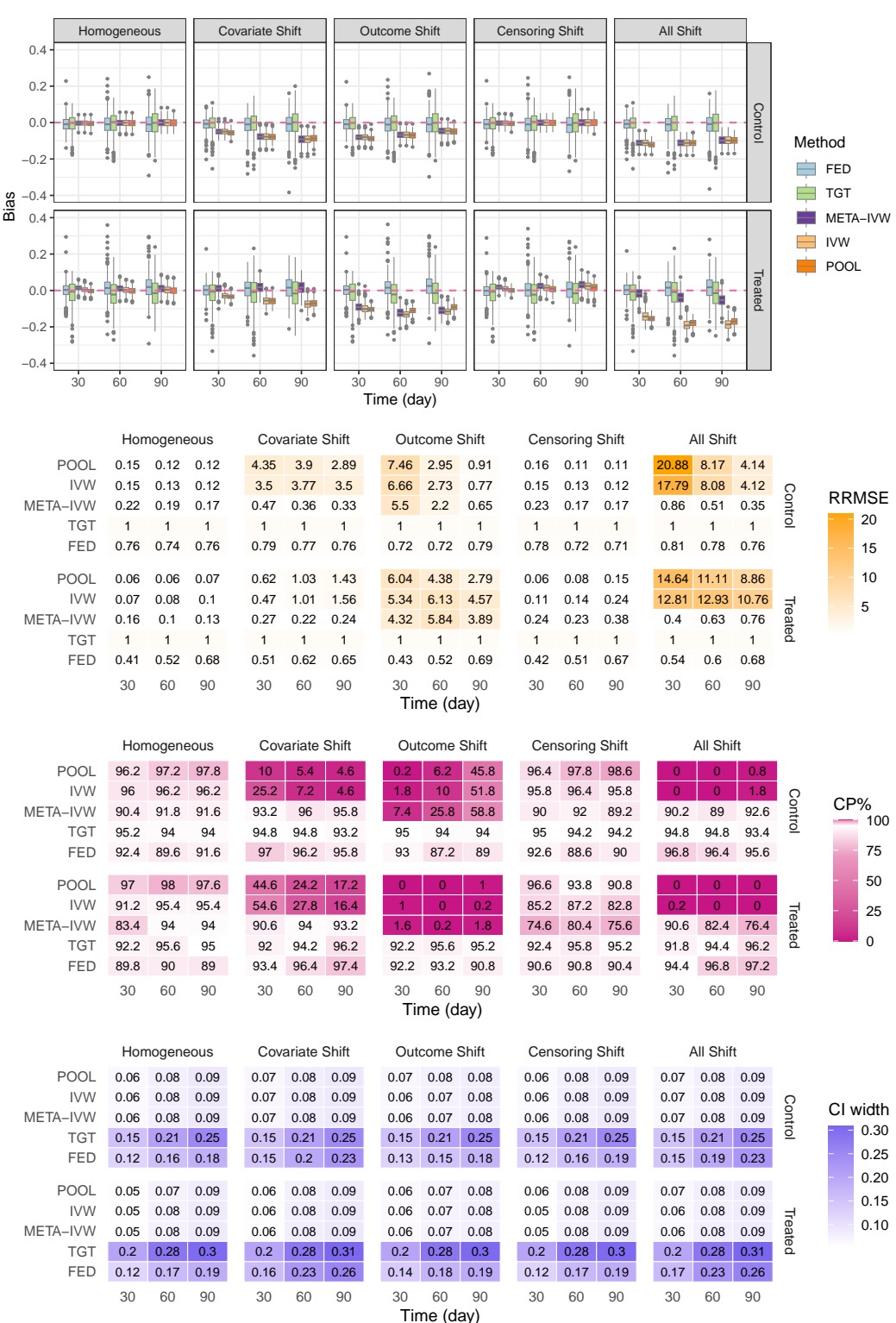

Figure 11: Estimation bias (boxplots), relative root mean square error (RRMSE) compared to TGT, coverage probability (CP%) with 95% nominal coverage level, and width of 95% CI under $n_k = 300$ ($k = 1, 2, 3, 4$), with limited propensity score overlap in the target site, evaluated at days 30, 60 and 90 in simulation.

## C Additional Results for Real Data Analysis

### C.1 AMP trial data

Table 1 presents summary statistics for baseline covariates and outcomes in the AMP trial data, stratified by region and treatment group. Comparing the treatment groups—both overall and within each region—we observe that the treated group consistently shows a lower average event proportion. Additionally, some covariates appear to shift across regions; for example, among treated participants, the standardized risk scores exhibit notably different means when comparing SA to BP and US.

| | Treated (bnAb) group | | | | |
| --- | --- | --- | --- | --- | --- |
| | **Total** ($n = 3,076$) | **SA** ($n = 679$) | **OA** ($n = 608$) | **BP** ($n = 846$) | **US** ($n = 943$) |
| Age (year) at baseline | 25.9 (4.60) | 27.0 (5.19) | 25.4 (4.59) | 25.1 (3.70) | 26.2 (4.68) |
| Standardized risk score | 0.0 (1.00) | -0.01 (1.00) | 0.02 (1.00) | 0.76 (0.67) | -0.68 (0.71) |
| Weight at baseline (kg) | 72.8 (15.64) | 68.8 (14.24) | 65.2 (12.63) | 70.9 (12.42) | 82.3 (16.43) |
| HIV diagnosis by week-80 | 107 (3.48%) | 27 (3.98%) | 20 (3.29%) | 46 (5.44%) | 14 (1.49%) |
| | Control (placebo) group | | | | |
| | **Total** ($n = 1,535$) | **SA** ($n = 340$) | **OA** ($n = 297$) | **BP** ($n = 428$) | **US** ($n = 470$) |
| Age (year) at baseline | 25.9 (4.72) | 26.6 (5.28) | 25.4 (4.78) | 25.2 (3.94) | 26.1 (3.79) |
| Standardized risk score | 0.0 (1.00) | 0.02 (0.92) | -0.02 (0.98) | 0.75 (0.67) | -0.68 (0.73) |
| Weight at baseline (kg) | 72.5 (16.35) | 67.6 (14.77) | 65.1 (13.64) | 71.1 (12.84) | 81.8 (17.5) |
| HIV diagnosis by week-80 | 67 (4.36%) | 16 (4.71%) | 13 (4.38%) | 29 (6.78%) | 9 (1.91%) |

Table 1: Summary statistics of AMP trial data by treatment group and region. The standardized risk score is a baseline score built by machine learning models (Corey et al., 2021) that is predictive to the time-to-event outcome. Age, standardized risk score and weight are summarized by mean (standard deviation), while the HIV diagnosis by week-80 is summarized by count (percentage).

In Figure 12, we plot the region-specific survival curves of all the 4 regions we considered (SA, OA, BP and US) for a direct comparison on region heterogeneity, using their target-site-only (TGT) estimators, to showcase the heterogeneous effects of the bnAb antibody treatment on different target populations.

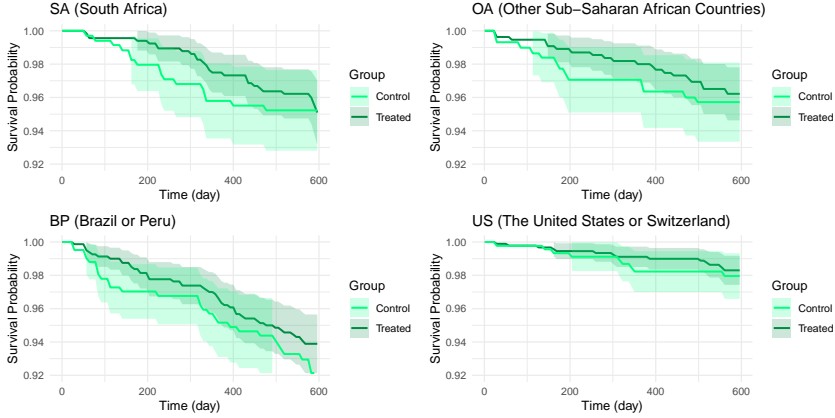

Figure 12: Estimated region-specific survival curves of the HVTN 704/HPTN 085 and HVTN 703/HPTN 081 trials. SA (our target region in the main text) and OA exhibit relatively similar curves, indicating less heterogeneity of these two regions. In contrast, both BP and US regions show significant differences to SA, which also confirms why they often have small or zero federated weights in Panel (B) of Figure 4 in the main text. The BP and US also show a substantial difference on their curves.

## OA (Other Sub-Saharan African Countries)

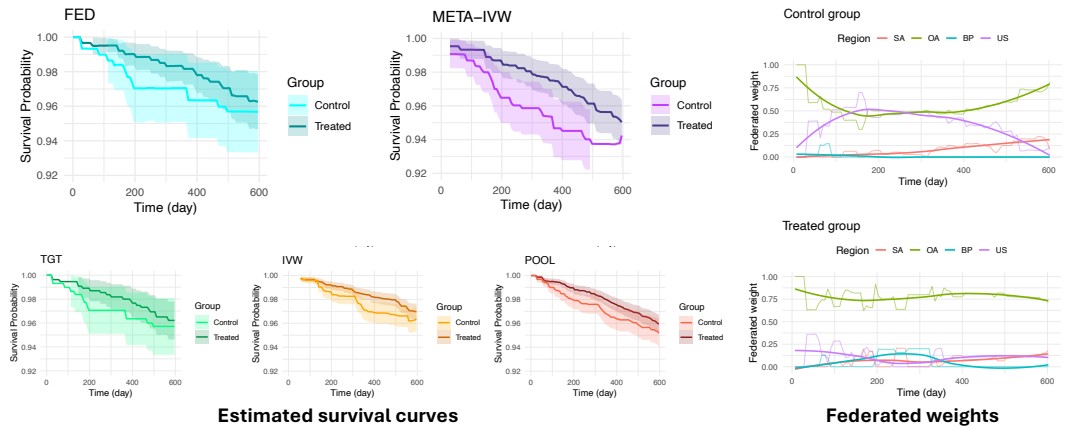

## BP (Brazil or Peru)

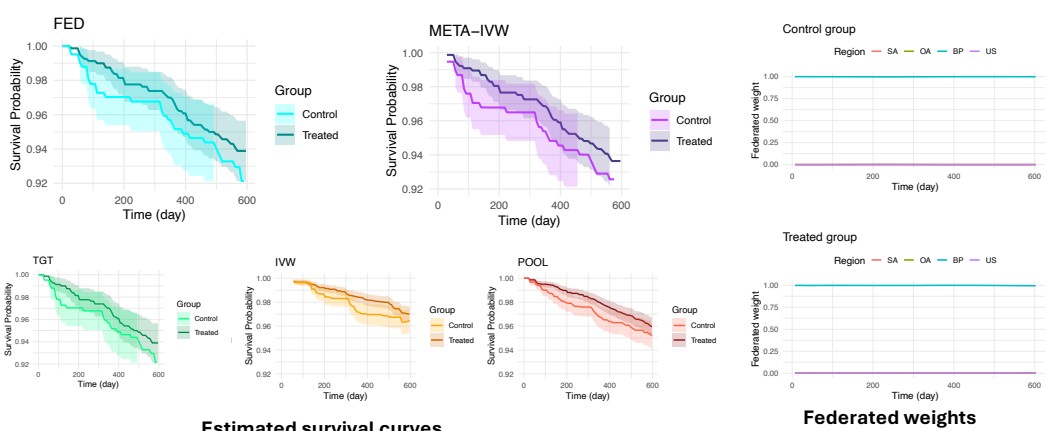

## US (The United States or Switzerland)

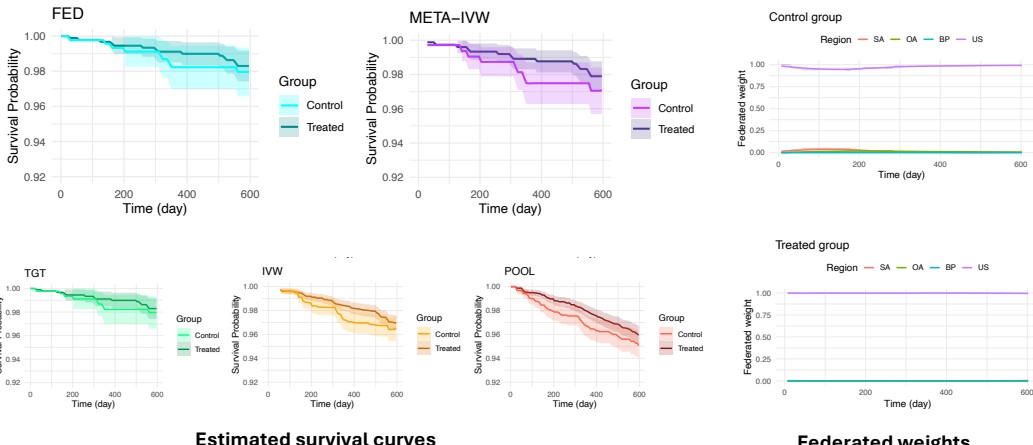

Figure 13: Additional data analysis results when treating the other three regions (OA, BP and US) as the target site. Time-specific federated weights with locally weighted smoothing (only a representation tool; Cleveland & Devlin (1988)).

Furthermore, in Figure 13, we present the results—including survival curve estimations and federated weights—using three regions other than SA as the target population. For the federated weights, similar to Figure 4 in the main text, we applied locally weighted regression (Cleveland & Devlin, 1988) to smooth the observed weights over the study period, providing a clearer visualization of temporal trends in this specific example.

From Figure 13, we observe that for each region, the FED method yields results similar to the TGT estimator, while also recovering some interval estimations at earlier time points. This finding is consistent with the observations made in Figure 4. In contrast, the IVW and POOL methods deviate noticeably from the TGT and FED results—especially for the BP and US regions—indicating potential biases introduced by site heterogeneity.

Finally, regarding federated weights, the results for the OA region resemble those of SA in Figure 4. However, for the BP and US regions, the federated weights are nearly 1 for the target site and 0 for all other sites. This pattern suggests that when targeting the survival curves of BP or US, other sites contribute substantial biases—an observation that corroborates our findings in Figure 12.

## C.2 "FLCHAIN" DATASET FROM R PACKAGE `SURVIVAL`

The "flchain" dataset, obtained from the Mayo Clinic Study of Serum Free Light Chain (FLC) and Mortality, comprises data on 7,874 individuals followed between 1995 and 2009 to investigate the prognostic value of serum free light chains for survival (Dispenzieri et al., 2012; Kyle et al., 2006). This dataset is freely available in R package `survival`.

This dataset does not contain a natural treatment variable, but to illustrate and extend the use of our framework, we investigate the sex difference in mortality. Since sex (female vs. male) is assigned at birth, it can be viewed as a "treatment" variable for methodological purposes, as it precedes the occurrence of any outcomes. While not manipulable in the conventional sense, causal inference methods allow us to frame sex as an exposure to quantify disparities in survival outcomes, rather than as an intervention subject to policy or clinical decision-making. Similar approaches have been employed to assess disparities associated with non-manipulable variables such as race (Li & Li, 2023; Liu et al., 2025; 2024b; Matsouaka et al., 2025; Li et al., 2025).

| | **Male** | | | |
| --- | --- | --- | --- | --- |
| | **Total** ($n = 3,524$) | **Group A** ($n = 972$) | **Group B** ($n = 1,429$) | **Group C** ($n = 1,123$) |
| Age (year) at baseline | 63.1 (9.62) | 60.1 (7.80) | 62.6 (9.25) | 66.4 (10.5) |
| MGUS | 0.01 (0.11) | 0.04 (0.20) | 0.00 (0.05) | 0.00 (0.00) |
| Sample year | 1996.9 (1.84) | 1996.7 (1.72) | 1996.9 (1.87) | 1996.9 (1.90) |
| Concentration of $\kappa$ light chain | 1.5 (1.01) | 0.9 (0.34) | 1.4 (0.45) | 2.2 (1.44) |
| Concentration of $\lambda$ light chain | 1.8 (1.19) | 1.1 (0.35) | 1.6 (0.47) | 2.5 (1.77) |
| Mortality | 1,004 (28.5%) | 159 (16.4%) | 372 (26.0%) | 473 (42.1%) |
| | **Female** | | | |
| | **Total** ($n = 4,350$) | **Group A** ($n = 1,399$) | **Group B** ($n = 1,771$) | **Group C** ($n = 1,180$) |
| Age (year) at baseline | 65.2 (11.01) | 62.2 (9.57) | 65.0 (10.8) | 69.1 (11.8) |
| MGUS | 0.02 (0.12) | 0.05 (0.21) | 0.00 (0.05) | 0.0 (0.00) |
| Sample year | 1996.7 (1.70) | 1996.6 (1.55) | 1996.7 (1.68) | 1996.9 (1.87) |
| Concentration of $\kappa$ light chain | 1.4 (0.78) | 0.9 (0.34) | 1.3 (0.43) | 2.1 (1.03) |
| Concentration of $\lambda$ light chain | 1.6 (0.88) | 1.1 (0.35) | 1.6 (0.46) | 2.4 (1.22) |
| Mortality | 1,165 (26.8%) | 231 (16.5%) | 455 (25.7%) | 479 (40.6%) |

Table 2: Summary statistics of "flchain" data by sex group and the site variable we defined. All baseline covariates are summarized by mean (standard deviation), while the mortality is summarized by count (percentage).

We include age, the presence of monoclonal gammopathy of undetermined significance (MGUS) and sample year as baseline covariates for nuisance models. The primary outcome consists of follow-up time in days and an event indicator for all-causes death (mortality).

A categorical variable (`flc.grp`, taking values $1, 2, \ldots, 10$) related to $\kappa$ and $\lambda$ concentration levels is available in the data. We construct the "site" variable ($R$ in our notation) based on `flc.grp` as follows: (i) Group A for `flc.grp` $\in \{1, 2, 5\}$; (ii) Group B for `flc.grp` $\in \{3, 4, 6, 9\}$; and (iii) Group C for `flc.grp` $\in \{7, 8, 10\}$. Several categories were merged in this way to ensure a sufficient sample size within each group, allowing 5-fold cross-fitting to train different nuisance functions reliably. In addition, we allow the groups to share nearby values of `flc.grp` (e.g., 5 in Group A, 6 in Group B, and 7 in Group C) so that each site retains comparable information, enabling borrowing across groups. We emphasize that this grouping method is adopted solely for illustrative purposes in demonstrating our framework.

Table 2 presents the summary statistics of baseline covariates and mortality for the "flchain" data. Across Groups A, B, and C, we observe clear covariate shifts, accompanied by differences in the marginal death rates. In contrast, when comparing the two "treatment" (sex) groups, the distributions of baseline covariates and mortality appear overall similar.

We analyzed the sex-specific survival curves over the first 10 years for the three groups in Figure 14. We used a 5-fold cross-fitting, and estimated conditional survival for both event and censoring processes by an ensemble of Kaplan–Meier, Cox regression and survival random forest models via the `survSuperLearner` package (Westling et al., 2024). The propensity score and density ratio (used in federated method) models were fitted by the ensemble of logistic regression and LASSO using the Super Learner (van der Laan et al., 2007).

Overall, the FED method yields point estimates that closely track those of the TGT estimator, while producing slightly narrower confidence bands. By calculations, the efficiency gain (by estimated standard error of FED to that of TGT) can achieve 3%–10%, consistent with the findings from both our simulation studies and the AMP trial data. The IVW and POOL estimators exhibit noticeably different survival curve patterns relative to TGT and FED when Groups A and C are regarded as targets, suggesting potential biases. The META-IVW method yields similar but slightly different curves compared to TGT and FED.

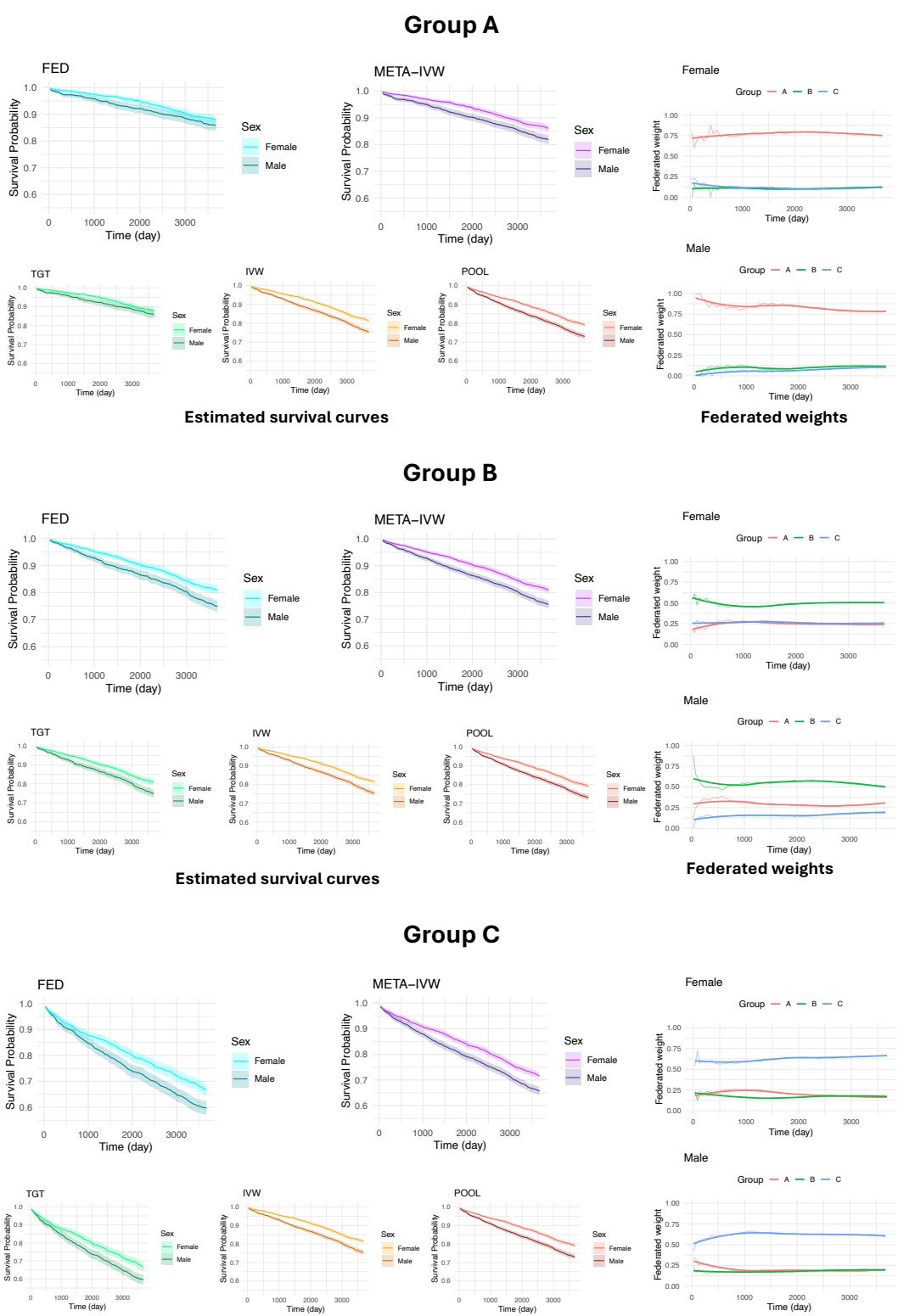

Figure 14: "flchain" data analysis results. Estimated sex-specific survival curves and federated weights for sites (Groups A, B and C defined by `flc.grp` variable) Time-specific federated weights with locally weighted smoothing (only a representation tool; Cleveland & Devlin (1988)).

# D    IMPLEMENTATION DETAILS

In the following Algorithm 2, we detail the double machine learning procedure (Wang et al., 2025; Chernozhukov et al., 2018) for fitting and predicting nuisance functions in Algorithm 1.

---

**Algorithm 2** Double/debiased machine learning algorithm for nuisance function estimations and influence function calculations in Algorithm 1 at a given time point and treatment.

---

1: **Input:** Observed multi-source right-censored data $\mathcal{O} = \{\mathcal{O}_i = (\mathbf{X}_i, A_i, Y_i, \Delta_i, R_i), i = 1, \ldots, n\} = \mathcal{O}^0 \cup \mathcal{O}^1 \cup \cdots \cup \mathcal{O}^{K-1}$, where $R_i \in \{0, 1, \ldots, K - 1\}$ and $\mathcal{O}^k$ represents the data for site $R = k$; Given treatment group $A = a$ and a specific time point $t$; The number of disjoint folds into which the data are split, $M$, where $M \in \{2, 3, \ldots, \lfloor n^*/2 \rfloor\}$ with $n^* = \min\{n, n_1, \ldots, n_{K-1}\}$.
2: **Output:** Estimated influence functions for each individual.
3: Partition $\mathcal{O}^0$ into $M$ approximately equal-sized, disjoint validation folds $\mathcal{V}_1^0, \ldots, \mathcal{V}_M^0$, allowing a size difference of at most $\pm 1$ between folds.
4: **for** $m = 1, \ldots, M$ **do**
5:     Define the training set $\mathcal{T}_m^0 = \mathcal{O}^0 \backslash \mathcal{V}_m^0$;
6:     Fit nuisance functions $S^0$, $G^0$, $\pi^0$ on $\mathcal{T}_m^0$, using some methods ensemble from `survSuperLearner` and `SuperLearner`;
7:     Predict nuisance functions on $\mathcal{V}_m^0$ as $\widehat{S}_m^0, \widehat{G}_m^0$ and $\widehat{\pi}_m^0$.
8: **end for**
9: Train a model of $S^0$ by the entire data of the target site $\mathcal{O}^0$, denoted as $S^{0,\text{full}}$, using chosen methods ensemble from `survSuperLearner`.
10: **for** $k = 1, \ldots, K - 1$ **do**
11:     Partition $\mathcal{O}^k$ into $M$ approximately equal-sized, disjoint validation folds $\mathcal{V}_1^k, \ldots, \mathcal{V}_M^k$, allowing a size difference of at most $\pm 1$ between folds.
12:     **for** $m = 1, \ldots, M$ **do**
13:         Define the training set $\mathcal{T}_m^k = \mathcal{O}^k \backslash \mathcal{V}_m^k$;
14:         Fit the density ratio $\omega^{k,0}$ using only covariate data of $\mathcal{T}_m^0 \cup \mathcal{T}_m^k$, or by just passing through some coarsening level summary statistics;
15:         Fit nuisance functions $G^k$, $\pi^k$ on $\mathcal{T}_m^k$, using chosen methods ensembles from `survSuperLearner` and `SuperLearner`;
16:         Predict above nuisance functions on $\mathcal{V}_m^k$ as $\widehat{G}_m^k, \widehat{\omega}_m^{k,0}$ and $\widehat{\pi}_m^k$;
17:         Predict nuisance function $S^k$ on $\mathcal{V}_m^k$ using the pre-trained $S^{0,\text{full}}$ model, and denote the predicted value by $\widehat{S}_m^k$.
18:     **end for**
19:     Aggregate all predicted nuisance functions over $M$ folds as $\widehat{S}^k, \widehat{G}^k, \widehat{\omega}^{k,0}$ and $\widehat{\pi}^k$;
20: **end for**
21: **Return:** The estimated EIFs, by plugging-in their predicted nuisance function values, $\widehat{\varphi}_{t,a}^{*k,0}(\mathcal{O}; \widehat{\mathbb{P}}) = \widehat{\varphi}_{t,a}^{*k,0}(\mathcal{O}; \widehat{S}^k, \widehat{S}^0, \widehat{G}^k, \widehat{\pi}^k, \widehat{\omega}^{k,0})$, for all $k \in \{0, 1, \ldots, K - 1\}$.

---

**Remark D.1.** To ensure the monotonicity of the estimated survival curves, we invoke isotonic regression techniques (Westling et al., 2020), which enforce a non-increasing constraint on the site-specific survival and censoring estimates $\widehat{S}^k$ and $\widehat{G}^k$, for $k \in \{0, 1, \ldots, K - 1\}$, thereby maintaining their logical consistency over time.

# E    TECHNICAL PROOFS

We begin by recalling notation for probability, expectation, and variance. Throughout, $\mathbb{P}$ denotes the true probability under the data-generating distribution, $\mathbb{P}_n$ the empirical average, and $\widehat{\mathbb{P}}$ the evaluation with estimated nuisance functions (as introduced in the main text). In addition, $\mathbb{E}$ denotes the population expectation, $\mathbb{V}$ the population variance, and Cov the population covariance.

We further adopt the following notation throughout this appendix: (i) $\mathbb{P}_\infty$ denotes a general probability limit, and the nuisance functions under $\mathbb{P}_\infty$ are denoted with subscript $\infty$, e.g., $S_\infty^0$ for the limit of $\widehat{S}^0$; (ii) $\widehat{\mathbb{P}}$ means the corresponding nuisance functions are replaced by their estimates, and $\widehat{\mathbb{P}}$ may

converge to a general limit $\mathbb{P}_\infty$; (iii) $\mathbb{P}_n^m[f(\mathcal{O})] = |\mathcal{V}_m|^{-1} \sum_{i \in \mathcal{I}_m} f(\mathcal{O}_i)$ to denote the empirical average on the $m$-th validation set $\mathcal{V}_m$ by cross-fitting, $m = 1, \ldots, M$.

Furthermore, we distinguish notation $\mathbb{P}(f)$ and $\mathbb{E}_\mathbb{P}(f)$: $\mathbb{P}(f) = \int f(\mathcal{O})d\mathbb{P}$ denotes an integral over a new observation $\mathcal{O} \sim \mathbb{P}$, treating $f$, which possibly depends on training data (e.g., some estimated parameters for nuisance functions), as fixed. In contrast, $\mathbb{E}_\mathbb{P}(f)$ is the usual mathematical expectation of random variable/element $f$ under distribution $\mathbb{P}$, a fixed value without randomness.

### E.1 THEORY OF THE LOCAL ESTIMATOR

#### E.1.1 PROOF OF THEOREM 2.5

Recall that a mean zero, finite variance function $\varphi_{t,a}^{*0}(\mathcal{O}; \mathbb{P})$ is called an *influence function* of the target estimand (a functional) $\theta^0(t, a) = \theta^0(t, a; \mathbb{P})$ at $\mathbb{P}$ if, for any one-dimensional regular parametric submodel $\{\mathbb{P}_\epsilon : \epsilon \in [0, 1)\}$ through $\mathbb{P} \equiv \mathbb{P}_0$,

$$\frac{\partial}{\partial \epsilon} \theta^0(t, a; \mathbb{P}_\epsilon)\bigg|_{\epsilon=0} = \mathbb{E}_\mathbb{P}[\varphi_{t,a}^{*0}(\mathcal{O}; \mathbb{P})\dot{\ell}(\mathcal{O})],$$

where $\dot{\ell}(\mathcal{O})$ is the score function of the submodel at $\epsilon = 0$ (i.e., typically, $\dot{\ell}(\mathcal{O}) = \partial \log\{p_\epsilon(\mathcal{O})\}/\partial\epsilon |_{\epsilon=0}$), where $p_\epsilon(\cdot)$ denotes the probability density (likelihood) function under submodel $\mathbb{P}_\epsilon$ (Bickel et al., 1993).

Recall the site-$k$ CCOD assumption made in Theorem 2.5, $S^0(t \mid a, \mathbf{X}) = S^0(t \mid a, \mathbf{X})$ almost surely. To find the EIF, we begin by writing the following equation:

$$0 = \frac{\partial}{\partial\epsilon} \theta^0(t, a)\bigg|_{\epsilon=0} = \frac{\partial}{\partial\epsilon} \mathbb{E}\{S_\epsilon^0(t \mid a, \mathbf{X}) \mid R = 0\}\bigg|_{\epsilon=0}$$

$$= \mathbb{E}\{[S^0(t \mid a, \mathbf{X}) - \theta^0(t, a)]\dot{\ell}_{\mathbf{X}|R=0} \mid R = 0\} + \mathbb{E}\left\{\int \frac{\partial}{\partial\epsilon} S_\epsilon^0(t \mid a, \mathbf{x})\bigg|_{\epsilon=0} \mu(d\mathbf{x}) \bigg| R = 0\right\}$$

$$= \mathbb{E}\{[S^0(t \mid a, \mathbf{X}) - \theta^0(t, a)]\dot{\ell}_{\mathbf{X}|R=0} \mid R = 0\} + \mathbb{E}\left\{\int \frac{\partial}{\partial\epsilon} S_\epsilon^k(t \mid a, \mathbf{x})\bigg|_{\epsilon=0} \mu(d\mathbf{x}) \bigg| R = 0\right\},$$

(3)

where $\mu(\cdot)$ denotes the distribution of $\mathbf{X}$ induced by $\mathbb{P}$ and, for any sets of variables $V$ and $W$, $\dot{\ell}_{V|W}$ denotes the conditional score function of $V$ given $W$, i.e., typically $\partial \log\{p_\epsilon(V \mid W)\}/\partial\epsilon |_{\epsilon=0}$—note that such scores always satisfy $\mathbb{E}_\mathbb{P}(\dot{\ell}_{V|W} \mid W) = 0$ (Bickel et al., 1993).

For the derivative of $S_\epsilon^k$ with respect to $\epsilon$, by the chain rule, we decompose it as $(\partial S_\epsilon^k/\partial\Lambda_\epsilon^k) \times (\partial\Lambda_\epsilon^k/\partial\epsilon)$. For the first part $\partial S_\epsilon^k/\partial\Lambda_\epsilon^k$, we leverage Theorem 8 in Gill & Johansen (1990). Specifically, the mapping $H \mapsto S^k(t; H) := \prod_{(0,t]}\{1 + H(du)\}$ is Hadamard differentiable at $H$ relative to the supremum norm with derivative

$$\alpha \mapsto S^k(t; H) \int_0^t \frac{S^k(u-; H)}{S^k(u; H)} \alpha(du).$$

Thus, by letting $H(t) = \Lambda_\epsilon^k(t \mid a, \mathbf{x})$ and the chain rule, the integrand in the second term becomes

$$\frac{\partial}{\partial\epsilon} \prod_{(0,t]}\{1 - \Lambda_\epsilon^k(du \mid a, \mathbf{x})\}\bigg|_{\epsilon=0} = -S^k(t \mid a, \mathbf{x}) \int_0^t \frac{S^k(u- \mid a, \mathbf{x})}{S^k(u \mid a, \mathbf{x})} \frac{\partial}{\partial\epsilon} \Lambda_\epsilon^k(du \mid a, \mathbf{x})\bigg|_{\epsilon=0}.$$

Furthermore, recall that

$$\Lambda^k(t \mid a, \mathbf{X}) = \int_0^t \frac{N_1^k(du \mid a, \mathbf{X})}{D^k(u \mid a, \mathbf{X})},$$

where $N_1^k(t \mid a, \mathbf{X}) = \mathbb{P}(Y \leq t, \Delta = 1 \mid A = a, \mathbf{X}, R = k)$ and $D^k(t \mid a, \mathbf{X}) = \mathbb{P}(Y \geq t \mid A = a, \mathbf{X}, R = k)$. Hence,

$$\frac{\partial}{\partial\epsilon} \Lambda_\epsilon^k(du \mid a, \mathbf{x})\bigg|_{\epsilon=0} = \frac{\frac{\partial}{\partial\epsilon} N_{1,\epsilon}^k(du \mid a, \mathbf{x}) |_{\epsilon=0}}{D^k(u \mid a, \mathbf{x})} - \frac{\frac{\partial}{\partial\epsilon} D_\epsilon^k(u \mid a, \mathbf{x}) |_{\epsilon=0} N_1^k(du \mid a, \mathbf{x})}{D^k(u \mid a, \mathbf{x})^2}.$$

In addition,

$$
\begin{aligned}
\frac{\partial}{\partial \epsilon} N_{1,\epsilon}^k(du \mid a, \mathbf{x})\Big|_{\epsilon=0} &= \frac{\partial}{\partial \epsilon} \mathbb{P}_\epsilon(Y \le u, \Delta = 1 \mid A = a, \mathbf{X} = \mathbf{x}, R = k)\Big|_{\epsilon=0} \\
&= \frac{\partial}{\partial \epsilon} \iint \mathbb{I}(y \le u, \delta = 1) \mathbb{P}_\epsilon(dy, d\delta \mid a, \mathbf{x}, k)\Big|_{\epsilon=0} \\
&= \iint \mathbb{I}(y \le u, \delta = 1) \dot{\ell}(y, \delta \mid a, \mathbf{x}) \mathbb{P}(dy, d\delta \mid a, \mathbf{x}, k) \\
&= \int_\delta \mathbb{I}(\delta = 1) \dot{\ell}(u, \delta \mid a, \mathbf{x}) \mathbb{P}(du, d\delta \mid a, \mathbf{x}, k),
\end{aligned}
$$

and

$$
\begin{aligned}
\frac{\partial}{\partial \epsilon} D_\epsilon^k(u \mid a, \mathbf{x})\Big|_{\epsilon=0} &= \frac{\partial}{\partial \epsilon} \mathbb{P}_\epsilon(Y \ge u \mid A = a, \mathbf{X} = \mathbf{x}, R = k)\Big|_{\epsilon=0} \\
&= \frac{\partial}{\partial \epsilon} \iint \mathbb{I}(y \ge u) \mathbb{P}_\epsilon(dy, d\delta \mid a, \mathbf{x}, k)\Big|_{\epsilon=0} \\
&= \iint \mathbb{I}(y \le u) \dot{\ell}(y, \delta \mid a, \mathbf{x}) \mathbb{P}(dy, d\delta \mid a, \mathbf{x}, k).
\end{aligned}
$$

We can then express the integrand of (3) as

$$
\begin{aligned}
&\frac{\partial}{\partial \epsilon} \iint \prod_{(0,t]} \{1 - \Lambda_\epsilon^k(du \mid a, \mathbf{x})\} \mu(d\mathbf{x})\Big|_{\epsilon=0} \\
&= \iiint -\mathbb{I}(y \le t, \delta = 1) \frac{S^k(t \mid a, \mathbf{x}) S^k(y- \mid a, \mathbf{x})}{S^k(y \mid a, \mathbf{x}) D^k(y \mid \mathbf{x})} \dot{\ell}(y, \delta \mid a, \mathbf{x}, k) \mathbb{P}(dy, d\delta \mid a, \mathbf{x}, k) \mu(d\mathbf{x}) \\
&\quad + \iiiint \mathbb{I}(u \le t, u \le y) \frac{S^k(t \mid a, \mathbf{x}) S^k(u- \mid a, \mathbf{x})}{S^k(u \mid a, \mathbf{x}) D^k(u \mid \mathbf{x})} \\
&\qquad \times \dot{\ell}(y, \delta \mid a, \mathbf{x}, k) \mathbb{P}(dy, d\delta \mid a, \mathbf{x}, k) N_1^k(du \mid a, \mathbf{x}) \mu(d\mathbf{x}) \\
&= \iiint -\mathbb{I}(y \le t, \delta = 1) \frac{S^k(t \mid a, \mathbf{x}) S^k(y- \mid a, \mathbf{x})}{S^k(y \mid a, \mathbf{x}) D^k(y \mid \mathbf{x})} \dot{\ell}(y, \delta \mid a, \mathbf{x}, k) \mathbb{P}(dy, d\delta \mid a, \mathbf{x}, k) \mu(d\mathbf{x}) \\
&\quad + \iiint S^k(t \mid a, \mathbf{x}) \int_0^{t \wedge y} \frac{S^k(u- \mid a, \mathbf{x})}{S^k(u \mid a, \mathbf{x}) D^k(u \mid \mathbf{x})^2} N_1^k(du \mid a, \mathbf{x}) \\
&\qquad \times \dot{\ell}(y, \delta \mid a, \mathbf{x}, k) \mathbb{P}(dy, d\delta \mid a, \mathbf{x}, k) \mu(d\mathbf{x}) \\
&= \mathbb{E}\bigg[ S^k(t \mid a, \mathbf{X}) \frac{\mathbb{I}(A = a)}{\pi^k(a \mid \mathbf{X})} \bigg\{ H^k(t \wedge Y, a, \mathbf{X}) - \frac{\mathbb{I}(Y \le t, \Delta = 1) S^k(Y- \mid a, \mathbf{X})}{S^k(Y \mid a, \mathbf{X}) D^k(Y \mid a, \mathbf{X})} \bigg\} \\
&\qquad \times \dot{\ell}(Y, \Delta \mid a, \mathbf{X}, R = k) \bigg],
\end{aligned}
$$

where

$$
H^k(t, a, \mathbf{x}) = \int_0^t \frac{S^k(u- \mid a, \mathbf{x}) N_1^k(du \mid a, \mathbf{x})}{S^k(u \mid a, \mathbf{x}) D^k(u \mid a, \mathbf{x})^2}.
$$

Now, we note that

$$
\mathbb{E}\left[ \frac{\mathbb{I}(Y \le t, \Delta = 1) S^k(Y- \mid A, \mathbf{X})}{S^k(Y \mid A, \mathbf{X}) D^k(Y \mid A, \mathbf{X})} \,\Big|\, A = a, \mathbf{X} = \mathbf{x}, R = k \right] = \int_0^t \frac{S^k(y- \mid a, \mathbf{x}) N_1^k(dy \mid a, \mathbf{x})}{S^k(y \mid a, \mathbf{x}) D^k(y \mid a, \mathbf{x})},
$$

and

$$\mathbb{E}\{H^k(t \wedge Y, A, \mathbf{X}) \mid A = a, \mathbf{X} = \mathbf{x}, R = k\}$$

$$= \iint^t \mathbb{I}(u \leq y) \frac{S^k(u- \mid a, \mathbf{x}) N_1^k(du \mid a, \mathbf{x})}{S^k(u \mid a, \mathbf{x}) D^k(u \mid a, \mathbf{x})^2} \mathbb{P}(dy \mid a, \mathbf{x}, k)$$

$$= \int_0^t \mathbb{P}(Y \geq u \mid A = a, \mathbf{X} = \mathbf{x}, R = k) \frac{S^k(u- \mid a, \mathbf{x}) N_1^k(du \mid a, \mathbf{x})}{S^k(u \mid a, \mathbf{x}) D^k(u \mid a, \mathbf{x})^2} \mathbb{P}(dy \mid a, \mathbf{x}, k)$$

$$= \int_0^t \frac{S^k(u- \mid a, \mathbf{x}) N_1^k(du \mid a, \mathbf{x})}{S^k(u \mid a, \mathbf{x}) D^k(u \mid a, \mathbf{x})}.$$

Therefore,

$$\mathbb{E}\left[H^k(t \wedge Y, A, \mathbf{X}) - \frac{\mathbb{I}(Y \leq t, \Delta = 1) S^k(Y- \mid A, \mathbf{X})}{S^k(Y \mid A, \mathbf{X}) D^k(Y \mid A, \mathbf{X})} \,\Big|\, A, \mathbf{X}, R = k\right] = 0$$

almost surely. By properties of score functions and the tower property, the above implies that

$$\frac{\partial}{\partial \epsilon} \iint \prod_{(0,t]} \{1 - \Lambda_\epsilon^k(du \mid a, \mathbf{x})\} \mu(d\mathbf{x}) \Big|_{\epsilon=0}$$

$$= \mathbb{E}\left[S^k(t \mid a, \mathbf{X}) \frac{\mathbb{I}(R = k)}{\mathbb{P}(R = k \mid \mathbf{X})} \frac{\mathbb{I}(A = a)}{\pi^k(a \mid \mathbf{X})}\right.$$

$$\left. \times \left\{H^k(t \wedge Y, A, \mathbf{X}) - \frac{\mathbb{I}(Y \leq t, \Delta = 1) S^k(Y- \mid A, \mathbf{X})}{S^k(Y \mid A, \mathbf{X}) D^k(Y \mid A, \mathbf{X})}\right\} \dot{\ell}(\mathcal{O})\right].$$

Combining these results with the facts that $N_1^k(du \mid a, \mathbf{x})/D^k(u \mid a, \mathbf{x}) = \Lambda^k(du \mid a, \mathbf{x})$ and $D^k(u \mid a, \mathbf{x}) = S^k(u- \mid \mathbf{x}) G^k(u \mid a, \mathbf{x})$, we can rewrite (3) at the beginning as follows:

$$\frac{\partial}{\partial \epsilon} \theta^0(t, a) \Big|_{\epsilon=0}$$

$$= \mathbb{E}\left[\frac{\mathbb{I}(R = 0)}{\mathbb{P}(R = 0)} [S^k(t \mid a, \mathbf{X}) - \theta^0(t, a)] \dot{\ell}(\mathcal{O}) - \frac{\mathbb{I}(R = 0)}{\mathbb{P}(R = 0)} \mathbb{E}\left\{S^k(t \mid a, \mathbf{X}) \frac{\mathbb{I}(R = k)}{\mathbb{P}(R = k \mid \mathbf{X})}\right.\right.$$

$$\left.\left. \times \frac{\mathbb{I}(A = a)}{\pi^k(a \mid \mathbf{X})} \left\{\frac{\mathbb{I}(Y \leq t, \Delta = 1)}{S^k(y \mid \mathbf{X}) G^k(y \mid a, \mathbf{X})} - \int_0^{t \wedge y} \frac{\Lambda^k(du \mid a, \mathbf{X})}{S^k(u \mid \mathbf{X}) G^k(u \mid a, \mathbf{X})}\right\} \dot{\ell}(\mathcal{O}) \,\Big|\, \mathbf{X}\right\}\right]$$

$$= \mathbb{E}\left[\frac{\mathbb{I}(R = 0)}{\mathbb{P}(R = 0)} \{S^k(t \mid a, \mathbf{X}) - \theta^0(t, a)\} \dot{\ell}(\mathcal{O})\right] - \mathbb{E}\left[\frac{\mathbb{I}(R = k)}{\mathbb{P}(R = 0)} \frac{\mathbb{P}(R = 0 \mid \mathbf{X})}{\mathbb{P}(R = k \mid \mathbf{X})} S^k(t \mid a, \mathbf{X})\right.$$

$$\left. \times \frac{\mathbb{I}(A = a)}{\pi^k(a \mid \mathbf{X})} \left\{\frac{\mathbb{I}(Y \leq t, \Delta = 1)}{S^k(y \mid \mathbf{X}) G^k(y \mid a, \mathbf{X})} - \int_0^{t \wedge y} \frac{\Lambda^k(du \mid a, \mathbf{X})}{S^k(u \mid \mathbf{X}) G^k(u \mid a, \mathbf{X})}\right\} \dot{\ell}(\mathcal{O})\right].$$

Therefore, an EIF of $\theta^0(t, a)$ at $\mathbb{P}$ is found as

$$\varphi_{t,a}^{*k,0}(\mathcal{O}; \mathbb{P})$$

$$= \frac{\mathbb{I}(R = 0)}{\mathbb{P}(R = 0)} \{S^0(t \mid a, \mathbf{X}) - \theta^0(t, a)\} - \frac{\mathbb{I}(R = k) \mathbb{P}(R = 0 \mid \mathbf{X})}{\mathbb{P}(R = 0) \mathbb{P}(R = k \mid \mathbf{X})} S^k(t \mid a, \mathbf{X})$$

$$\times \frac{\mathbb{I}(A = a)}{\pi^k(a \mid \mathbf{X})} \left[\frac{\mathbb{I}(Y \leq t, \Delta = 1)}{S^k(Y \mid a, \mathbf{X}) G^k(Y \mid a, \mathbf{X})} - \int_0^{t \wedge Y} \frac{\Lambda^k(du \mid a, \mathbf{X})}{S^k(u \mid a, \mathbf{X}) G^k(u \mid a, \mathbf{X})}\right].$$

Observe that, by Bayes's rule,

$$\frac{\mathbb{P}(R = 0 \mid \mathbf{X})}{\mathbb{P}(R = k \mid \mathbf{X})} = \underbrace{\frac{\mathbb{P}(\mathbf{X} \mid R = 0)}{\mathbb{P}(\mathbf{X} \mid R = k)}}_{\omega^{k,0}(\mathbf{X})} \cdot \frac{\mathbb{P}(R = 0)}{\mathbb{P}(R = k)},$$

where $\omega^{k,0}(\mathbf{X})$ is a covariates density ratio function. We then find that the EIF form in Theorem 2.5:

$$\varphi_{t,a}^{*k,0}(\mathcal{O}; \mathbb{P}) = \frac{\mathbb{I}(R = 0)}{\mathbb{P}(R = 0)} \{S^0(t \mid a, \mathbf{X}) - \theta^0(t, a)\} - \frac{\mathbb{I}(R = k)}{\mathbb{P}(R = k)} \omega^{k,0}(\mathbf{X}) S^k(t \mid a, \mathbf{X})$$

$$\times \frac{\mathbb{I}(A = a)}{\pi^k(a \mid \mathbf{X})} \left[\frac{\mathbb{I}(Y \leq t, \Delta = 1)}{S^k(Y \mid a, \mathbf{X}) G^k(Y \mid a, \mathbf{X})} - \int_0^{t \wedge Y} \frac{\Lambda^k(du \mid a, \mathbf{X})}{S^k(u \mid a, \mathbf{X}) G^k(u \mid a, \mathbf{X})}\right].$$

### E.1.2 REGULARITY CONDITIONS FOR THEOREM 2.6

We now state regularity conditions for Theorem 2.6. For site $R = k$, we denote $\pi^k$, $G^k$, $\omega^{k,0}$, $\Lambda^k$ and $S^k$ the truths of nuisance functions. We use $\pi^k_\infty$, $\omega^{k,0}_\infty$, $G^k_\infty$, $\Lambda^k_\infty$ and $S^k_\infty$ to denote some general probability limits for nuisance function estimators.

**Condition E.1.** *There exist $\pi^k_\infty$, $\omega^{k,0}_\infty$, $G^k_\infty$, $\Lambda^k_\infty$ and $S^k_\infty$ such that*

*(a)* $\max_m \mathbb{P} \left[ \dfrac{1}{\widehat{\pi}^k_m(a \mid \mathbf{X})} - \dfrac{1}{\pi^k_\infty(a \mid \mathbf{X})} \right]^2 \to_p 0;$

*(b)* $\max_m \mathbb{P} \left[ \widehat{\omega}^{k,0}_m(\mathbf{X}) - \omega^{k,0}_\infty(\mathbf{X}) \right]^2 \to_p 0;$

*(c)* $\max_m \mathbb{P} \left[ \sup_{u \in [0,t]} \left| \dfrac{1}{\widehat{G}^k_m(u \mid a, \mathbf{X})} - \dfrac{1}{G^k_\infty(u \mid a, \mathbf{X})} \right| \right]^2 \to_p 0;$

*(d)* $\max_m \mathbb{P} \left[ \sup_{u \in [0,t]} \left| \dfrac{\widehat{S}^k_m(t \mid a, \mathbf{X})}{\widehat{S}^k_m(u \mid a, \mathbf{X})} - \dfrac{S^k_\infty(t \mid a, \mathbf{X})}{S^k_\infty(u \mid a, \mathbf{X})} \right| \right]^2 \to_p 0.$

**Condition E.2.** *There exists an $\eta \in (0, \infty)$ such that for $\mathbb{P}$-almost all $\mathbf{x}$, $\widehat{\pi}^k_m(a \mid \mathbf{x}) \geq 1/\eta$, $\pi^k_\infty(a \mid \mathbf{x}) \geq 1/\eta$, $1/\eta \leq \widehat{\omega}^{k,0}_m(\mathbf{x}) \leq \eta$, $1/\eta \leq \omega^{k,0}_\infty(\mathbf{x}) \leq \eta$, $\widehat{G}^k_m(t \mid a, \mathbf{x}) \geq 1/\eta$, and $G^k_\infty(t \mid a, \mathbf{x}) \geq 1/\eta$ with probability tending to 1.*

**Condition E.3.** *Define*

$$r^k_{n,t,a,1} = \max_m \mathbb{P} \left| \{ \widehat{\pi}^k_m(a \mid \mathbf{X}) - \pi^k(a \mid \mathbf{X}) \} \{ \widehat{S}^k_m(t \mid a, \mathbf{X}) - S^k(t \mid a, \mathbf{X}) \} \right|,$$

$$r^k_{n,t,a,2} = \max_m \mathbb{P} \left| \{ \widehat{\omega}^{k,0}_m(\mathbf{X}) - \omega^{k,0}(\mathbf{X}) \} \{ \widehat{S}^k_m(t \mid a, \mathbf{X}) - S^k(t \mid a, \mathbf{X}) \} \right|, \text{ and}$$

$$r^k_{n,t,a,3} = \max_m \mathbb{P} \left| \widehat{S}^k_m(t \mid a, \mathbf{X}) \int_0^t \left\{ \frac{G^k(u \mid a, \mathbf{X})}{\widehat{G}^k_m(u \mid a, \mathbf{X})} - 1 \right\} \left( \frac{S^k}{\widehat{S}^k_m} - 1 \right) (du \mid a, \mathbf{X}) \right|.$$

*Then, it holds that $r^k_{n,t,a,1} = o_p(n^{-1/2})$, $r^k_{n,t,a,2} = o_p(n^{-1/2})$ and $r^k_{n,t,a,3} = o_p(n^{-1/2})$.*

Next, to prove Theorem 2.6, we first introduce some useful results and lemmata in the next section.

### E.1.3 USEFUL LEMMATA FOR THE LOCAL ESTIMATOR

We start by examining the difference $\widehat{\theta}^{k,0}_n(t, a) - \theta^0(t, a)$. Recall $\mathbb{P}^m_n$ is the empirical distribution corresponding to the $m$-th validation set $\mathcal{V}_m$ from the entire data $\mathcal{O}$, and denote $\mathbb{G}^m_n$ the corresponding empirical process. A result exactly following Westling et al. (2024) is that

$$\widehat{\theta}^{k,0}_n(t, a) - \theta^0(t, a) = \mathbb{P}_n[\varphi^{*k,0}_{\infty,t,a}] + \frac{1}{M} \sum_{m=1}^M \frac{M n_m^{1/2}}{n} \mathbb{G}^m_n \left[ \widehat{\varphi}^{k,0}_{n,m,t,a} - \varphi^{k,0}_{\infty,t,a} \right]$$

$$+ \frac{1}{M} \sum_{m=1}^M \frac{M n_m}{n} \mathbb{P} \left[ \widehat{\varphi}^{k,0}_{t,a} - \theta^0(t, a) \right]. \tag{4}$$

We then establish the $L_2(\mathbb{P})$ norm distance (bound) between the estimated EIF and its underlying limit for the local estimator by the following lemma.

**Lemma E.1.** *Under Condition E.2, there exists a universal constant $C = C(\eta)$ such that for each $k$, $m$, $n$, $t$, and $a$,*

$$\mathbb{P}[\widehat{\varphi}^{k,0}_{t,a} - \varphi^{k,0}_{\infty,t,a}]^2 \leq C(\eta) \sum_{j=1}^6 A^k_{j,n,m,t,a},$$

*where*

$$A_{1,n,m,t,a}^k = \mathbb{P}\left[\frac{1}{\mathbb{P}_n^m(R=0)} - \frac{1}{\mathbb{P}(R=0)}\right]^2,$$

$$A_{2,n,m,t,a}^k = \mathbb{P}\left[\frac{1}{\mathbb{P}_n^m(R=k)} - \frac{1}{\mathbb{P}(R=k)}\right]^2,$$

$$A_{3,n,m,t,a}^k = \mathbb{P}\left[\widehat{\omega}_m^{k,0}(a\mid\mathbf{X}) - \omega_\infty^{k,0}(a\mid\mathbf{X})\right]^2,$$

$$A_{4,n,m,t,a}^k = \mathbb{P}\left[\frac{1}{\widehat{\pi}_m^k(a\mid\mathbf{X})} - \frac{1}{\pi_\infty^k(a\mid\mathbf{X})}\right]^2,$$

$$A_{5,n,m,t,a}^k = \mathbb{P}\left[\sup_{u\in[0,t]}\left|\frac{1}{\widehat{G}_m^k(u\mid a,\mathbf{X})} - \frac{1}{G_\infty^k(u\mid a,\mathbf{X})}\right|\right]^2,$$

$$A_{6,n,m,t,a}^k = \mathbb{P}\left[\sup_{u\in[0,t]}\left|\frac{\widehat{S}_m^k(t\mid a,\mathbf{X})}{\widehat{S}_m^k(u\mid a,\mathbf{X})} - \frac{S_\infty^k(t\mid a,\mathbf{X})}{S_\infty^k(u\mid a,\mathbf{X})}\right|\right]^2.$$

*Proof.* We first denote

$$B^k(\mathcal{V}_m) = \frac{\mathbb{I}(A=a)}{\pi^k(a\mid\mathbf{X})}S^k(t\mid a,\mathbf{X})$$

$$\times\left[\frac{\mathbb{I}(Y\le t,\Delta=1)}{S^k(Y\mid a,\mathbf{X})G^k(Y\mid a,\mathbf{X})} - \int_0^{t\wedge Y}\frac{\Lambda^k(du\mid a,\mathbf{X})}{S^k(u\mid a,\mathbf{X})G^k(u\mid a,\mathbf{X})}\right],$$

$$C^k(\mathcal{V}_m) = B^k(\mathcal{V}_m)\omega^{k,0}(\mathbf{X}).$$

Then, we first have the following decomposition:

$$\widehat{\varphi}_{t,a}^{k,0} - \varphi_{\infty,t,a}^{k,0} = \sum_{j=1}^4 U_{j,n,m,t,a}^k,$$

where

$$U_{1,n,m,t,a}^k = \left[\frac{\mathbb{I}(R=0)}{\mathbb{P}_n^m(R=0)} - \frac{\mathbb{I}(R=0)}{\mathbb{P}(R=0)}\right]\widehat{S}_m^0(t\mid a,\mathbf{x}),$$

$$U_{2,n,m,t,a}^k = \frac{\mathbb{I}(R=0)}{\mathbb{P}(R=0)}\left[\widehat{S}_m^0(t\mid a,\mathbf{x}) - S_\infty^0(t\mid a,\mathbf{x})\right],$$

$$U_{3,n,m,t,a}^k = \left[\frac{\mathbb{I}(R=k)}{\mathbb{P}_n^m(R=k)} - \frac{\mathbb{I}(R=k)}{\mathbb{P}(R=k)}\right]\widehat{C}_m^k(\mathcal{V}_m),$$

$$U_{4,n,m,t,a}^k = \frac{\mathbb{I}(R=k)}{\mathbb{P}(R=k)}\left[\widehat{C}_m^k(\mathcal{V}_m) - C_\infty^k(\mathcal{V}_m)\right].$$

Now, for $U_{4,n,m,t,a}^k$, we further decompose it as

$$U_{4,n,m,t,a}^k = \frac{\mathbb{I}(R=k)}{\mathbb{P}(R=k)}\sum_{j=1}^2 V_{j,n,m,t,a}^k,$$

where

$$V_{1,n,m,t,a}^k = B_\infty^k(\mathcal{V}_m)\left[\widehat{\omega}_m^{k,0}(\mathbf{X}) - \omega_m^{k,0}(\mathbf{X})\right],$$

$$V_{2,n,m,t,a}^k = \widehat{\omega}_m^{k,0}(\mathbf{X})\left[\widehat{B}_m^k(\mathcal{V}_m) - B_\infty^k(\mathcal{V}_m)\right].$$

The expression of $\widehat{B}_m^k(\mathcal{V}_m) - B_\infty^k(\mathcal{V}_m)$ is exactly the same as the Lemma 3 in Westling et al. (2024), while we only need to replace the corresponding nuisance functions by the site-$k$ version here, so the detail is omitted. By the triangle inequality, we have $\mathbb{P}[\widehat{\varphi}_{t,a}^{k,0} - \varphi_{\infty,t,a}^{k,0}]^2 \le \left\{\sum_{j=1}^4\{\mathbb{P}[(U_{j,n,m,t,a}^k)^2]\}^{1/2}\right\}^2$. Therefore, under Assumption 2.3 and Condition E.2, there exists a universal constant $C = C(\eta)$ such that the result in the statement holds. $\square$

Furthermore, we need to bound the empirical process term $\mathbb{G}_n^m\left[\widehat{\varphi}_{n,m,t,a}^{k,0} - \varphi_{\infty,t,a_0}^{k,0}\right]$ by $o_p(n^{-1/2})$. This is formally shown below in Lemma E.2.

**Lemma E.2.** *If Conditions E.1–E.2 hold,* $M^{-1}\sum_{m=1}^M n^{-1}Mn_m^{1/2}\mathbb{G}_n^m\left[\widehat{\varphi}_{n,m,t,a}^{k,0} - \varphi_{\infty,t,a_0}^{k,0}\right] = o_p(n^{-1/2}).$

*Proof.* We follow notation in Lemma E.1. First, we note that

$$\frac{Mn_m^{1/2}}{n} \leq \frac{M(|n_m - n/M| + n/M)^{1/2}}{n} \leq \frac{M|n_m - n/M|^{1/2} + M|n/M|^{1/2}}{n} \leq \sqrt{\frac{M}{n}} + \frac{M}{n},$$

for all $m$ since $|n_m - n/M| \leq 1$ by assumption on $n_m$. Then, we have that

$$\frac{1}{M}\sum_{m=1}^M \frac{Mn_m^{1/2}}{n} \sup_{u\in[0,t]} \left|\mathbb{G}_n^m\left[\widehat{\varphi}_{n,m,t,a}^{k,0} - \varphi_{\infty,t,a}^{k,0}\right]\right|$$

$$\leq O(n^{-1/2})\frac{1}{M}\sum_{m=1}^M \sup_{u\in[0,t]} \left|\mathbb{G}_n^m\left[\widehat{\varphi}_{n,m,t,a}^{k,0} - \varphi_{\infty,t,a}^{k,0}\right]\right|,$$

since $K = O(1)$.

Therefore, we turn to show $M^{-1}\sum_{m=1}^M \left|\mathbb{G}_n^m\left[\widehat{\varphi}_{n,m,t,a}^{k,0} - \varphi_{\infty,t,a}^{k,0}\right]\right| = o_p(1)$. Using conditional argument, we write

$$\mathbb{E}\left|\mathbb{G}_n^m\left[\widehat{\varphi}_{n,m,t,a}^{k,0} - \varphi_{\infty,t,a}^{k,0}\right]\right| = \mathbb{E}\left[\mathbb{E}\left|\mathbb{G}_n^m\left[\widehat{\varphi}_{n,m,t,a}^{k,0} - \varphi_{\infty,t,a}^{k,0}\right]\right| \mid \mathcal{T}_m\right],$$

where $\mathcal{T}_m = \mathcal{O}\backslash\mathcal{V}_m$ is the $m$-th training set. Note that the randomness in the inner expectation of the right-hand side above, by conditioning on the training set, is only induced from $\mathbb{G}_n^m$ by averaging over the observations on the validation set. Therefore,

$$\mathbb{E}\left[\mathbb{E}\left|\mathbb{G}_n^m\left[\widehat{\varphi}_{n,m,t,a}^{k,0} - \varphi_{\infty,t,a}^{k,0}\right]\right| \mid \mathcal{T}_m\right] = \mathbb{P}\left|\mathbb{G}_n^m(\widehat{\varphi}_{n,m,t,a}^{k,0} - \varphi_{\infty,t,a}^{k,0})\right|.$$

Defining $\mathcal{F}_{n,m,t,a}^{k,0}$ as the singleton class of functions $\widehat{\varphi}_{n,m,t,a}^{k,0} - \varphi_{\infty,t,a}^{k,0}$, we further have

$$\mathbb{P}\left|\mathbb{G}_n^m(\widehat{\varphi}_{n,m,t,a}^{k,0} - \varphi_{\infty,t,a}^{k,0})\right| = \mathbb{P}\left[\sup_{f\in\mathcal{F}_{n,m,t,a}^{k,0}} |\mathbb{G}_n^m(f)|\right].$$

By Theorem 2.1.14 in Van der Vaart & Wellner (1996), the covering number of $\mathcal{F}_{n,m,t,a}^{k,0}$ is 1 for all $\varepsilon$, so the uniform entropy integral $J(1, \mathcal{F}_{n,m,t,a}^{k,0})$ is 1 relative to the natural envelope $|\widehat{\varphi}_{n,m,t,a}^{k,0} - \varphi_{\infty,t,a}^{k,0}|$. Therefore, there is a universal constant $C'$ such that

$$\mathbb{P}\left[\sup_{f\in\mathcal{F}_{n,m,t,a}^{k,0}} |\mathbb{G}_n^m(f)|\right] \leq C'\left\{\mathbb{P}(\widehat{\varphi}_{n,m,t,a}^{k,0} - \varphi_{\infty,t,a}^{k,0})^2\right\}^{1/2} \leq C''\sum_{j=1}^6 \bar{A}_{j,n,m,t,a},$$

following definition of $\bar{A}_{j,n,m,t,a}$ terms in Lemma E.1, so that $M^{-1}\sum_{m=1}^M \mathbb{E}\left|\mathbb{G}_n^m\left[\widehat{\varphi}_{n,m,t,a}^{k,0} - \varphi_{\infty,t,a}^{k,0}\right]\right|$ is bounded up to $C'''\sum_{j=1}^6 \mathbb{E}[\max_m(\bar{A}_{j,n,m,t,a})]$ for some constant $C'''$. It is straightforward that by Conditions E.1 and E.2, this upper bound tends to zero, so $M^{-1}\sum_{m=1}^M \left|\mathbb{G}_n^m\left[\widehat{\varphi}_{n,m,t,a}^{k,0} - \varphi_{\infty,t,a}^{k,0}\right]\right| = o_p(1)$. $\square$

Finally, the only difference we have not characterized in (4) is $\mathbb{P}[\varphi_{t,a}^{k,0}(\mathcal{O}; \mathbb{P}_\infty)] - \theta^0(t, a)$, which we show it below.

**Lemma E.3.** *Consider some general nuisance functions under* $\mathbb{P}_\infty$, *denoted by* $S_\infty^0, S_\infty^k, G_\infty^k, \pi_\infty^k$, *and* $\omega_\infty^{k,0}$ *(equals 1 if $k = 0$). Then,* $\mathbb{P}[\varphi_{t,a}^{k,0}(\mathcal{O}; \mathbb{P}_\infty)] - \theta^0(t, a)$ *equals*

$$\mathbb{E}\left[\frac{q^0(\mathbf{X})}{\mathbb{P}(R=0)}S_\infty^k(t \mid a, \mathbf{X})\int_0^t \frac{S^k(y- \mid a, \mathbf{X})}{S_\infty^k(y \mid a, \mathbf{X})}\right.$$

$$\left.\times \left\{\frac{\omega_\infty^{k,0}(\mathbf{X})G^k(y \mid a, \mathbf{X})\pi^k(a \mid \mathbf{X})}{\omega^{k,0}(\mathbf{X})G_\infty^k(y \mid a, \mathbf{X})\pi_\infty^k(a \mid \mathbf{X})} - 1\right\}(\Lambda_\infty^k - \Lambda^k)(dy \mid a, \mathbf{X})\right].$$

*Proof.* By direct calculations, $\mathbb{P}[\varphi_{t,a}^{k,0}(\mathcal{O};\mathbb{P}_\infty)] - \theta^0(t,a)$ equals

$$
\mathbb{E}\left[\frac{\mathbb{I}(R=0)}{\mathbb{P}(R=0)}\{S_\infty^0(t\mid a,\mathbf{X}) - S^0(t\mid a,\mathbf{X})\} + \frac{q^k(\mathbf{X})}{\mathbb{P}(R=k)}\omega_\infty^{k,0}(\mathbf{X})S_\infty^k(t\mid a,\mathbf{X})\frac{\pi^k(a\mid\mathbf{X})}{\pi_\infty^k(a\mid\mathbf{X})}\right.
$$
$$
\left.\times\int_0^t\frac{S^k(y-\mid a,\mathbf{X})G^k(y\mid a,\mathbf{X})}{S_\infty^k(y\mid a,\mathbf{X})G_\infty^k(y\mid a,\mathbf{X})}(\Lambda_\infty^k - \Lambda^k)(dy\mid a,\mathbf{X})\right]
$$
$$
=\mathbb{E}\left[\frac{q^0(\mathbf{X})}{\mathbb{P}(R=0)}\{S_\infty^0(t\mid a,\mathbf{X}) - S^0(t\mid a,\mathbf{X})\} + \frac{q^0(\mathbf{X})}{\mathbb{P}(R=0)}\frac{\omega_\infty^{k,0}(\mathbf{X})}{\omega^{k,0}(\mathbf{X})}S_\infty^k(t\mid a,\mathbf{X})\frac{\pi^k(a\mid\mathbf{X})}{\pi_\infty^k(a\mid\mathbf{X})}\right.
$$
$$
\left.\times\int_0^t\frac{S^k(y-\mid a,\mathbf{X})G^k(y\mid a,\mathbf{X})}{S_\infty^k(y\mid a,\mathbf{X})G_\infty^k(y\mid a,\mathbf{X})}(\Lambda_\infty^k - \Lambda^k)(dy\mid a,\mathbf{X})\right].
$$

In the second "$\mathbb{E}$" after "=", we used the following relationship:

$$
\frac{q^0(\mathbf{X})}{q^k(\mathbf{X})} = \omega^{k,0}(\mathbf{X})\frac{\mathbb{P}(R=0)}{\mathbb{P}(R=k)}
$$

by Bayes's rule. Furthermore, by Duhamel equation in Gill & Johansen (1990),

$$
\mathbb{P}[\varphi_{t,a}^{k,0}(\mathcal{O};\mathbb{P}_\infty)] - \theta^0(t,a)
$$
$$
=\mathbb{E}\left[\frac{q^0(\mathbf{X})}{\mathbb{P}(R=0)}S_\infty^k(t\mid a,\mathbf{X})\int_0^t\frac{S^k(y-\mid a,\mathbf{X})}{S_\infty^k(y\mid a,\mathbf{X})}\right.
$$
$$
\left.\times\left\{\frac{\omega_\infty^{k,0}(\mathbf{X})G^k(y\mid a,\mathbf{X})\pi^k(a\mid\mathbf{X})}{\omega^{k,0}(\mathbf{X})G_\infty^k(y\mid a,\mathbf{X})\pi_\infty^k(a\mid\mathbf{X})} - 1\right\}(\Lambda_\infty^k - \Lambda^k)(dy\mid a,\mathbf{X})\right]. \tag{5}
$$

$\square$

### E.1.4 PROOF OF THEOREM 2.6

By (4) with $\pi_\infty^k = \pi^k, \omega_\infty^{k,0} = \omega^{k,0}, G_\infty^k = G^k$, and $S_\infty^k = S^k$,

$$
\widehat{\theta}_n^{k,0}(t,a) - \theta^0(t,a) = \mathbb{P}_n[\varphi_{t,a}^{*k,0}] + \frac{1}{M}\sum_{m=1}^M\frac{Mn_m^{1/2}}{n}\mathbb{G}_n^m\left[\widehat{\varphi}_{n,m,t,a}^{k,0} - \varphi_{t,a}^{k,0}\right]
$$
$$
+ \frac{1}{M}\sum_{m=1}^M\frac{Mn_m}{n}\mathbb{P}\left[\widehat{\varphi}_{t,a}^{k,0} - \theta^0(t,a)\right].
$$

By Conditions E.1 and E.2, the second summand on the right-hand-side is $o_p(n^{-1/2})$ by Lemma E.2. By Lemma E.3, $\mathbb{P}[\widehat{\varphi}_{t,a}^{k,0}] - \theta^0(t,a)$ equals

$$
\mathbb{E}\left[\frac{q^0(\mathbf{X})}{\mathbb{P}(R=0)}\widehat{S}_m^k(t\mid a,\mathbf{X})\int_0^t\frac{S^k(y-\mid a,\mathbf{X})}{\widehat{S}_m^k(y\mid a,\mathbf{X})}\right.
$$
$$
\left.\times\left\{\frac{\widehat{\omega}_m^{k,0}(\mathbf{X})G^k(y\mid a,\mathbf{X})\pi^k(a\mid\mathbf{X})}{\omega^{k,0}(\mathbf{X})\widehat{G}_m^k(y\mid a,\mathbf{X})\widehat{\pi}_m^k(a\mid\mathbf{X})} - 1\right\}(\widehat{\Lambda}_m^k - \Lambda^k)(dy\mid a,\mathbf{X})\right].
$$

By Duhamel equation in Gill & Johansen (1990) and Condition E.3, we find that the above bias term can be bounded by $\eta^2\{r_{n,t,a,1}^k + r_{n,t,a,2}^k + r_{n,t,a,3}^k\}$ over $m$. Since $M^{-1}\sum_{m=1}^M n^{-1}Mn_m \le 2$, we have

$$
\left|\frac{1}{M}\sum_{m=1}^M\frac{Mn_m}{n}\mathbb{P}\left[\widehat{\varphi}_{t,a}^{k,0} - \theta^0(t,a)\right]\right| \le 2\eta^2\left\{r_{n,t,a,1}^k + r_{n,t,a,2}^k + r_{n,t,a,3}^k\right\} = o_p(n^{-1/2}),
$$

by Condition E.3. This established the pointwise RAL property: $\widehat{\theta}_n^{k,0}(t,a) = \theta^0(t,a) + \mathbb{P}_n(\varphi_{t,a}^{*k,0}) + o_p(n^{-1/2})$. Since $\varphi_{t,a}^{*k,0}$ is uniformly bounded, $\mathbb{P}\{(\varphi_{t,a}^{*k,0})^2\} < \infty$ and since $\mathbb{P}\{\varphi_{t,a}^{*k,0}\} = 0$, then

$$
n^{1/2}\mathbb{P}_n(\widehat{\varphi}_{t,a}^{*k,0}) \to_d \mathcal{N}(0,\mathbb{P}\{(\varphi_{t,a}^{*k,0})^2\}).
$$

**Remark E.4** (Technical version of Theorem 2.8). If we only need the consistency of $\widehat{\theta}_n^k(t, a)$, then condition $\pi_\infty^k = \pi^k$, $\omega_\infty^{k,0} = \omega^{k,0}$, $G_\infty^k = G^k$, and $S_\infty^k = S^k$ can be replaced by the following statement: For $\mathbb{P}$-almost all $\mathbf{X}$, there exist measurable sets $\mathcal{S}_x^k, \mathcal{G}_x^k \subseteq [0, t]$ such that $\mathcal{S}_x^k \cup \mathcal{G}_x^k = [0, t]$ and $\Lambda^k(u \mid a, \mathbf{X}) = \Lambda_\infty^k(u \mid a, \mathbf{X})$ for all $u \in \mathcal{S}_x^k$ and $G(u \mid a, \mathbf{X}) = G_\infty^k(u \mid a, \mathbf{X})$ for all $u \in \mathcal{G}_x^k$. In addition, if $\mathcal{S}_x^k$ is a strict subset of $[0, t]$, then $\pi^k(a \mid \mathbf{X}) = \pi_\infty^k(a \mid \mathbf{X})$ and $\omega^{k,0}(\mathbf{X}) = \omega_\infty^{k,0}(\mathbf{X})$ as well. Then, $\widehat{\theta}_n^k(t, a)$ is consistent if Conditions E.1 and E.2 hold.

To prove Remark E.4, we decompose the integral $\int_0^t$ as $\int_{\mathcal{S}_x^k} + \int_{\mathcal{S}_x^{k,c}}$, where $\mathcal{S}_x^{k,c}$ is the complement of set $\mathcal{S}_x^k$, and $\mathcal{S}_x^{k,c} \subseteq \mathcal{G}_x^k$ by definition. Then, it is straightforward to verify that when the statement in Remark E.4 holds, the following integral

$$\int_0^t \frac{S^k(y- \mid a, \mathbf{X})}{S_\infty^k(y \mid a, \mathbf{X})} \left\{ \frac{G^k(y \mid a, \mathbf{X})\pi^k(a \mid \mathbf{X})}{G_\infty(y \mid a, \mathbf{X})\pi_\infty^k(a \mid \mathbf{X})} - 1 \right\} (\Lambda_\infty^k - \Lambda^k)(dy \mid a, \mathbf{X})$$

$$= \left( \int_{\mathcal{S}_x^k} + \int_{\mathcal{G}_x^k} \right) \frac{S^k(y- \mid a, \mathbf{X})}{S_\infty^k(y \mid a, \mathbf{X})} \left\{ \frac{G^k(y \mid a, \mathbf{X})\pi^k(a \mid \mathbf{X})}{G_\infty^k(y \mid a, \mathbf{X})\pi_\infty^k(a \mid \mathbf{X})} - 1 \right\} (\Lambda_\infty^k - \Lambda^k)(dy \mid a, \mathbf{X}) = 0,$$

which further implies $\mathbb{P}[\varphi_{\infty,t,a}^{*k,0}] = 0$.

### E.2 THEORY FOR THE FEDERATED ESTIMATOR

In this section, we present the properties of the federated estimator. Given that our proposed weights, $\boldsymbol{\eta}_{t,a}$, are both time- and treatment-specific, we focus on the pointwise convergence properties.

Let the set of all source site indices be $\mathcal{S} = \{1, \ldots, K-1\}$. We then define the oracle selection space for $\boldsymbol{\eta}_{t,a}$, and the corresponding weight space as:

$$\mathcal{S}_{t,a}^* = \{k \in \mathcal{S} : \theta^k(t, a) = \theta^0(t, a)\}, \text{ and } \mathbb{R}^{\mathcal{S}_{t,a}^*} = \{\boldsymbol{\eta}_{t,a} \in \mathbb{R}^{K-1} : \eta_{t,a}^j = 0, \forall j \notin \mathcal{S}_{t,a}^*\},$$

respectively.

The space $\mathcal{S}_{t,a}^*$ is both time- and treatment-varying, indicating that a source site may not consistently be useful or unhelpful across different time points or treatments. However, it offers the advantage of increased flexibility and adaptivity, allowing for more effective borrowing of information at different points along the survival functions. Based on the theory presented in Section E.1, for $k \in \mathcal{S}_{t,a}^*$, the site-specific estimator $\widehat{\theta}_n^{k,0}(t, a)$ is consistent for $\theta^0(t, a)$ for any given $t \in [0, \tau]$ and $a \in \{0, 1\}$.

We begin by assuming fixed $\boldsymbol{\eta}_{t,a} = (\eta_{t,a}^0, \eta_{t,a}^1, \ldots, \eta_{t,a}^{K-1})$. We invoke Lemmata 4 and 5 in Han et al. (2025), which state that the proposed adaptive estimation for $\eta_{t,a}^k$ as shown in (2) allows for (i) the recovery of the optimal $\eta_{t,a}^k$ by the estimator $\widehat{\eta}_{t,a}^k$, and (ii) the uncertainty induced by $\widehat{\eta}_{t,a}^k$ is negligible when estimating $\theta^0(t, a)$. We require regularity Conditions E.1, E.2 and E.3 for the pointwise convergence result in Theorem 2.6 hold. Let us denote the federated estimator by plugging-in the fixed $\boldsymbol{\eta}_{t,a}$ as

$$\widehat{\theta}_n^{\text{fed}}(t, a; \boldsymbol{\eta}_{t,a}) = \left( 1 - \sum_{k \in \mathcal{S}} \eta_{t,a}^k \right) \widehat{\theta}_n^0(t, a) + \sum_{k \in \mathcal{S}} \eta_{t,a}^k \widehat{\theta}_n^{k,0}(t, a).$$

Recall that notation $\mathcal{H}_{t,a}$ defined in (1):

$$\mathcal{H}_{t,a}(\mathcal{O}; S, G) = \frac{\mathbb{I}(Y \leq t, \delta = 1)}{S(Y \mid a, \mathbf{X})G(Y \mid a, \mathbf{X})} - \int_0^{t \wedge Y} \frac{\Lambda(du \mid a, \mathbf{X})}{S(u \mid a, \mathbf{X})G(u \mid a, \mathbf{X})}.$$

Let us then write

$$\xi^{0,(1)}(\mathcal{O}) = S^0(t \mid a, \mathbf{X}) \frac{\mathbb{I}(A = a)}{\pi^0(a \mid \mathbf{X})} \mathcal{H}_{t,a}(\mathcal{O}; S^0, \Lambda^0, G^0),$$

$$\xi^{k,0,(1)}(\mathcal{O}) = \omega^{k,0}(\mathbf{X})S^k(t \mid a, \mathbf{X}) \frac{\mathbb{I}(A = a)}{\pi^k(a \mid \mathbf{X})} \mathcal{H}_{t,a}(\mathcal{O}; S^k, \Lambda^k, G^k),$$

$$\xi^{0,(2)}(\mathcal{O}) = S^0(t \mid a, \mathbf{X}) - \theta^0(t, a),$$

and $n_k = \sum_{i=1}^{n} \mathbb{I}(R_i = k)$ for $k = 0, 1, \ldots, K - 1$.

Then,

$$
\begin{aligned}
&\widehat{\theta}_n^{\text{fed}}(t, a; \boldsymbol{\eta}_{t,a}) - \theta^0(t, a) \\
&= \left(1 - \sum_{k \in \mathcal{S}} \eta_{t,a}^k\right) \left\{\widehat{\theta}_n^0(t, a) - \theta^0(t, a)\right\} + \sum_{k \in \mathcal{S}} \eta_{t,a}^k \left\{\widehat{\theta}_n^{k,0}(t, a) - \theta^0(t, a)\right\} \\
&= \left(1 - \sum_{k \in \mathcal{S}} \eta_{t,a}^k\right) \frac{1}{n_0} \sum_{i=1}^{n} \mathbb{I}(R_i = 0) \left\{\widehat{\xi}^{0,(2)}(\mathcal{O}_i) - \widehat{\xi}^{0,(1)}(\mathcal{O}_i)\right\} \\
&\quad + \sum_{k \in \mathcal{S}} \frac{1}{n_0} \sum_{i=1}^{n} \mathbb{I}(R_i = 0) \eta_{t,a}^k \widehat{\xi}^{0,(2)}(\mathcal{O}_i) - \sum_{k \in \mathcal{S}} \frac{1}{n_k} \sum_{i=1}^{n} \mathbb{I}(R_i = k) \eta_{t,a}^k \widehat{\xi}^{k,0,(1)}(\mathcal{O}_i) \\
&= \frac{1}{n} \sum_{i=1}^{n} \left(1 - \sum_{k \in \mathcal{S}} \eta_{t,a}^k\right) \mathbb{I}(R_i = 0) \frac{\widehat{\xi}^{0,(2)}(\mathcal{O}_i) - \widehat{\xi}^{0,(1)}(\mathcal{O}_i)}{\widehat{\mathbb{P}}(R_i = 0)} \\
&\quad + \frac{1}{n} \sum_{i=1}^{n} \mathbb{I}(R_i = 0) \left(\sum_{k \in \mathcal{S}} \eta_{t,a}^k\right) \frac{\widehat{\xi}^{0,(2)}(\mathcal{O}_i)}{\widehat{\mathbb{P}}(R_i = 0)} - \frac{1}{n} \sum_{k \in \mathcal{S}} \sum_{i=1}^{n} \mathbb{I}(R_i = k) \eta_{t,a}^k \frac{\widehat{\xi}^{k,0,(1)}(\mathcal{O}_i)}{\widehat{\mathbb{P}}(R_i = k)}. \quad (6)
\end{aligned}
$$

The asymptotic variance of $\widehat{\theta}_n^{\text{fed}}(t, a; \boldsymbol{\eta}_{t,a})$ equals the variance of the influence function of (6). Let us denote it as $\mathcal{V}_{t,a}^{\text{fed}} = \mathcal{V}_{t,a}^{\text{fed}}(\boldsymbol{\eta}_{t,a})$. We highlight its dependence to the federated weights vector $\boldsymbol{\eta}_{t,a}$ here because in the below (8), we consider an optimization program for deriving the weights based on minimizing the (estimated) asymptotic variance.

Under the assumption of i.i.d. participants within each site, we have

$$
\begin{aligned}
\mathcal{V}_{t,a}^{\text{fed}} &= \left(1 - \sum_{k \in \mathcal{S}} \eta_{t,a}^k\right)^2 \frac{\mathbb{V}\{\xi^{0,(2)}(\mathcal{O}_i) - \xi^{0,(1)}(\mathcal{O}_i) \mid R_i = 0\}}{\mathbb{P}(R_i = 0)} \\
&\quad + \left(\sum_{k \in \mathcal{S}} \eta_{t,a}^k\right)^2 \frac{\mathbb{V}\{\xi^{0,(2)}(\mathcal{O}_i) \mid R_i = 0\}}{\mathbb{P}(R_i = 0)} \\
&\quad + 2 \left(1 - \sum_{k \in \mathcal{S}} \eta_{t,a}^k\right) \left(\sum_{k \in \mathcal{S}} \eta_{t,a}^k\right) \frac{\text{Cov}\{\xi^{0,(2)}(\mathcal{O}_i) - \xi^{0,(1)}(\mathcal{O}_i), \xi^{0,(2)}(\mathcal{O}_i) \mid R_i = 0\}}{\mathbb{P}(R_i = 0)} \\
&\quad + \sum_{k \in \mathcal{S}} (\eta_{t,a}^k)^2 \frac{\mathbb{V}\{\xi^{k,0,(1)}(\mathcal{O}_i) \mid R_i = k\}}{\mathbb{P}(R_i = k)}. \quad (7)
\end{aligned}
$$

With appropriate boundedness conditions on conditional variance and covariance terms above, $\mathcal{V}_{t,a}^{\text{fed}} < \infty$ (see Lemma E.6). Consequently, the asymptotic distribution of $\widehat{\theta}_n^{\text{fed}}(t, a; \boldsymbol{\eta}_{t,a})$ is given by

$$
\sqrt{n} \left\{\widehat{\theta}_n^{\text{fed}}(t, a; \boldsymbol{\eta}_{t,a}) - \theta^0(t, a)\right\} \to_d \mathcal{N}(0, \mathcal{V}_{t,a}^{\text{fed}}).
$$

**Remark E.5.** Based on the derivations in (6) and (7), an influence-function–based asymptotic variance estimator of $\widehat{\theta}_n^{\text{fed}}(t, a)$ ($\widehat{\mathcal{V}}_{t,a}^{\text{fed}}$ in Theorem 2.10), is obtained by replacing the population proportions, variances, and covariances in (7) with their sample (empirical) counterparts and plugging in the estimated weight vector $\widehat{\boldsymbol{\eta}}_{t,a}$.

We further define the optimal adaptive weights $\bar{\boldsymbol{\eta}}_{t,a}$ as follows:

$$
\bar{\boldsymbol{\eta}}_{t,a} = \underset{\eta_{t,a}^k = 0, \forall k \notin \mathcal{S}_{t,a}^*}{\arg \min} \mathcal{V}_{t,a}^{\text{fed}}(\boldsymbol{\eta}_{t,a}). \quad (8)
$$

We adapt two lemmata from Han et al. (2025) for recovering the optimal weights $\bar{\boldsymbol{\eta}}_{t,a}$ with negligible uncertainty for estimating $\theta^0(t, a)$ if we estimate $\boldsymbol{\eta}_{t,a}$ using (2), akin to adaptive Lasso (Zou, 2006; Fan et al., 2024).

**Lemma E.6** (adapted from Lemma 4 in Han et al. (2025)). *Under Conditions E.1—E.3, along with the following mild conditions on covariates support and covariances: (i) The covariates $\mathbf{X}$ and density ratio $\omega^{k,0}(\mathbf{X})$ are in compact sets $\mathbf{X} \in [-B, B]^p$ and $\omega^{k,0}(\mathbf{X}) \in [-B, B]$ for all $k = 1, \dots, K - 1$ with probability 1; and (ii) The variance of $\xi^{k,0,(1)}(\mathcal{O}) \in [\varepsilon, M]$, and the variance-covariance matrix $\mathcal{V}\big[(\xi^{0,(1)}, \xi^{0,(2)})' \mid R = 0\big]$ has eigenvalues in $[\varepsilon, B]$ for some positive constants $\varepsilon$ and $B$. Then, it holds that*

$$\lim_{n \to \infty} \mathbb{P}(\widehat{\boldsymbol{\eta}}_{t,a} \in \mathbb{R}^{S^*_{t,a}}) = 1, \quad \|\widehat{\boldsymbol{\eta}}_{t,a} - \bar{\boldsymbol{\eta}}_{t,a}\| = O_p(n^{-1/2}),$$

*for all $(t, a) \in [0, \tau] \times \{0, 1\}$.*

**Lemma E.7** (adapted from Lemma 5 in Han et al. (2025)). *Under conditions in Lemma E.6,*

$$\sqrt{n}\left( \widehat{\theta}^{fed}(t, a; \widehat{\boldsymbol{\eta}}_{t,a}) - \theta^0(t, a) \right) \to_d \mathcal{N}\big(0, \mathcal{V}^{fed}_{t,a}(\bar{\boldsymbol{\eta}}_{t,a})\big),$$

*for all $(t, a) \in [0, \tau] \times \{0, 1\}$.*

The consistency of $\widehat{\mathcal{V}}^{\text{fed}}_{t,a} = \widehat{\mathcal{V}}^{\text{fed}}_{t,a}(\widehat{\boldsymbol{\eta}}_{t,a})$ follows when we can effectively approximate $\mathcal{V}^{\text{fed}}_{t,a}(\bar{\boldsymbol{\eta}}_{t,a})$ with $\widehat{\mathcal{V}}^{\text{fed}}_{t,a}$. Thus,

$$\sqrt{n/\widehat{\mathcal{V}}^{\text{fed}}_{t,a}}\left\{ \widehat{\theta}^{\text{fed}}_n(t, a) - \theta^0(t, a) \right\} \to_d \mathcal{N}(0, 1).$$

We now analyze the efficiency gain resulting from the federation process. The estimator relies only on the target data is denoted as $\widehat{\theta}^0_n(t, a) = \widehat{\theta}^{\text{fed}}_n(t, a; \boldsymbol{\eta}^0_{t,a})$, where $\boldsymbol{\eta}^0_{t,a}$ assigns all weights to the target and none to the source. In contrast, the estimator that leverages the proposed adaptive ensemble approach is denoted as $\widehat{\theta}^{\text{fed}}_n(t, a; \widehat{\boldsymbol{\eta}}_{t,a})$. Here $\widehat{\boldsymbol{\eta}}_{t,a}$ can recover the optimal weights $\bar{\boldsymbol{\eta}}_{t,a}$ that are associated with the minimum asymptotic variance. Consequently, the variance of $\widehat{\theta}^{\text{fed}}_n(t, a; \widehat{\boldsymbol{\eta}}_{t,a})$ is no larger than that of the estimator relying solely on the target data since $\boldsymbol{\eta}^0_{t,a}$ is generally not the variance minimizer.

To establish that the asymptotic variance of $\widehat{\theta}^{\text{fed}}_n(t, a; \widehat{\boldsymbol{\eta}}_{t,a})$ is strictly smaller than that of the estimator based solely on the target data $\widehat{\theta}^0_n(t, a)$, we adopt Proposition 1 in Han et al. (2025) with a modified informative source condition (modified Assumption 3(b) in Han et al. (2025)).

Specifically, for each source site $s \in \mathcal{S}^*_{t,a}$, we define $\widehat{\theta}^{\text{fed}}_n(t, a; \eta^s_{t,a})$ a federated estimator where $\eta^s_{t,a}$ is the optimal ensemble weight of site $s$ if we only consider target site and this source site $s$ for the federation. Then, the modified informative source condition is given as

$$\left| \text{Cov}\left[ \sqrt{n}\widehat{\theta}^0_n(t, a), \sqrt{n}\left\{ \widehat{\theta}^{\text{fed}}_n(t, a; \eta^s_{t,a}) - \widehat{\theta}^0_n(t, a) \right\} \right] \right| \geq \varepsilon,$$

for some $\varepsilon > 0$, where $\widehat{\theta}^{\text{fed}}_n(t, a; \eta^s_{t,a}) - \widehat{\theta}^0_n(t, a)$ can be expressed as

$$
\begin{aligned}
&\widehat{\theta}^{\text{fed}}_n(t, a; \eta^s_{t,a}) - \widehat{\theta}^0_n(t, a) \\
&= \left\{ \widehat{\theta}^{\text{fed}}_n(t, a; \eta^s_{t,a}) - \theta^0(t, a) \right\} - \left\{ \widehat{\theta}^0_n(t, a) - \theta^0(t, a) \right\} \\
&= \frac{1}{n} \sum_{i=1}^n \mathbb{I}(R_i = 0)(1 - \eta^s_{t,a}) \frac{\widehat{\xi}^{0,(2)}(\mathcal{O}_i) - \widehat{\xi}^{0,(1)}(\mathcal{O}_i)}{\widehat{\mathbb{P}}(R_i = 0)} + \frac{1}{n} \sum_{i=1}^n \mathbb{I}(R_i = 0)\eta^s_{t,a} \frac{\widehat{\xi}^{0,(2)}(\mathcal{O}_i)}{\widehat{\mathbb{P}}(R_i = 0)} \\
&\quad - \frac{1}{n} \sum_{i=1}^n \mathbb{I}(R_i = s)\eta^s_{t,a} \frac{\widehat{\xi}^{s,0,(1)}(\mathcal{O}_i)}{\widehat{\mathbb{P}}(R_i = s)} - \frac{1}{n} \sum_{i=1}^n \mathbb{I}(R_i = 0) \frac{\widehat{\xi}^{0,(2)}(\mathcal{O}_i) - \widehat{\xi}^{0,(1)}(\mathcal{O}_i)}{\widehat{\mathbb{P}}(R_i = 0)} \\
&= \frac{1}{n} \sum_{i=1}^n \mathbb{I}(R_i = 0)\eta^s_{t,a} \frac{\widehat{\xi}^{0,(1)}(\mathcal{O}_i)}{\widehat{\mathbb{P}}(R_i = 0)} - \frac{1}{n} \sum_{i=1}^n \mathbb{I}(R_i = s)\eta^s_{t,a} \frac{\widehat{\xi}^{s,0,(1)}(\mathcal{O}_i)}{\widehat{\mathbb{P}}(R_i = s)}.
\end{aligned}
$$

Therefore, it is straightforward to see that the modified condition can be achieved if $\eta^s_{t,a} > 0$.

