# OpenReview forum: "Privacy-Protected Causal Survival Analysis Under Distribution Shift"
_ICLR.cc/2026/Conference — ICLR 2026 Poster_

### Official Review · Reviewer_n3Hz · 2025-10-24

**Soundness:** 3
**Presentation:** 3
**Contribution:** 2
**Rating:** 4
**Confidence:** 4

**Summary:**

This paper aims at introducing a new procedure to perform generalization of treatment-specific survival functions to a centralized target population, where the effect is learned of multiple decentralized sources, distinct from the target, and without exchanging raw individual data as in a federated constraint. The authors tackle the challenging setting where the source and target are heterogeneous on their joint distributions. The method proposed is 1. each site fits local nuisance models and EIF for pseudo outcomes at each treatment arm and time point reweighted by estimated density ratios to account for covariate shift relative to the target, 2. the EIFs and other summary statistics are sent to the server and aggregated with weights $\eta^k$ which penalize sites according to their estimated discrepancy from the target's nuisance functions.

**Strengths:**

The work presented here is a nice foundation for survival curve estimation in a federated setting, which is crucial and where privacy is clearly a big stake.

The theory seems solid with asymptotic unbiasedness and accounts for heterogeneous designs. I believe most of the strength of the paper is on theorem E.1, which should be the central theorem of the (main) paper instead of the appendix. Assessing how original this theorem is in comparison to [1] is not properly addressed though.

[1] Westling, Ted, et al. "Inference for treatment-specific survival curves using machine learning." Journal of the American Statistical Association 119.546 (2024): 1541-1553.

**Weaknesses:**

## Main Concerns

- **Related work and positioning:**
  The discussion of related work is incomplete. Several recent contributions have explored federated causal inference, such as [1] and [2], which could serve as meaningful baselines even if they are not specific to survival outcomes, or at least be referred to. In addition, key competitors have been omitted, including [2] and [3], who propose federated external control arm approaches that relax the positivity assumption across sites. The paper should more clearly situate itself within this literature and justify how it advances beyond these frameworks.

- **Conceptual novelty:**
  The proposed method appears to be primarily a survival adaptation of FACE [4]. The paper would benefit from a clearer statement of what new theoretical or methodological insights are introduced beyond this earlier work. Without this clarification, the contribution could read as incremental.

- **Motivation and scope:**
  The motivation for privacy preservation in survival analysis should be better articulated. The authors should explain why privacy is especially relevant for survival data and why access to large multi-institutional datasets is critical for estimating treatment-specific survival functions. Currently, the privacy aspect appears overstated relative to its actual methodological treatment.

- **Estimator behavior and information sharing:**
  The estimator defined in Theorem 2.5 does not seem to borrow information from other source sites, effectively acting as a generalization of local survival estimators to the target population. The moment-matching aggregation step does not clearly improve efficiency. This raises questions about how much “federation” is actually achieved beyond one-shot aggregation of summaries.

- **Density-ratio estimation and implementation details:**
  The section on density-ratio estimation is too brief (line 194) and should be expanded. Estimation under federated constraints is non-trivial, and the so-called “flexible” exponential tilt models are better described as parametric. The authors should discuss the impact of misspecification and the propagation of density-ratio estimation error to the final survival estimates. Moreover, the requirement to share empirical covariance matrices across sites raises both dimensionality and privacy issues that require explicit discussion.

---

## Minor Comments

- The section titled “Limitations of existing work” should be reformulated into a proper **Related Work** section, emphasizing connections to federated causal inference and data fusion.
- Clarify in the text which quantities are exchanged across sites (EIFs, covariance matrices, model parameters). This will help readers understand whether the method is truly federated or semi-centralized.
- Some claims about “flexible” models or “privacy-protected” estimation should be moderated or substantiated.
- Improve notation consistency and specify whether expectations and variances in key equations are empirical or population-level.
- The role of overlap (Assumption 2.3) should be highlighted in the discussion of limitations, as it may fail in realistic multi-center observational data.

---

## Recommendations

1. **Reorganize and expand the related work section** to include recent developments in federated causal inference, external control arm methods, and meta-analytic baselines.
2. **Clarify the contribution relative to FACE**, identifying specific theoretical or computational advances introduced in this survival extension.
3. **Provide additional methodological details** on the estimation of density ratios, including how model misspecification on exponential tilt model and limited covariate overlap affect performance.
4. **Discuss communication and privacy aspects** more explicitly, particularly the trade-offs between sharing covariance matrices and preserving data confidentiality.
5. **Include stronger baselines** (e.g., meta-analysis and FedECA) in empirical evaluations to demonstrate under which regimes the proposed method achieves measurable improvements.

---

### References

[1] Xiong, Ruoxuan, et al. “Federated causal inference in heterogeneous observational data.” *Statistics in Medicine* 42.24 (2023): 4418–4439.

[2] Archetti, Alberto, et al. “Heterogeneous datasets for federated survival analysis simulation.” *Companion of the 2023 ACM/SPEC International Conference on Performance Engineering.* 2023.

[3] Terrail, Jean Ogier du, et al. “FedECA: A federated external control arm method for causal inference with time-to-event data in distributed settings.” *arXiv preprint* arXiv:2311.16984 (2023).

[4] Han, Larry, et al. “Federated adaptive causal estimation (FACE) of target treatment effects.” *Journal of the American Statistical Association* (2025): 1–14.

**Questions:**

**1. Conceptual positioning and relation to prior work**
In what concrete ways does your approach differ from a meta-analysis of EIF-based local estimators, possibly preceded by a density-ratio reweighting step?
Could the proposed method be viewed as a *meta-analysis with adaptive weighting*, and if so, what theoretical or practical advantages does it offer?
More broadly, how does the framework go beyond FACE [5]? Since the EIF-based structure and penalized weighting are similar, what are the novel theoretical or computational insights specific to survival analysis?
What new theoretical guarantees are provided beyond those of FACE and Westling et al. [1]?

---

**2. Methodology, penalization, and communication**
How is the regularization parameter $\lambda$ in the penalized objective cross-validated in practice?
Is cross-validation performed locally, centrally, or through additional communication between sites?
Please also specify what quantities are exchanged across sites (EIF evaluations, sample moments, covariance matrices, or model parameters), especially for moment matching and density-ratio estimation.
This clarification would help determine whether the method is truly *federated* or rather *semi-centralized*.
Additionally, what is the communication complexity of the procedure in terms of the number of sites $K$, covariate dimension $d$, and time grid $\tau$?
Can the method accommodate time-varying covariates, and is it computationally scalable for large $d$ or fine-grained time grids?

---

**3. Assumptions and robustness**
Are there assumptions ensuring that the density ratios $\omega_{k,0}(X) = P(X|R=0)/P(X|R=k)$ are well-defined, particularly when $P(X|R=k)$ can approach zero under strong covariate shift?
The framework also assumes overlap (Assumption 2.3) across all sites—how robust is the method to *partial or local violations* of this assumption?
In such cases, meta-analysis often remains a strong competitor. Can you demonstrate scenarios where the proposed method outperforms simpler aggregated-data baselines?
Furthermore, could the approach extend to situations with multiple non-mergeable target sites or a target population defined as a subpopulation of the sources, as is common in observational studies?

---

**4. Empirical evaluation and interpretation**
The variance gain from the federated estimator relative to the local-target estimator appears small in the experiments.
Are there particular regimes (e.g., limited target size, moderate source bias, partial overlap) where the federated approach provides substantial efficiency improvements?
Additionally, why is the pooled estimator biased under shift scenarios in the simulations?
Would adjusting for the site indicator as a covariate in the pooled model yield comparable results?

---

**5. Privacy and model specification**
Could you specify which privacy risks the method actually mitigates and how this differs from standard aggregated-data sharing?
Since no formal privacy guarantees (e.g., differential privacy) are implemented, should the term *privacy-protected* remain in the title?
Finally, the density ratios are estimated via exponential-tilt models—could you justify calling these models “flexible,” and discuss how misspecification of the density-ratio model affects the bias or variance of the estimated survival functions?

---
Finally, I am willing to increase my grade if the authors properly address the above questions, especially my concerns on the framing of the article around privacy, explaining clearly what objects are communicated and the challenges in small sample sizes and high(er) dimensions, and the true contribution of the paper beyond the mere extension of the FACE article to the survival analysis setting.

---

### References

[1] Westling, Ted, et al. “Inference for treatment-specific survival curves using machine learning.”
*Journal of the American Statistical Association* 119.546 (2024): 1541–1553.

[5] Han, Larry, et al. “Federated adaptive causal estimation (FACE) of target treatment effects.”
*Journal of the American Statistical Association* (2025): 1–14.

---

> ### Author Response · Authors · 2025-11-22
> **Author response (1/6)**
>
> Thank you very much for your thorough and comprehensive review. Below, we summarized your comments, recommendations, and questions to a number of points, followed by our responses. We sincerely appreciate your time and consideration. All revisions have been incorporated into a new PDF uploaded to OpenReview; changes are marked in blue.
>
> In response to the reviewer’s concern about whether formal privacy protection is guaranteed and to avoid confusion with different privacy paradigms (e.g., differential privacy), we have replaced the term **“Privacy-protected”** with **“Federated” in the paper title.** Please see this change reflected in the PDF. We are currently looking into how to update the title directly on the OpenReview site. Since ICLR permits title modifications during the rebuttal period, we assure you that the title will be updated accordingly.
>
> **1. Theoretical novelty beyond prior work.**
>
> Upon further reflection, we agree that Theorem E.1 was previously underemphasized. We have now moved it to the main text as **Theorem 2.6** and added clarifying comments regarding the relevant regularity conditions, as well as the novel aspects of our results in terms of efficiency and double robustness. These additions highlight the theoretical contributions of our work beyond prior studies, including FACE (Han et al., 2025) [2] and Westling et al. (2024) [1].
>
> First, **Theorem 2.6.** (previous Theorem E.1) provides a key foundation and bridge for the federation procedure. It establishes that, under mild regularity conditions, the proposed source-site estimators are consistent and asymptotically normal, thereby ensuring that they can target the target-site estimand under appropriate conditions, an insight not discussed in FACE of Han et al.  [2].
>
> Second, the theory culminating in **Theorem 2.6** establishes the semiparametric efficiency bound for estimating the target-site survival functions under a weaker pairwise partial CCOD assumption. Importantly, it shows that each source-site estimator attains this efficiency bound under this assumption. Neither this pairwise assumption nor the corresponding efficiency theory were considered in Han et al. [2].
>
> Third, the accompanying regularity conditions (Conditions E.1–E.3) explicitly characterize the density-ratio term as a nuisance component and require its consistency and product-type error control, which is novel relative to Han et al. [2].
> The above novelties are now mentioned before **Remark 2.7** as a paragraph “Theoretical novelty”:
>
> “Theorem 2.6 establishes the semiparametric efficiency bound under the pairwise partial CCOD assumption in Theorem 2.5, and the source-site estimators attain this bound under this assumption. These results add theoretical novelty to prior work on continuous outcomes (Han et al., 2025). The interactions between the density ratio and the other nuisance functions also represent previously unexplored theoretical components.”
>
> We also highlight the **double-robustness** property of the source-site estimator (not established by Han et al., 2025), previously stated only in Remark E.4, and now included in the main text **(Theorem 2.8)** for clarity:
>
> *"For consistency of $\hat{\theta}\_n^k(t,a)$, it is not necessary that $(\pi\_{\infty}^k, \omega^{k,0}\_\infty, G\_\infty^k, S\_\infty^k)=(\pi^k, \omega^{k,0}, G^k, S^k)$ in Theorem 2.6 must hold. Instead, at any single time point $t$, if either (i) the conditional survival model $S^k$; or (ii) other nuisance functions $G^k$, $\pi^k$ and $\omega^{k,0}$ are correctly specified, $\hat\theta_n^k(t,a)$ is consistent."*
>
> **2. Difference to meta-analysis and positioning of our work.**
>
> The reviewer pointed out that our method is similar to a one-shot aggregation meta-analysis. We clarify that classical meta-analysis operates on site-level summary statistics (e.g., point estimates and variances), whereas our method aggregates **individual-level EIF summaries** from each site. This distinction is fundamental: EIF-based aggregation preserves information about the full empirical distribution at each site, which cannot be recovered from meta-analytic summaries alone. Second, the standard IVW estimator targets the **pooled population** across all $K$ sites, whereas our goal is to estimate the **target-site** survival functions ($R=0$). Using IVW to target the $R=0$ population would require either (i) the target estimand satisfies $\theta^0(t,a)=P(T^{(a)}>t\mid R=0)=P(T^{(a)}>t)$, or (ii) the distributions across sites are homogeneous, conditions that typically do not hold under site heterogeneity.
>
> We now add this point in the **“Related work”** section:
>
> “...Related meta-analysis approaches, such as aggregating site-specific estimators using inverse-variance weighting, possibly after density-ratio correction, also implicitly require such conditional homogeneity across sites …”

---

> ### Author Response · Authors · 2025-11-22
> **Author response (2/6)**
>
> We also extended **Remark 2.9** as:
>
> “...Our method also differs from meta-analysis (Borenstein et al. 2021), which relies only on coarse population-level summaries (such information is insufficient in our setting) and often targets the pooled population of all sites.”
>
> For these reasons, referring to our method as “meta-analysis with adaptive weighting’’ is not technically accurate, as both the estimand and the data-sharing structure differ from classical meta-analytic formulations.
>
> The reviewer also made a good point that our work needs stronger motivation of why the data-sharing constraint is an issue in this survival setting. We now highlight in Section 1 (i) the importance of survival data in high-impact domains; (ii) why integrating survival data can be helpful; (iii) that time-stamped event histories are often considered identifiable information under GDPR and HIPAA and thus is subject to data-sharing constraints. Please see the revised first paragraph of **Section 1:**
>
> “Data fusion is essential in many high-impact domains. For example, in medicine, clinicians assess how long treatments delay progression or readmission; and in finance, analysts track the time until a portfolio reaches a drawdown threshold. Across these settings, integrating survival data from multiple sources can improve efficiency, especially for rare events, and support broader causal conclusions. However, such integration (or data fusion) is challenging: distributional shifts in covariates, outcomes, or censoring can invalidate na\"ive pooling, and time-stamped event histories are considered identifiable information under General Data Protection Regulation (GDPR) and Health Insurance Portability and Accountability Act (HIPAA) regulations, limiting cross-institution data sharing. Federated learning provides a practical alternative by enabling collaboration through aggregate-level statistics rather than raw survival trajectories.”
>
> To more clearly situate ourselves and the setting we consider within the literature, we added a summary sentence of what we exclusively focus on and the definition of our setting in **Section 1:**
>
> “We consider multiple right-censored survival datasets each with two treatment groups, with restrictions on data sharing, and possible heterogeneity in covariates, outcomes, and censoring. Our goal is to estimate the survival function for a given target site while borrowing information from the additional source sites in a federated learning-based approach.”
>
> **3. Density-ratio details & information sharing across sites & time complexity.**
>
> Thank you for highlighting the need to clarify the term “flexible models.” We agree that our earlier usage could be confusing. The exponential tilt models in Han et al. (2025) are parametric and represent just one option for specifying the density-ratio model. In the revision, we removed the phrase “under flexible models” and now direct readers to **a newly added Remark 2.7,** where we discuss model choices, trade-offs, and the impacts of model misspecification.
>
> Regarding the model choices, we write:
>
> “To estimate the density ratio while respecting data-sharing constraints, a common approach is to adopt the exponential tilt model detailed in Han et al. (2025): $\omega^{k,0}(X) = \exp(\gamma_k'\psi(X))$, where $ \gamma_k$ is the model parameter and $ \psi(\cdot) $ is a set of basis functions of the covariates. A simple choice is $ \psi(X)= X $ for a linear component, and higher-order terms can be added to capture non-linearities in estimating $\omega^{k,0}(X)$. To estimate each $\gamma_k$ via maximum likelihood, only the target-site sample mean of $\psi(X)$ needs to be shared with the source sites.”
>
> Regarding the trade-offs by increasing model flexibility, we write:
>
> “...In addition to this model, more flexible nonparametric or machine learning approaches may be used, but these typically require sharing covariance matrices and/or other higher dimensional summaries.”
>
> For the impact of model misspecifications, we write:
>
> “...Finally, while the $\omega^{k,0}$ model may be misspecified, this does not necessarily invalidate our framework or estimators. As noted in Remark 2.8, our estimator is doubly robust: under Condition E.2, errors in estimating $\omega^{k,0}$ influence the final estimator only through a product-type term that enters the second-order remainder term. ”
>
> Regarding the reviewer’s question about what quantities are exchanged across sites and whether the method is truly federated, we have clarified that only target-site conditional survival model ($S^0$) parameters and summary statistics for density ratio estimation need to be shared from the target to source sites, and EIFs obtained by each site need to be sent to the leading analysis center, as shown in the updated Figure below.

---

> ### Author Response · Authors · 2025-11-22
> **Author response (3/6)**
>
> Therefore, our setting is **federated** since no raw participants covariate or outcome data are shared. We now clarified this in **Remark 2.9:**
>
> “...Source sites receive only the target-site $S^0$ model parameters and summary statistics for the density-ratio model, and the leading analysis center receives only EIFs…”
>
> In addition, the federated learning flow figure **(Figure 2)** was updated as below, where we added the information of target-site $S^0$ model parameters and summary statistics for $\omega^{k,0}$ are first transmitted to source sites:
>
> https://anonymous.4open.science/r/FuseSurvSubmission-3D16/flow_ICLR_v2.png
>
> The regularization parameter $\lambda$ is selected **centrally at the leading analysis center** via cross-validation. No raw data or iterative communication is required. The cross-validation is thus performed centrally. No additional communication between sites is required. We now clarified this in **Algorithm 1** explicitly:
>
> “...and $\lambda$ is a tuning parameter chosen by cross-validation centrally at the leading analysis center; no additional communication between sites is required.”
>
> For time complexity, we add in the **Discussion:**
>
> “For a discrete evaluation time grid of size $n_\tau$, site $k$ transmits an $n_k \times n_\tau$ matrix of subject-level EIF evaluations. Therefore, the total communication complexity is $O(n\cdot n_\tau)$, where $n=\sum_{i=0}^{K=1}n_k$. Future work should pursue smoothing strategies to capture temporal trends more efficiently to reduce such complexity to a lower level.”
>
> **4. Additional simulations for higher efficiency gains.**
>
> We appreciate the reviewer’s comment that our FED method has relatively small efficiency gains compared to TGT in experiments, which motivated us to explore additional simulation regimes. In particular, we examined settings where the target-site propensity score exhibits reduced overlap while the source sites maintain good overlaps, mimicking an observational target-population with partial overlap (labeled “limited overlap”). Under this regime, we found that our method can yield substantially higher efficiency. We have added the description of this setting in **Section 3.1 (Data generating process):**
>
> “...We also include a scenario with $n_k=300 (k=1,2,3,4)$ where the target-site propensity score $\pi^0(a\mid X)$ is more dependent on $X$ (‘limited overlap’) to highlight a regime with larger efficiency gains…”
>
> The updated simulation results are shown in the figure linked below, and we have replaced the main simulation results in the paper (Figure 3) with this updated figure. For brevity, the main text reports results only for the treated-arm ($A=1$) survival function under both good and limited overlap, while the full set of additional simulation results is provided in the appendix (Figures 8–11).
>
> https://anonymous.4open.science/r/FuseSurvSubmission-3D16/sim_main_ICLR_v2.pdf
>
> Under the limited-overlap regime, FED achieves substantially higher efficiency than TGT, as reflected by consistently narrower boxplots and smaller RRMSE across all scenarios. No competing method (IVW, META-IVW, or POOL) achieves both consistently higher efficiency and stable estimation across all cases. The intuition for the larger efficiency gains by FED under limited overlap is that source sites offer improved overlap, which increases the effective sample size for estimating the survival functions and allows FED to borrow information more effectively. We added the following comment to the simulation results in the main text:
>
> “...Under limited target-site propensity score overlap, FED also attains higher efficiency, likely because the improved overlaps at the source data help stabilize the source-site estimators.”
>
> In addition, we note that FED generally achieves higher efficiency at earlier time points within each setting, as the site-specific survival curves are more similar near the beginning of follow-up. This shows that FED borrows information in a time-specific manner. We add the following text to the simulation results:
>
> “...Across all scenarios in Appendix B.3, FED achieves larger efficiency gains at earlier time points, when site-specific survival curves more closely resemble the target (see Figure 5) and the source-site EIFs align better with the target….”
>
> We also clarify that while efficiency gains under good overlap may appear modest, or may not surpass IVW, META-IVW, or POOL in homogeneous settings, none of these methods achieves both consistently higher efficiency and consistency across all scenarios. By our theory **(Theorem 2.10),** FED improves efficiency relative to the semiparametrically efficient target-only estimator without imposing additional structural assumptions such as conditional homogeneity across sites (e.g., CCOD in Assumption 2.4) while maintaining consistency. This introduces a **bias–variance tradeoff** relative to other baselines, and we note that POOL does not even satisfy the data-sharing constraints.

---

> ### Author Response · Authors · 2025-11-22
> **Author response (4/6)**
>
> The reviewer also asked if the pooled estimator is expected to be biased even when the site indicator is included in the nuisance models. Our answer is **Yes**. This is because our estimand is the target-site survival curve, not the overall pooled survival curve. In the pooled analysis, the target site effectively becomes a subgroup, and the pooled fit cannot recover its site-specific survival function when there is conditional covariate or outcome heterogeneity across sites. Moreover, including the site variable in the nuisance functions does not resolve this issue, since these functions remain nuisance components and cannot transfer the overall estimand to the target site. We include the pooled estimator only to illustrate that its efficiency gains rely on a much stronger requirement that all sites be homogeneous in their distributions. This assumption is not made or needed by our method.
>
> **5. Additional baselines.**
>
> The reviewer recommended us to include stronger baselines in experiments (meta-analysis and FedECA [5]).
> For meta-analysis, we now included a density-ratio–weighted inverse-variance weighted estimator (labeled **“META-IVW”**) in all our simulation scenarios and two data applications. Specifically, we compute the target-site estimator and the density-ratio–weighted source estimators, and then combine them using standard inverse-variance weights. Note that the original IVW comparator (labeled as “IVW”) does not incorporate density-ratio adjustment. Our simulations show that META-IVW outperforms POOL and IVW (without density-ratio adjustment) overall and exhibits small bias under Homogenous, Covariate Shift and Censoring Shift, highlighting the benefit of density-ratio correction for this setting. However, its performance degrades substantially under Outcome Shift and All Shifts, where density-ratio adjustment alone cannot address the underlying heterogeneity.
>
> For FedECA, we noted that it addresses a fundamentally different problem from ours, so it is not applicable here. Specifically, FedECA constructs external control arms using federated inverse-probability-weighted (IPTW) Cox models for observational data, whereas our method targets site-specific causal survival curves under distribution shift in multi-source settings. The two frameworks therefore target different estimands. As we understand it, FedECA implicitly targets the population defined by the RCT treated-arm and reweights external observational controls to estimate the treatment effect in that pseudo-population. In contrast, our setting allows the target population to be either randomized or observational and assumes both treated and control units are available within that population.
>
> That said, our framework is related to the FedECA setting. Hence, we review this method now in the “Related work” section of Introduction:
>
> “Recent work such as FedECA (Ogier du Terrai et al. 2025) develops federated external control arms for single-arm trials, but this setting differs from the more general multi-source survival integration problem.”
>
> We also discussed its extension in the **Discussion** section:
>
> “..., adapting to external controls settings such as FedECA (Ogier du Terrai et al. 2025), as well as extending their inverse probability weighted-Cox approach to incorporate EIF and ensemble learning,...”
>
> **6. Estimator behavior.**
>
> The reviewer pointed out that the source-site estimator from **Theorem 2.5** does not seem to borrow information. While the EIF representation in **Theorem 2.5** appears locally defined, the estimator **does borrow information from the sources**, but only through the augmented components of the EIF. This design is intentional.
>
> We highlight this augmented term based on source data more clearly in Theorem 2.5 by labeling the components of the EIF as the “anchoring term using target data’’ and the “augmented term using source data’’; see below.
>
> https://anonymous.4open.science/r/FuseSurvSubmission-3D16/Th2.5updated.png
>
> The target-site component is anchored to preserve consistency and robustness, while the augmentation terms incorporate source-site outcome, treatment and censoring information after being reweighted by the optimal federated weights. These weights are learned by solving a moment-matching optimization that enforces alignment of EIF moments across sites. As a result, the effective influence function is a weighted combination of target and source EIFs, and its variance is reduced relative to the target-only estimator.
>
> In other words, federation occurs through (i) sharing summary statistics that characterize the EIFs at each site, and (ii) solving a global optimization that determines how much each site contributes to the final estimator. Although the procedure is similar to a one-shot aggregation, the optimization step is performed across the full set of source summaries and **directly targets variance reduction** under the constraint that consistency for the target estimand is preserved.

---

> ### Author Response · Authors · 2025-11-22
> **Author response (5/6)**
>
> Thus, we agree that the estimator generalizes the standard local estimator, in that it retains the target-site anchor for robustness. Importantly, however, it also uses source-site EIF information to construct an optimally weighted, federated estimator that can achieve efficiency gains, as formalized in **Corollary 2.11.**
>
> We have added this point at the end of **Section 2:**
>
> “We note from Corollary 2.11 that the efficiency gains rely on the consistency of at least some source estimator. To preserve robustness while borrowing information, our methodology anchors each source estimator to the target-site estimate and incorporates source data only through the augmented term of the EIF in Theorem 2.5. In the federation stage, we solve a global optimization problem that determines each site's contribution to the final estimator. While this resembles one-shot aggregation, the optimization operates over all source summaries and explicitly targets variance reduction while preserving the target estimand.”
>
> **7. Literature review, limitations and extensions.**
>
> We have incorporated references suggested by the reviewer [3–5] into the Introduction. Please see the updated **“Related work”** section (**previously labeled** “Limitations of existing work”).
>
> We now cite Xiong et al. [3] in the second sentence of the “Related work” section for federated causal data fusion methods.
> For Archetti et al. [4], we added “Archetti et al. (2023) have begun examining federated survival settings but they focus on data generation and simulation frameworks rather than estimation and inference.”
>
> For Ogier du Terrail et al. [5], we added “Recent work such as FedECA (Ogier du Terrail et al., 2025) develops federated external control arms for single-arm trials, but this setting differs from the more general multi-source integration problem.”
> This last addition makes clear that our setting does not align with “external control” settings, where source sites only contribute subjects under the control condition (i.e., $A = 0$).
>
> We also added some meta-analysis literature review as suggested by the reviewer for strengthening the connection between our method with one-shot aggregation approaches:
>
> “... Related meta-analysis approaches, which aggregate site-specific estimators using inverse-variance weighting, possibly after density-ratio correction, also implicitly require such conditional homogeneity assumptions across sites (DerSimonian & Laird, 1986; Marín-Martínez & Sánchez-Meca, 2010)...”
>
> We also now discuss the future extensions raised by the reviewer, including time-varying covariates, violations of positivity, limited covariate overlap and extensions to multiple non-mergable target sites.
> Our paper does not develop a detailed methodology for time-varying covariates, noting the fact that it is challenging for time-varying covariates to accommodate the continuous time setting. We acknowledge this limitation and mentioned in our Discussion: “Moreover, we did not incorporate time-varying covariates in our current framework due to the additional challenges they pose in continuous-time settings, but extending the method to leverage post-baseline information is an important direction for future work….”
>
> We also appreciate the helpful comments regarding the **positivity assumptions** for nuisance functions.
> For density ratio, as stated in **Condition E.2**, we explicitly require the density ratio and its estimate to be well-defined and bounded away from zero and infinity for $P$-almost all $\mathbf{x}$: “...$1/\eta \le \widehat{\omega}^{k,0}\_m (\mathbf{x}) \le \eta$ and $1/\eta \le \omega^{k,0}\_\infty(\mathbf{x}) \le \eta$...”
>
> Regarding the overlap assumption (Assumption 2.3) for other nuisances, we emphasize that it is mild in many survival settings. Conditional survival and censoring functions rarely take values near zero over a finite study period with pre-specified study horizon; for example, no participant has probability one of experiencing the event immediately. The condition $P(R=k) \ge 1/\eta$ is also reasonable because sites with extremely small sample sizes and proportions contribute limited information and are not useful for federation. For the treatment propensity score $\pi^k(a\mid X)$, randomized trials satisfy this assumption by design, such as $\pi^k(a\mid X) = 0.5$ under a 1:1 allocation.
>
> In observational studies, positivity violations on propensity score may arise, and as noted in the Discussion, these cases will be addressed in future extensions (Cheng et al., 2022; Xue et al., 2024):
>
> “Additionally, violations of the positivity assumption can render target estimand unidentifiable, e.g., when the two treatment groups in some sites differ systematically in their covariate distributions, or when certain participants are ineligible for specific treatments. Future work should investigate or leverage techniques to address such violations in our framework (Cheng et al., 2022; Xue et al., 2024).”

---

> ### Author Response · Authors · 2025-11-22
> **Author response (6/6)**
>
> In addition, the issue of **limited covariate overlap (different covariate variables across sites)** lies outside the scope of our current setting, so we have instead added a brief clarification regarding this limitation in the **Discussion** section:
>
> “...Furthermore, future work should consider settings where covariates differ across sites or have limited overlap; in such cases, density ratio estimation becomes difficult and requires additional sensitivity analysis…”
>
> The reviewer is interested in whether our method can naturally extend to settings with multiple non-mergeable target sites as well as to cases where the target is a subpopulation. The latter is straightforward, as one region can simply serve as the target while all participants across regions act as potential sources (e.g., our AMP trials analysis). For multiple target sites, each target can be treated as its own anchor, with a separate federated aggregation problem solved for each. When these target sites are comparable (e.g., satisfy the CCOD in Assumption 2.4), transfer learning methods may further improve nuisance estimation.
> Accordingly, we added the following extension to the **Discussion:**
>
> “...Our method also extends to multiple non-mergeable target sites by anchoring each one separately and solving a target-specific federated aggregation problem. When target sites are comparable (e.g., satisfy CCOD), transfer learning may be leveraged to further improve nuisance estimation.”
>
> **8. Notation improvement & other clarifications:**
>
> Finally, we improved the consistency of notation throughout. We now add in **Section 2** (see **“Target estimand”** paragraph):
> “Throughout, \mathbb{P} denotes the population-level probability under the true data-generating process, and with a subscript ‘$n$’, $\mathbb{P}\_n[f(\mathcal{O})] = n^{-1}\sum_{i=1}^n f(\mathcal{O}_i)$ denotes the empirical average.”
> In addition, we add below notation clarification for $\widehat{\mathbb{P}}$ in **Section 2.2:**
>
> “Furthermore, we use $\widehat{\mathbb{P}}$ to denote the EIF evaluated with estimated nuisance functions. This should not be confused with the empirical average $\mathbb{P}\_n$ introduced earlier. The same convention applies to all other EIFs throughout the paper.”
>
> For other notation such as expectation $\mathbb{E}$ and variance $\mathbb{V}$, since they only appear in the appendix, we now add the below clarifications at the **beginning of Appendix E:**
>
> “We begin by recalling notation for probability, expectation, and variance. Throughout, $\mathbb{P}$ denotes the true probability under the data-generating distribution, $\mathbb{P}\_n$ the empirical average, and $\widehat{\mathbb{P}}$ the evaluation with estimated nuisance functions (as introduced in the main text). In addition, $\mathbb{E}$ denotes the population expectation, $\mathbb{V}$ the population variance, and $\text{Cov}$ the population covariance.”
>
> *Again, we sincerely thank for your very constructive feedback that substantially improves our paper. We are happy to answer any remaining questions and concerns.*
>
> **References**
>
> [1] Westling, Ted, et al. “Inference for treatment-specific survival curves using machine learning.”
> Journal of the American Statistical Association 119.546 (2024): 1541–1553.
>
> [2] Han, Larry, et al. “Federated adaptive causal estimation (FACE) of target treatment effects.” Journal of the American Statistical Association, 120.551  (2025), 1503–1516.
>
> [3] Xiong, Ruoxuan, et al. “Federated causal inference in heterogeneous observational data.” Statistics in Medicine 42.24 (2023): 4418–4439.
>
> [4] Archetti, Alberto, et al. “Heterogeneous datasets for federated survival analysis simulation.” Companion of the 2023 ACM/SPEC International Conference on Performance Engineering (2023).
>
> [5] Ogier du Terrail, J., Klopfenstein, Q., Li, H. et al. “FedECA: federated external control arms for causal inference with time-to-event data in distributed settings.” Nat Commun 16, 7496 (2025).
>
> [6] Cheng, Chao, et al. "Addressing extreme propensity scores in estimating counterfactual survival functions via the overlap weights." American journal of epidemiology 191.6 (2022): 1140-1151.
>
> [7] Xue, Wu, et al. "RKHS-based covariate balancing for survival causal effect estimation." Lifetime Data Analysis 30.1 (2024): 34-58.
>
> [8] DerSimonian, Rebecca, and Nan Laird. "Meta-analysis in clinical trials." Controlled clinical trials 7.3 (1986): 177-188.
>
> [9] Marín-Martínez, Fulgencio, and Julio Sánchez-Meca. "Weighting by inverse variance or by sample size in random-effects meta-analysis." Educational and Psychological Measurement 70.1 (2010): 56-73.
>
> [10] Michael Borenstein, Larry V Hedges, Julian PT Higgins, and Hannah R Rothstein. “Introduction to meta-analysis.” John Wiley & Sons, 2021.

---

### Official Review · Reviewer_shKn · 2025-10-29

**Soundness:** 3
**Presentation:** 3
**Contribution:** 2
**Rating:** 6
**Confidence:** 3

**Summary:**

The paper proposes a federated learning strategy to estimate population-level causal effects in survival outcomes, combining the data of multiple sources, under the violation of the _common conditional outcome distribution_ (CCOD) assumption.
The strategy is composed by several steps:
1) Each different site estimate the nuisance functions for its own data: _survival function ($S$), cumulative hazard function ($\Lambda$), propensity score ($\pi$), the conditional survival function of censoring ($G$), etc._.
2) Also, global parameters have to be estimated from coarse statistics: $w^{k,0} = \frac{P(X|R=0)}{P(X|R=k)}$, $P(R=k)$, etc.
3) In each site, the target parameter, i.e., the treatment-specific survival function has a closed-form solution, given the nuisance functions and the coarse statistics.
4) Weights for collaborative learning are computed for each time-point and each site, aligning the EIF with the target distribution, and introducing an $\ell_1$ loss to remove the sites that do not contribute positively to the estimation of the target parameter in the target site. This problem is solved by optimization.
5) A federated estimator that combines the sites is the final estimation of the targed parameters.
The method has been validated over 1) synthetic data in which the distribution shifts are varied  on demand, and 2) real AMP and _flchain_ data (although _flchain_  is defered to the appendix).

Results show better RMSE and coverage percentage compared with Pooling and IVW, especially when there exist covariate and outcome shift.

**Strengths:**

- The motivation of the paper is clear and strong, the authors aim to give an unbiased estimators of population-level causal effects in survival outcome, given that the CCOD (common conditional outcome distribution) assumptions does not hold.

- The theoretic contribution is based on _Efficient influence function_ theory, and, although I have not studiend in detail the proofs of Appendix E, they seem to be consistent and valid.

- The results present clearly the improvements in comparison with pooling the data and using Inverse Variance Weighting (IVW). Both RMSE and CP show that TGT and FED are consistently better in synthetic data. The results in real data are not that conclusive, but still points in the right direction.

**Weaknesses:**

- My main concern relies on the data-adaptive weighting criterion. The parameters $\eta$ are the coefficients of a linear regression (omitting $\ell_1$) to fit the _local_ EIF ( $\hat{\varphi}_{t,a}^{*0}$ ), with the site specific EIFs. Therefore, if the local EIF is not well specified, the other sites may contribute negatively to the estimation. That is, the federated learning strategy can be harmful if the local-estimator is not well specified. Is not that exactly what federated learning tries to fight? The same thing happens with the $\ell_1$ regularization: the optimization process will remove the $\theta$ that are far from the local estimation, thus benefiting the errors in the local estimation to persist. Is there any explanation for this fact, or any way to fight this positive feedback?

- I have marked the contribution as _fair_ because this is an adaptation of (Han, 2025) to time-to-event data. Although that does not prevent me for accepting the paper, I do not see the contribution as disruptive.

- Results in real data are not conclusive. All the charts present similar results between TGT and FED. Coverage percentage is the most of the time equal, and metrics about RMSE statistical difference are not provided. This limits the interpretation and the strength of the results.

**Questions:**

> Minor concerns

- In the introduction, it seems that no previous work on _federated causal survival analysis_ have been done (especially in `line 054`: 'these methods remain focused on single-study...'). I miss some references to Van der Laan collaborative learning [1, 2]. So I would add some information about this work and its (likely) follows ups. (disclaimer: I have nothing to do with those works.)

- Can we get a explicit definition of what nuisance functions are? From the context, it can be understood that _survival function, hazard function, propensity score, etc._, are nuisance functions, but is not clear in the introduction.

- Can authors provide more intuition about  Equation of line 128? E.g. what the role of $\frac{I(R=0)}{P(R=0)}$ is? what the role of $\frac{I(A=a)}{\pi(a|X)}$ is? It would help to understand the equation.

- What does the RAL property in Appendix E.1.2 mean? It does  not seem to be defined.

- In `line 430-431`, authors say thet TGT yields wider confidence intervals than FED, but I cannot see that in the Figure 4(A). In fact, what I observe is that FED, between aproximately day 80 and day 180, has wider intervals than TGT. Am I missing something?

- Figure 4 (B) represents the weights given to each site in each timepoint. We can observe two curves, one very noisy, and another _filtered curve_. The _smoothing_ component is part of the approach or is it only a representation tool in the figure?

- In general, I would recommend to add more intuitions about theoretical implications and, especially, under which assumptions the oracle-optimal weights are recovered. The pointers to the appendix are overwhelming, and more plain intuitions would be very helpful

> Summary

In general, I would say that this is a good paper. It is based on other similar approaches that leverages _Efficient influence function theory_ and both the theoretical approach and the results are sound. However, I have several concerns, reflected in _Weaknesses_ and _Questions_ section that prevents me to give a higher score. I would be happy to raise my score if those concerns are solved.

> References

[1] van der Laan, M. J., & Gruber, S. (2010). Collaborative double robust targeted maximum likelihood estimation. The international journal of biostatistics, 6(1), 17.
[2] Stitelman, O. M., & van der Laan, M. J. (2010). Collaborative targeted maximum likelihood for time to event data.

---

> ### Author Response · Authors · 2025-11-22
> **Author response (1/4)**
>
> Thank you very much for your positive feedback, thoughtful review and helpful suggestions, which have substantially improved the clarity and overall quality of the manuscript.  Please see our point-by-point response below. **We have also updated the paper in PDF to OpenReview; changes are marked in blue.**
>
> We also appreciate your openness to assigning a higher score. Please let us know if you have any additional questions or comments. We would be happy to address them.
>
> **Main Concerns**
>
> **1. Would the federated weighting and L1-regularization be harmful?**
>
> Thank you very much for this insightful comment and observation. We agree with the reviewer that “the optimization process will remove the $\theta$ that are far from the local estimation, thus benefiting the errors in the local estimation to persist”. We acknowledge that a consistent target estimator is required, and that to achieve strictly positive efficiency gains, at least some source estimators must also be consistent, as stated in **Corollary 2.11.**
>
> We clarify that we are fully aware of this fact, and our methodological design incorporates **a crucial safeguard: anchoring each source estimator to the target-site estimate.** In the EIF in Theorem 2.5, only the augmented component draws on source-site information. Although this limits the amount of information borrowed from the sources, it protects the robustness of each source estimator and ensures that it continues to target the target-site estimand consistently.
>
> Below, we justified that in practice that the target and source estimators can achieve the consistency.
> First, while the proposed methodology allows each source and target to be either randomized or observational in general, in many applications for survival data, the treatment is assigned with a known randomization probability to participants. Therefore, in randomized trials (e.g., our AMP HIV trials in the data applications), the target-site estimator will be consistent because the propensity score for treatment ($\pi^0(a\mid X)$) is known and the EIF-based estimator has a double robustness property. Therefore, consistency of the estimator is guaranteed to be achieved when the treatment is randomly assigned in the target population (more precisely, when the treatment allocation probabilities are known by design).
>
> Second, for observational studies, we typically have large sample sizes, so that flexible nonparametric and machine learning models (e.g., ensemble learners like SuperLearner and survSuperLeaner that include a comprehensive library of component learning methods) used with our method can improve robustness to model misspecification of nuisance functions.
> We now add these points at **the end of Section 2:**
>
> “We note from Corollary 2.11 that strict efficiency gains rely on the consistency of at least some source estimators. To preserve robustness while borrowing information, our methodology anchors each source estimator to the target-site estimate and incorporates source data only through the augmented term of the EIF in Theorem 2.5. In the federation stage, we solve a global optimization problem that determines each site's contribution to the final estimator. While this resembles one-shot aggregation, the optimization operates over all source summaries and explicitly targets variance reduction while preserving the target estimand.”
>
> “....Although the framework allows both randomized and observational sites, many survival-data applications involve randomized treatment assignment, in which case the propensity score $\pi^0$ is known and the target-site estimator is consistent. For observational studies, it is common to have larger sample sizes that enable flexible nonparametric and machine learning methods to estimate nuisance functions with greater robustness to model misspecification.”
>
> **2. Contribution & theoretical novelty clarifications**
>
> We appreciate the thoughtful assessment about our contribution by the reviewer. We acknowledge that some motivating elements and specific techniques of our work follow Han et al. (2025). We understand the reviewer’s feeling that the contribution is not so disruptive, and we appreciate the comment that this would ultimately not preclude acceptance of the paper.
>
> In light of this comment, and to better highlight the novelty and situate the unique contributions of our work, in the revised paper we have clarified some aspects that represent **nontrivial** extensions and novel insights beyond the work of Han et al. (2025).
>
> First, in response to **Reviewer n3Hz,** we now place greater emphasis on Theorem E.1 from the previous version, which on further reflection represents a particularly novel perspective and theoretical underpinning for the proposed federated learning approach; analogous results were not described in Han et al. (2025). We have now moved this result to the main text as **Theorem 2.6.**

---

> ### Author Response · Authors · 2025-11-22
> **Author response (2/4)**
>
> As mentioned, this theorem provides the key foundation and bridge for the federation step. Namely, it establishes that, under mild regularity conditions, the proposed individual source-site estimators are consistent, asymptotically normal, and semiparametrically efficient under a weaker pairwise partial CCOD assumption. This shows how and when outcome data from sources that are sufficiently similar to the target site can be effectively used to estimate the target-site estimands, an insight not discussed in Han et al. (2025). In addition, the accompanying regularity conditions (Conditions E.1–E.3) explicitly characterize the density-ratio term as a nuisance component and its consistency and product-type error control, which gives new insights relative to Han et al. (2025).
> The above novelties are now mentioned before Remark 2.7 as a paragraph **“Theoretical novelty”:**
>
> “Theorem 2.6 establishes the semiparametric efficiency bound under the pairwise partial CCOD assumption in Theorem 2.5, and the source-site estimators attain this bound under this assumption. These results add theoretical novelty to prior work on continuous outcomes (Han et al., 2025). The interactions between the density ratio and the other nuisance functions also represent previously unexplored theoretical components.”
>
> We also highlight the double-robustness property of the source-site estimator (not established by Han et al., 2025), previously stated only in **Remark E.4,** and now included in the main text **(Theorem 2.8)** for clarity:
>
> “For consistency of $\widehat\theta\_n^k(t,a)$, it is not necessary that $(\pi\_\infty^k, \omega^{k,0}\_\infty, G\_\infty^k, S\_\infty^k)=(\pi^k, \omega^{k,0}, G^k, S^k)$ in Theorem 2.6 must hold. Instead, at any single time point $t$, if either (i) the conditional survival model $S^k$; or (ii) other nuisance functions $G^k$, $\pi^k$ and $\omega^{k,0}$ are correctly specified, $\widehat\theta_n^k(t,a)$ is consistent.”
>
> **3. Non-conclusive results & no RMSE in data analysis**
>
> We appreciate this careful observation and insightful comment. FED yields a curve with a shape similar to that of TGT because, by design (as we also responded in detail in the first comment), it anchors the target-site estimate while incorporating external information for improving efficiency. As a result, FED still depends primarily on the target-site sample size, nuisance models, and corresponding estimates, so its point estimates are expected to remain relatively close to those of TGT, an intentional choice that serves to preserve consistency. Although **Theorem 2.11** does not guarantee that FED will always achieve strictly and substantially higher efficiency than TGT, it guarantees at least that FED can be reduced to TGT if none of the source sites are informative (i.e., provide consistent estimates), so that the efficiency of FED is no-smaller than that of TGT.
>
> To highlight an example where greater precision is achieved by FED, **Figure 4** shows that under $A=1$, the FED curve generally exhibits narrower confidence intervals. This is also reflected in the relative efficiency metrics at the three selected days. Ratios of the estimated variances of FED to those of TGT are 0.51, 0.87, and 0.97 (all < 1), respectively.
>
> Additionally, we clarify that the RMSE metric (as well as Bias and CP) cannot be evaluated for a single real-world dataset because the true survival functions are unknown. That is, given the nature of these datasets, there is no definitive benchmark that can be treated as the truth; imposing one would be misleading and risky in guiding practice. Thus, the efficiency comparisons rely on estimated variances for illustrative purposes. We assess RMSE only in simulations, where the truth is known and repeated Monte Carlo replications are possible.
>
> Furthermore, we remark again that our proposal highlights a clear **bias–variance trade-off** relative to the baselines (POOL, IVW, and **a newly added META-IVW** in the revision). These baselines implicitly rely on conditional homogeneity of outcomes (and sometimes covariates) across sites to remain consistent. Although they may be more efficient under favorable conditions (e.g., the Homogeneous simulation settings), they become highly biased and unstable under outcome or covariate distributional shifts, as shown in our simulations. In contrast, our method delivers the most robust performance across all scenarios while still offering moderate efficiency gains over the target-only estimator. Importantly, **we do not impose any conditional homogeneity assumptions on the source sites relative to the target.** To our knowledge, improving efficiency relative to a semiparametric efficient estimator without additional restrictions is challenging, yet our estimator achieves this as long as some of the source sites provide consistent estimates.
>
> We hope this clarifies the purpose and some of the properties of our methodology, and resolves any potential misunderstandings.

---

> ### Author Response · Authors · 2025-11-22
> **Author response (3/4)**
>
> **Minor Concerns**
>
> **1. Additional literature review**
>
> Thank you for bringing additional papers (Van der Laan collaborative learning [1, 2]) to our attention. The first paper [1] does not address  time-to-event outcomes, but the second one [2] does. In addition, **neither method considers a multi-source setting;** even though the term “collaborative” appears in both works, it refers to a modeling concept rather than collaboration across data sources.
> Targeted maximum likelihood estimation (TMLE) is often used to improve the finite-sample performance of the conventional doubly robust estimator of the average treatment effect, for example by yielding range-preserving estimates for binary outcomes. These works [1,2] propose a collaborative-targeted maximum likelihood estimator (C-TMLE) as a further extension of the TMLE. The notion of “collaboration’’ in their framework reflects a “collaborative double robustness’’ property of their proposed estimator, meaning that the C-TMLE estimator can remain consistent even when both the propensity score and outcome regression models are misspecified, in contrast to the standard “double robustness” property that requires at least one of the models to be correctly specified.
> That said, these are valuable literature for estimation methodology and we now add the second literature to read as:
>
> “In addition, targeted maximum likelihood estimation (TMLE) can improve the finite-sample performance of doubly robust estimators (van der Laan & Rubin, 2006; Díaz et al., 2019), and the collaborative TMLE (C-TMLE) further enhances robustness to model misspecification (Stitelman & van der Laan, 2010).”
>
> In the last section, we added that future extension can focus on extending the TMLE and C-TMLE to our setting (both [1,2] are cited):
> “... Extensions include incorporating alternative estimators such as TMLE and C-TMLE  (van der Laan & Rubin, 2006; Stitelman & van der Laan, 2010; van der Laan & Gruber, 2010),...”
>
> **2. Nuisance function definition**
>
> The term “nuisance function” refers to auxiliary quantities that are not of primary scientific interest but are necessary for estimating the target parameter. This terminology is commonplace in nonparametric statistics and causal inference. In our case, the parameter of interest (estimand) is the treatment-specific survival function $\theta^0(t,a)$, and all other related quantities, including the propensity score, survival function and censoring function, are nuisance functions, and we only care about the inference of the main parameter, but not the nuisance functions.
> We now add
>
> “These quantities are referred to as nuisance functions, auxiliary components that are not of primary scientific interest but are essential for estimating the target parameter $\theta^0(t, a)$.”
>
> **under Assumption 2.3** when first introducing related nuisance functions.
>
> **3. The role of weighting terms**
>
> We appreciate the suggestion for adding more intuition about these weighting terms  $\frac{\mathbb{I}(R = 0)}{\mathbb{P}[R = 0]}$ and $\frac{\mathbb{I}(A =a)}{\pi^0(a\mid X)}$, which will increase clarity for readers who are not familiar with influence functions. The term $\frac{\mathbb{I}(R = 0)}{\mathbb{P}[R = 0]}$ reveals, through its numerator, that the nonparametric efficient influence function only uses data from the target site, with the denominator $\mathbb{P}[R = 0]$ appropriately re-scaling these contributions. Similarly, the terms multiplied by $\frac{\mathbb{I}(A =a)}{\pi^0(a\mid X)}$ only involve data for those treated with $A = a$ (which is needed for learning about the effect of an intervention that sets treatment to this value $\theta^0(t,a)$), while the propensity score weight in the denominator reweights these contributions to generalize to the whole population of interest.
>
> We now add the following interpretation of these weighting terms under the EIF of **Section 2.2** to help understanding:
>
> “...Furthermore, the weighting term ${\mathbb{I}(R=0)}/{\mathbb{P}(R=0)}$ selects target-site observations, and ${\mathbb{I}(A=a)}/{\pi^0(a\mid X)}$ restricts to units with treatment $A=a$ while reweighting them to represent the full target population.”
>
> **4. Definition of RAL**
>
> We apologize for omitting the definition of RAL in the paper. It stands for “regular and asymptotically linear”, a term which we take from the literature on semi- and non-parametric statistics. We now add in the main text in **Section 2** about its meaning:
> “A central result of this paper is the regular and asymptotically linear (RAL) property of the local estimator $\widehat\theta^{k,0}_n(t,a)$, stated in the following theorem. An estimator is RAL if it can be written as an i.i.d. average of influence functions plus a negligible remainder. This property allows the central limit theorem to be applied to obtain its asymptotic normal distribution....”

---

> ### Author Response · Authors · 2025-11-22
> **Author response (4/4)**
>
> **5. Clarification of results in Figure 4(A)**
>
> In Figure 4(A), FED appears to have wider confidence intervals between approximately days 80 and 180 (and at some earlier time points) as pointed out by the reviewer. We clarify that this is because TGT does not yield any variance estimates (i.e., they appear as “NA” in the output), so it is not because its variance is smaller. This issue is consistent with the findings reported by Westling et al. (2024). The primary reason is the insufficient effective sample size of individuals who experience the event early in the time axis. At these early time points, only a small number of events occur within the target site, making the influence-function–based variance estimator unstable or undefined. In contrast, several of these interval estimates are recovered under FED because borrowing information from external sites effectively increases the available sample size for estimating early hazards, thereby stabilizing the variance estimates.
>
> Consequently, the TGT curve lacks valid variance estimates at these time points, and no corresponding confidence intervals can be computed. In contrast, our FED method recovers interval estimates for some of these early time points. We acknowledge that the wording in lines 430–431 was confusing and will revise it to read:
>
> “TGT fails to yield valid intervals at certain early time points due to unstable or unavailable variance estimates, driven by the insufficient effective sample size individuals who experience the event at those times…”
>
> **6. Weight smoothing tool in Figure 4(B)**
>
> The federated weights are derived at each time point and can therefore be noisy; we have acknowledged this limitation in the Discussion section. The smoothing component in Figure 4(B) serves solely as a visual representation tool.
> For clarity, we have added
>
> “(only a representation tool)”
>
> to **the caption of Figure 4(B)** as well as figures in appendix for additional data analyses.
>
> **7. More intuition about conditions for the oracle-weighting**
>
> We appreciate that the reviewer asked for more intuition about the optimal weighting conditions. We now add some plain language summary in **Remark 2.13** (Efficiency gains):
>
> “...Under the mild regularity conditions stated in Appendix E.2, namely compact covariate support, bounded density ratios, and finite variance–covariance matrices of the EIFs across sites, we show that our federated estimator recovers the following oracle-optimal weights: …”
>
> *Once again, we greatly appreciate your thorough review and constructive feedback. Please feel free to let us know if further clarification is needed.*
>
> **References**
>
> [1] van der Laan, Mark J. and Gruber, Susan (2010). "Collaborative Double Robust Targeted Maximum Likelihood Estimation" The International Journal of Biostatistics, vol. 6, no. 1.
>
> [2] Stitelman, Ori M and van der Laan, Mark J (2010). "Collaborative Targeted Maximum Likelihood for Time to Event Data" The International Journal of Biostatistics, vol. 6, no. 1.

---

> > ### Comment · Reviewer_shKn · 2025-11-26
> > **Rebuttal acknowledgement and response to authors**
> >
> > I think that almost all the concers I raised have been addressed, and I will increase my score accordingly.
> >
> > In general, I think that the paper is now improved, the implications of the factors of the formulas and the contributions of the paper are clearer, the literature review is more complete, and implications of the theory, especially about the regularization and the _anchoring term_ are more understandable now. Other minor concerns that I had, regarding visualization tools, the definition of nuisance functions, and the clarification of the results of Figure 4 have been addressed.
> >
> > I checked, although not in detail, the concerns of the rest of reviewers and I see that authors have addressed succesfully the most of them. Therefore I think this paper is a good contribution.

---

> > > ### Author Response · Authors · 2025-11-26
> > >
> > > Dear Reviewer - Thank you very much for your positive feedback on our rebuttals! We are glad to hear that our clarifications addressed your concerns. Please feel free to let us know if any additional questions arise. Thanks again for your insightful comments and for your hard work on our manuscript! - Authors

---

### Official Review · Reviewer_YJNV · 2025-11-03

**Soundness:** 3
**Presentation:** 3
**Contribution:** 3
**Rating:** 6
**Confidence:** 5

**Summary:**

A federated learning framework for survival analysis is a missing topic. This paper presents a federated learning framework for causal survival analysis that aims to estimate treatment-specific survival functions across multiple heterogeneous datasets while respecting privacy constraints. The formulation mostly follows previous papers where we have to adjust the distribution shift between the source and the target dataset.

**Strengths:**

The paper does a good job transitioning the federated causal inference problems to this setting. They provide a complete and clean framework. I list the strengths below.

1. Timely and important setting
2. Clear formulation
3. Doubly robust estimator
4. Accounting for biased source states
5. Complete simulation and real-world studies

**Weaknesses:**

I believe that there are some important questions that the paper should answer:

1. Comparison with standard setting. Consider a degenerated case in survival analysis, where we do not have censoring and only have time step 1. This becomes the non-survival setting. In this case, does the estimator proposed degenerated to estimators the same with Han et al.? Answering this question can help the reader understand any tricky or nontrivial part in extending the estimator to survival analysis.

2. Estimating the distribution shift is the core problem in causal inference. It would be interesting to see discussions on different strategies on density ratio analysis and their effect on the final result.

3. It would be interesting to see the extension to better estimators, for example, Guo et al. (2024) proposes an estimator that is more efficient than meta-analysis styled estimators.

Guo, Tianyu, Sai Praneeth Karimireddy, and Michael I. Jordan. "Collaborative heterogeneous causal inference beyond meta-analysis." arXiv preprint arXiv:2404.15746 (2024).

**Questions:**

None

---

> ### Author Response · Authors · 2025-11-22
> **Author response (1/2)**
>
> Thank you for your helpful comments, which substantially improved the clarity of our paper. Please see our responses to your points and questions below. **We have also uploaded the revised PDF to OpenReview; changes are marked in blue.**
> Please let us know if you have any further questions or concerns. We would be grateful if you could review the changes in the revised paper overall and reassess whether it merits a higher score.
>
> **1. Comparison with the setting of FACE when there is no timing.**
>
> Thank you for this excellent insight. In the absence of timing and censoring, our estimator reduces to the FACE estimator of Han et al. (2025). To see this, consider a single time point $\tau$. The target estimand reduces to
>
> $$\theta^0(\tau, a) = \mathbb{E}[\mathbb{I}(T^{(a})>\tau)\mid R=0],$$
>
> which is the mean of a binary potential outcome. This is exactly a standard binary-outcome causal estimand. While Han et al. (2025) study the target-site ATE $\theta^0 = \mathbb{E}[Y(1)-Y(0)]$ under continuous outcomes, their results naturally apply to treatment-specific means of binary outcomes. The ensuing proposed efficiency theory and weighting approach also align with FACE in this case, where federated weights for survival outcomes are learned at different time points. We agree that making this connection explicitly will be helpful to readers and include the following in the **Discussion:**
>
> “In the absence of timing and censoring, our estimator reduces to the FACE estimator of Han et al. (2025) when the survival outcome is replaced by the binary indicator $\mathbb{I}(T^{(a)}>\tau)$. With censoring (i.e., missingness in this binary outcome), FACE would require modification to incorporate inverse-probability weights under a missing-at-random assumption.”
>
> Please also refer to our response to **Reviewer n3Hz** for further discussion on additional points of novelty in our extension of FACE to this survival setting, including the double robustness of the source-site estimator (new **Theorem 2.8** in the paper), rigorous establishment of regularity conditions of nuisance functions, semiparametric efficiency of the source-site estimator under the pairwise CCOD assumption, and insights regarding the rates of convergence needed for the density ratio model.
>
>
> **2. Discussion on different density ratio strategies**
>
> Thank you for this thoughtful suggestion. We have now extended our discussion of density ratio estimation and impact of potential model misspecifications in Section 2. Please see our added **Remark 2.7** for details.
> For estimation details, we now write:
>
> “To estimate the density ratio while respecting data-sharing constraints, a common approach is to adopt the exponential tilt model detailed in Han et al. (2025): $\omega^{k,0}(X) = \exp(\gamma_k'\psi(X))$, where $ \gamma_k$ is the model parameter and $ \psi(\cdot) $ is a set of basis functions of the covariates. A simple choice is $\psi(X)= X$ for a linear component, and higher-order terms can be added to capture non-linearities in estimating $\omega^{k,0}(X)$. To estimate each $\gamma_k$ via maximum likelihood, only the target-site sample mean of $\psi(X)$ needs to be shared with the source sites.”
>
> Regarding the trade-offs of using more flexible models, we write:
> “In addition to this model, more flexible nonparametric or machine learning approaches may be used, but these typically require sharing covariance matrices and/or other higher dimensional summaries.”
>
> For the impact of model misspecifications, we now add:
>
> “...Finally, while the $\omega^{k,0}$ model may be misspecified, this does not necessarily invalidate our framework or estimators. As noted in Theorem 2.8, our estimator is doubly robust: under Condition E.2, errors in estimating $\omega^{k,0}$ influence the final estimator only through a product-type term that enters the second-order remainder term. ”

---

> ### Author Response · Authors · 2025-11-22
> **Author response (2/2)**
>
> **3. Extensions to better estimators.**
>
> Thank you for pointing us to this helpful literature. We agree that Guo et al. (2024) offer an interesting direction for improving efficiency in heterogeneous settings, and their framework provides methodological ideas that inspire future developments. The method proposed by Guo et al. (2024) could potentially enhance adjustment for distribution shift beyond covariate shift through the “collaborative propensity score” weighting.
>
> Based on reading Guo et al. (2024), we note that additional efficiency gains may come at the cost of additional assumptions, such as a collaboration-selection mechanism for individuals (their Assumption 1). This suggests that the Guo et al. approach operates within a particular regime of cross-site similarity, whereas our method does not explicitly impose such restrictions. We also observe that their Assumption 3 (overall overlap/positivity), combined with the collaboration-selection mechanism, may be weaker than the site-specific positivity assumption required in our current formulation. It would therefore be valuable for future work to examine how our federated weighting and density-ratio framework might be extended to operate under this weaker, combined-source positivity structure, allowing the positivity requirement on the density ratio to be relaxed in our setting, and to assess whether such extensions could further improve efficiency for right-censored survival data.
>
> We therefore add the following discussion to the **“Limitations and Future Directions”** section:
>
> “...Finally, although our density-ratio weighting effectively addresses covariate shift, investigating alternative weighting strategies, such as extending the collaborative propensity score weighting (Guo et al., 2024) to survival data, is left for future work.”
>
> *Again, we sincerely appreciate your careful review of our paper and your insightful comments. Please let us know if any additional clarification would be helpful.*
>
> **Reference**
>
> Guo, Tianyu, Sai Praneeth Karimireddy, and Michael I. Jordan. "Collaborative heterogeneous causal inference beyond meta-analysis." arXiv preprint arXiv:2404.15746 (2024).

---

### Author Response · Authors · 2025-11-22
**Global Response**

We thank our area chair and all reviewers for their hard work on our manuscript. We appreciate all thoughtful and constructive feedback provided. In this global response, we summarize the main strengths they highlighted, followed by the key concerns raised across reviews and how we have addressed them in the revision. **We have also uploaded the revised PDF to OpenReview; changes are marked in blue.**

**Strengths**

- **Timeliness and importance of the problem (YJNV, shKn, n3Hz).** All reviewers agree that federated causal inference for survival outcomes is an important and underdeveloped area, and that our work addresses a relevant gap in settings with distributional shift, privacy constraints, and heterogeneous multi-site data.

- **Clear formulation and overall methodological soundness (YJNV, shKn).** Reviewers appreciated the clean framework, clarity of notation, and organization of the estimator in terms of local EIF construction, density-ratio reweighting, and federated aggregation.

- **Use of semiparametric efficiency and double robustness (YJNV).** The strengths of EIF-based derivations, double robustness, and the efficiency-theoretic foundation were positively noted, with particular emphasis on the completeness of the framework.

- **Simulation and real-data studies (YJNV, shKn).** Reviewers found the experimental results compelling for demonstrating consistency, validity under shift, and efficiency improvements in many settings.

- **Theoretical validity (shKn, n3Hz).** Both reviewers stated that the theory appears consistent, with solid derivations and correct use of the machinery from influence-function methods.

**Key concerns & Outlines of our responses**

**1. Relationship and novelty to prior work (FACE, collaborative learning, and meta-analysis) (Raised by: YJNV, shKn, n3Hz)**

Reviewers asked for clarification on:

- how the proposed method differs from FACE (Han et al., 2025),
- how the method compares to meta-analysis–style EIF aggregation.

Our revision:
- Extended the **Related Work** section (formerly “Limitations of existing work”).
- Explicitly compared our survival-specific EIF, pairwise partial CCOD assumptions, and efficiency characterization to FACE, highlighting   **important extensions** (pairwise CCOD, the efficiency bound, regularity conditions on density ratio).
- Discussed how and when our method reduces to a special case of FACE under a binary outcome setting without timing or censoring.
- Added a new baseline comparator from meta-analysis, **META-IVW,** in all experiments.
- Added discussion distinguishing our data-adaptive weighting from classical meta-analysis.

**2. Density-ratio estimation details (Raised by: YJNV, n3Hz)**

Questions focused on flexibility of models, exponential-tilt misspecification, overlap requirements, and the effect of strong covariate shift.

Our revision:
- Added a new Remark for the density-ratio estimation details, clarified model options, and removed the vague word “flexible”, and discussed model misspecification and how errors propagate.
- Clarified Assumption 2.3 (overlap), explained why it can be mild for survival data, and added simulation scenarios under limited target-site overlap, which showcase our method's performance advantage more clearly.
- Clarified that density ratios and EIFs remain well-defined under the boundedness assumptions in Condition E.2.

**3. Behavior of adaptive weighting and L1 penalty (Raised by: shKn)**

Concerns were raised that penalization could amplify errors in local estimates.

Our revision:

- Added an explanation in Section 2 clarifying that our methodological design incorporates a key safeguard: each source estimator is anchored to the target-site estimator, ensuring consistency first and then enabling efficiency gains.
- Added discussion that consistency is attainable in many practical settings, including randomized trials and large-sample observational studies.

**4. Clarifying what information is communicated across sites and the privacy scope (Raised by: n3Hz)**

Questions included whether the setting is truly federated and whether the privacy emphasis is appropriate.

Our revision:

- Clarified what is shared: Target-site survival model parameters and density-ratio summaries from target to source sites; EIFs are transmitted to the leading analysis center. Therefore, we justified our framework as truly federated.
- Softened and clarified the framing of “privacy-protected,” and changed the paper title to be “Federated Causal Survival Analysis Under Distribution Shift.”

**5. Experimental clarity and additional baseline comparisons (Raised by: n3Hz)**

Requests included more baseline comparators, clarity on real-data results, and demonstration of regimes with larger gains.

Our revision:

- Presented new simulation scenarios where FED shows **substantial efficiency gains** (importantly, in limited target-site propensity score overlap).
- Added META-IVW baseline across all experiments.

---

### Meta-Review · Area_Chair_nxew · 2025-12-16

**Summary:**

The summary of each reviewer's core comments and the authors' responses is provided below.

1. Reviewer YJNV (Score: 6, Confidence: 5): The reviewer requested (a) a comparison with the standard setting; and (b) different strategies on density ratio analysis. The authors satisfactorily responded.

2. Reviewer shKn (Score: 6, Confidence: 3): The reviewer indicated that the authors satisfactorily addressed their concerns and hence the reviewer increased the score, possibly to 8.

3. Reviewer n3Hz (Score: 4, Confidence: 4): The core comment by the reviewer would be "framing of the article around privacy". The authors decided to change the paper's title to address this criticism. I find the authors' responses satisfactory regarding the remaining concerns raised by the reviewer.

Overall, the paper feels improved significantly during the rebuttal period. Hence, I recommend accepting the paper.

**Reviewer Concerns:**

See above.

**Reviewer Scores:**

I think the most critical reviewer is Reviewer n3Hz (Score: 4, Confidence: 4). Although I cannot make a very accurate estimate of the final score by this reviewer, I personally feel the authors' responses are adequate, and the authors improved their paper significantly based on the reviewer's feedback. If I were the reviewer, the final score might have been either 6 or 8, which would have made the average score above 6.

---

### Decision · Program_Chairs · 2026-01-26

Accept (Poster)